# Analytic Insights into Structure and Rank of Neural Network Hessian Maps

Sidak Pal Singh* [a,b], Gregor Bachmann* [a] and Thomas Hofmann[a,b]

[a]ETH Zürich
[b]Max Planck ETH Center for Learning Systems

## Abstract

The Hessian of a neural network captures parameter interactions through second-order derivatives of the loss. It is a fundamental object of study, closely tied to various problems in deep learning, including model design, optimization, and generalization. Most prior work has been empirical, typically focusing on low-rank approximations and heuristics that are blind to the network structure. In contrast, we develop theoretical tools to analyze the range of the Hessian map, which provide us with a precise understanding of its rank deficiency and the structural reasons behind it. This yields exact formulas and tight upper bounds for the Hessian rank of deep linear networks — allowing for an elegant interpretation in terms of rank deficiency. Moreover, we demonstrate that our bounds remain faithful as an estimate of the numerical Hessian rank, for a larger class of models such as rectified and hyperbolic tangent networks. Further, we also investigate the implications of model architecture (e.g. width, depth, bias) on the rank deficiency. Overall, our work provides novel insights into the source and extent of redundancy in overparameterized neural networks. [1]

## 1 Introduction

Since the very infancy of neural networks, the Hessian matrix has been a central object of study. This is because the Hessian captures pairwise interactions of parameters via second-order derivatives of the loss function. As a result, the Hessian was productively employed, for instance, in (quasi-Newton) optimization methods [1, 2], model design and pruning [3–5], generalization [6], network calibration [7], automatically tuning hyper-parameters [8]. But, from the outset the main practical challenge has been its size, scaling quadratically with the model dimensionality. This makes the problem severe for today's DNNs which have millions or even billions of parameters [9, 10].

Consequently, most prior work has focused on designing scalable Hessian approximations, which either take the route of Hessian-vector products (R-operator) [11–13] or employ positive definite approximations by appealing to the Fisher information matrix. Additional approximations — without exception — are needed on top, such as diagonal approximations [3, 14] in the former or K-FAC [15–17], restricted to layerwise or arbitrary blocks on the diagonal [18–20], in the latter.

---

*Detailed list of contributions are: Sidak first discovered that the Hessian rank formula, in an early form, holds experimentally to high fidelity, thus kick-starting the project. Sidak came up with the proof technique and proved Theorems 3, 5, 9, 12. Sidak wrote essentially the entire paper and noted the rank-deficiency interpretation. Gregor proved Lemma 8, assisted in a part of Theorem 3, empirically observed the eventual formula for the Hessian rank, and essentially ran all the experiments for the final submission and made the corresponding figures. Correspondence to sidak.singh@inf.ethz.ch.

[1]The most recent version of our paper can be found at https://arxiv.org/abs/2106.16225 and the code corresponding to the experiments is located at https://github.com/dalab/hessian-rank.

35th Conference on Neural Information Processing Systems (NeurIPS 2021).

The goal of this paper is to advance the analytical understanding of the Hessian map of a neural network. We pursue the fundamental question of how the model architecture induces structural properties of the Hessian. In particular, we analyze the dimension of its range (i.e., the rank) and identify the sources of rank deficiency. Understanding the range of the Hessian map, in turn, delivers insights into the important aspect of how gradients change between iterations.

A reason why such an important direction currently remains sidelined is that non-linearities in a neural network result in an increased dependence on the data distribution, making a suitable theoretical analysis seem intractable. Following Saxe et al. [21], Kawaguchi [22], who delivered useful general insights on neural networks by looking at the linear case, we take a step back and rigorously characterize the range of the Hessian map and determine the resultant rank deficiency for deep linear networks. *The key result of our paper is an exact formula along with tight upper bound on the rank of the Hessian — which effectively depend on the sum of hidden-layer widths.* This stands opposed to the total number of parameters which are proportional to the sum of squared layer widths, thus implying a significant redundancy in the parameterization of neural networks (see Fig. 1).

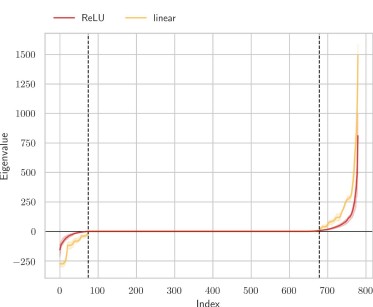

**Figure 1:** Hessian spectrum of linear and ReLU networks at initialization. Dashed lines indicate our rank predictions. Results have been averaged over 3 runs.

The exact quantification of the Hessian rank gives a precise yet interpretable ballpark on the inherent complexity of neural networks since rank naturally measures the effective number of parameters (as best illustrated by the case of a quadratic minimization problem). This relationship is further reinforced by connection to the classical complexity measure of Gull [23], MacKay [24], which is equivalent to rank for a sufficiently small constant controlling the prior. Therefore, this sheds a novel perspective on the nature and degree of overparameterization in neural networks, and opens up interesting avenues for future investigation.

**Contributions.** The main contributions of our paper can be summarized as follows: (i) Section 3: We characterize the structure of the Hessian range by exhibiting it via matrix derivatives. (ii) Section 4: We prove tight upper bounds and provide exact formulas on the Hessian rank that are neatly interpretable. *To the best of our knowledge, this is the very first time that such formulas and bounds are made available for neural networks.* (iii) Section 5.2: In the non-linear case, we show that our (linear) rank formulae faithfully capture the numerical rank. (iv) Section 6: We demonstrate via experiments and theory that such rank bounds also hold throughout the course of training. (v) Section 7.1: In the non-linear case, we also provide a pessimistic yet non-trivial bound, which provably establishes degeneracy of the Hessian at the minimum. (vi) Section 7.2, 5.1: We extend our rank results to the case of bias and investigate the effects of architectural components (such as width, depth, bias) on rank. (vii) Appendix S8: As a by-product, our analysis also reveals interesting properties of the Hessian spectrum, and we prove the presence of additional redundancies due to repeated eigenvalue plateaus.

**Related work.** The study of the Hessian, in recent times, has been re-invigorated by the empirical observations of Sagun et al. [25, 26] who noted a high degree of degeneracy experimentally, and characterized the spectrum as being composed of a bulk around zero and few outlier eigenvalues. Since then, works such as [27, 28] have scaled the empirical analysis to bigger networks via efficient spectral density calculations and better explained the observations in [26]. However, a drawback is that due to their empirical nature, it remains hard to uncover the exact level of degeneracy via these methods [29], and neither are the exact factors that affect the rank deficiency discerned. Another approach has been to fit models from random matrix theory to match the observed Hessian spectrum in neural networks [29, 30], however, such methods do not properly capture the sharp peak at zero observed in empirical spectra [31]. Yet others have investigated the properties of the spectrum in the asymptotic regime [32, 33], but the bounds, if any, are quite coarse (see Fig. 4). Moreover, a common issue underlying most of these approaches is that they are blind to the layerwise compositional structure of neural networks, and are often motivated by a black-box decomposition of the Hessian.

In regards to indicating the parameter redundancy in neural networks, we are far from being the first work. Prior empirical studies have long reported similar observations, like in the form

of, post-training pruning [3, 4, 19, 20, 34] or inherently contained sub-networks (Lottery Ticket Hypothesis [35]) or intrinsic dimension of the loss landscape [36]. Recently, [37] have experimentally argued for the effective dimensionality from [23, 24] to be a good predictor of double descent [38]. Nevertheless, determining the precise extent of redundancy — and *showing it provably* — as well as the structural reasons behind it have remained illusory in such approaches.

## 2 Setup and Formalism

**General notation.** We use the shorthand, $\mathbf{W}^{k:l}$, to refer to the matrix product chain $\mathbf{W}^k \cdots \mathbf{W}^l$, when $k > l$. When $k < l$, $\mathbf{W}^{k:l}$ will stand for the transposed product chain $\mathbf{W}^{k^\top} \cdots \mathbf{W}^{l^\top}$. Besides, $\otimes$ denotes the Kronecker product of two matrices, $\mathrm{vec}_r$ indicates the row-wise vectorization of matrices, rk refers to the rank of a matrix, and $\mathbf{I}_k$ is the identity matrix of size $k \times k$.

**Deep Neural Networks (DNNs).** A feedforward DNN is a composition of maps, i.e., $F = F^L \circ \cdots \circ F^1$, where the $l$-th layer map $F^l : \mathbb{R}^{M_{l-1}} \to \mathbb{R}^{M_l}$, with input dimension $d := M_0$, output dimension $K := M_L$, total number of hidden neurons $M := \sum_{l=1}^{L-1} M_l$. Each layer map is parameterized by a weight matrix $\mathbf{W}^l$ and applies an elementwise activation function $\sigma^l$. So we have,

$$F^l = \sigma^l \circ \mathbf{W}^l \text{ with } \mathbf{W}^l \in \mathbb{R}^{M_l \times M_{l-1}} \text{ and } \sigma^l : \mathbb{R} \to \mathbb{R}.$$

For the sake of tractability, we will often investigate linear DNNs where $\sigma^l = \mathrm{id}$, and so $F(\mathbf{x}) = \mathbf{W}^{L:1}\mathbf{x}$. For compactness, we also represent the entire set of parameters by $\boldsymbol{\theta} := \{\mathbf{W}^1, \cdots, \mathbf{W}^L\}$, and where emphasis requires, we will subscript the DNN map $F$ with it and write $F_{\boldsymbol{\theta}}$.

Next, assume that we are given a dataset $S = \{(\mathbf{x}_i, \mathbf{y}_i)\}_{i=1}^N$ of $N$ input-output pairs, drawn i.i.d from an underlying distribution $p_{\mathbf{x},\mathbf{y}}$. Our focus will be on the squared loss (MSE), $\ell_{\mathbf{x},\mathbf{y}}(\boldsymbol{\theta}) = \frac{1}{2}\|\mathbf{y} - \hat{\mathbf{y}}\|^2$ and its residual $\boldsymbol{\delta}_{\mathbf{x},\mathbf{y}} := \hat{\mathbf{y}} - \mathbf{y} = \frac{\partial \ell_{\mathbf{x},\mathbf{y}}}{\partial \hat{\mathbf{y}}}$, where $\hat{\mathbf{y}} = F_{\boldsymbol{\theta}}(\mathbf{x})$ is the DNN prediction. The population loss, $\mathcal{L}$ is: $\mathcal{L}(\boldsymbol{\theta}) = \mathbf{E}_{p_{\mathbf{x},\mathbf{y}}}[\ell_{\mathbf{x},\mathbf{y}}(\boldsymbol{\theta})]$. Finally, we will analyze the Hessian matrix $\mathbf{H}_{\mathcal{L}} = \frac{\partial^2 \mathcal{L}}{\partial \boldsymbol{\theta} \, \partial \boldsymbol{\theta}}$.

**Backpropagation in matrix derivatives.** As all parameters are collected into matrices, we often work with matrix-matrix derivatives by vectorizing *row-wise* in the numerator (Jacobian) layout, i.e., $\frac{\partial \mathbf{Y}}{\partial \mathbf{X}} := \frac{\partial \, \mathrm{vec}_r(\mathbf{Y})}{\partial \, \mathrm{vec}_r(\mathbf{X})^\top}$, see [39] and Appendix S2.5. Alongside this, we use the following rule:

$$\frac{\partial \mathbf{AWB}}{\partial \mathbf{W}} = \mathbf{A} \otimes \mathbf{B}^\top, \quad \text{e.g.} \quad \frac{\partial F}{\partial \mathbf{W}^k} = \mathbf{W}^{L:k+1} \otimes \mathbf{x}^\top \mathbf{W}^{1:k-1} \in \mathbb{R}^{K \times M_k M_{k-1}}. \tag{1}$$

By the usual chain rule (backpropagation) one has for a linear DNN, at a sample $(\mathbf{x}, \mathbf{y})$:

$$\frac{\partial \ell_{\mathbf{x},\mathbf{y}}}{\partial \mathbf{W}^k} = \underbrace{\left[\mathbf{W}^{k+1:L}\boldsymbol{\delta}_{\mathbf{x},\mathbf{y}}\right]}_{\text{backward } \in \mathbb{R}^{M_k}} \cdot \underbrace{\left[\mathbf{W}^{k-1:1}\mathbf{x}\right]^\top}_{\text{forward } \in \mathbb{R}^{M_{k-1}}} = \mathbf{W}^{k+1:L}\left[\mathbf{W}^{L:1}\mathbf{x}\mathbf{x}^\top - \mathbf{y}\mathbf{x}^\top\right]\mathbf{W}^{1:k-1}. \tag{2}$$

The above gradient with respect to $\mathbf{W}^k$ is of first order in $\mathbf{W}^k$ itself and second order in the other weight matrices. Lastly, let us setup the following shorthand, $\boldsymbol{\Omega} := \mathbf{E}\left[\boldsymbol{\delta}_{\mathbf{x},\mathbf{y}}\mathbf{x}^\top\right] = \mathbf{E}\left[\mathbf{W}^{L:1}\mathbf{x}\mathbf{x}^\top - \mathbf{y}\mathbf{x}^\top\right]$, $\boldsymbol{\Sigma}_{\mathbf{xx}} := \mathbf{E}\left[\mathbf{x}\mathbf{x}^\top\right]$, and $\boldsymbol{\Sigma}_{\mathbf{yx}} := \mathbf{E}\left[\mathbf{y}\mathbf{x}^\top\right]$, which we will use throughout the paper.

## 3 Hessian Maps of Linear DNNs

### 3.1 Hessian structure

The Hessian map has a natural block structure defined by the layers and their dimensionality. In order to leverage this structure, we directly take the derivative of the loss gradient in a matrix-by-matrix fashion. First, consider the $k$-th diagonal block of the Hessian, which is independent of $\mathbf{y}$ and is given by,

$$\mathbf{H}_{\mathcal{L}}^{kk} := \frac{\partial^2 \mathcal{L}}{\partial \mathbf{W}^k \partial \mathbf{W}^k} = \mathbf{W}^{k+1:L}\mathbf{W}^{L:k+1} \otimes \mathbf{W}^{k-1:1}\boldsymbol{\Sigma}_{\mathbf{xx}}\mathbf{W}^{1:k-1}.$$

This follows from the matrix-derivative rule in Eq. (1) along with the Eq. (2) and taking expectation. The calculation of the off-diagonal Hessian blocks ($kl$-th block of size $M_k M_{k-1} \times M_l M_{l-1}$)

involves the product rule. Note that the two occurrences of a weight matrix $\mathbf{W}^l$, in Eq. (2), are once non-transposed (in $\mathbf{W}^{L:1}\mathbf{x}\mathbf{x}^\top$) and once transposed (in $\mathbf{W}^{k+1:L}$ or $\mathbf{W}^{1:k-1}$ respectively for $k < l$ or $k > l$). For simplicity, let us express these parts without adding them and directly write the other Hessian contribution with respect to the *transposed matrix*, giving:

$$\forall\, k \neq l, \quad \widetilde{\mathbf{H}}_{\mathcal{L}}^{kl} := \frac{\partial^2 \mathcal{L}}{\partial \mathbf{W}^l \partial \mathbf{W}^k} = \mathbf{W}^{k+1:L}\mathbf{W}^{L:l+1} \;\otimes\; \mathbf{W}^{k-1:1}\,\mathbf{\Sigma}_{\mathbf{xx}}\,\mathbf{W}^{1:l-1}\,. \tag{3}$$

$$\forall\, k < l \quad \widehat{\mathbf{H}}_{\mathcal{L}}^{kl} := \frac{\partial^2 \mathcal{L}}{\partial \mathbf{W}^{l^\top} \partial \mathbf{W}^k} = \mathbf{W}^{k+1:l-1} \;\otimes\; \mathbf{W}^{k-1:1}\,\mathbf{\Omega}^\top\,\mathbf{W}^{L:l+1}\,. \tag{4}$$

$$\forall\, k > l \quad \widehat{\mathbf{H}}_{\mathcal{L}}^{kl} := \frac{\partial^2 \mathcal{L}}{\partial \mathbf{W}^{l^\top} \partial \mathbf{W}^k} = \mathbf{W}^{k+1:L}\,\mathbf{\Omega}\,\mathbf{W}^{1:l-1} \;\otimes\; \mathbf{W}^{k-1:l+1}\,. \tag{5}$$

**Equivalence to Gauss-Newton Decomposition.** A common approach is to look at the Hessian map from the perspective of the Hessian chain rule, where we have that, $\mathbf{H}_{\mathcal{L}} = \mathbf{H}_o + \mathbf{H}_f$, with

$$\mathbf{H}_o = \mathbf{E}_{\,p_{\mathbf{x},\mathbf{y}}}\left[\nabla_{\boldsymbol{\theta}} F(\mathbf{x})^\top \,[\partial^2 \ell_{\mathbf{x},\mathbf{y}}]\,\nabla_{\boldsymbol{\theta}} F(\mathbf{x})\right]\,, \text{ and } \quad \mathbf{H}_f = \mathbf{E}_{\,p_{\mathbf{x},\mathbf{y}}}\left[\sum_{c=1}^{K}[\partial \ell_{\mathbf{x},\mathbf{y}}]_c\,\nabla_{\boldsymbol{\theta}}^2 F_c(\mathbf{x})\right]\,.$$

For the MSE loss, the Gauss-Newton decomposition is in fact equivalent to what we discussed before (see details in Appendix S1), where $\mathbf{H}_o$ contains the blocks $\mathbf{H}_{\mathcal{L}}^{kk}$ and $\widetilde{\mathbf{H}}_{\mathcal{L}}^{kl}$, while $\mathbf{H}_f$ consists of $\widehat{\mathbf{H}}_{\mathcal{L}}^{kl}$ (although with the non-transposed matrix). Henceforth, we will refer to the first term $\mathbf{H}_o$ as the *outer-product Hessian*, while we coin the second-term $\mathbf{H}_f$ as the *functional Hessian*.

### 3.2 Range of the Hessian map

Assuming a local Taylor-series approximation of the loss' gradient, we have that for $\|\Delta\boldsymbol{\theta}\| < \epsilon$, $\nabla_{\boldsymbol{\theta}+\Delta\boldsymbol{\theta}}\,\mathcal{L} \approx \nabla_{\boldsymbol{\theta}}\mathcal{L} + \mathbf{H}_{\mathcal{L}}\,\Delta\boldsymbol{\theta}$. This indicates how the gradients will change over any local perturbation $\Delta\boldsymbol{\theta}$. Also, this holds over successive iterations of an optimization algorithm such as gradient descent. As a result, this serves to show the significance of the Hessian range as since its dimension (i.e., rank) will dictate the dimension of the space where gradients evolve — something which has been discussed only empirically before [40].

Moving on, to best highlight the layer-wise structure present in the Hessian range, let us multiply the two parts of the Hessian with a vector $\Delta\boldsymbol{\theta}$, which we decompose as $\Delta\boldsymbol{\theta} = \left[\text{vec}_r(\Delta\mathbf{W}^1)^\top \;\ldots\; \text{vec}_r(\Delta\mathbf{W}^L)^\top\right]^\top$.

**Range of the outer-product $\mathbf{H}_o$** The product of the $k$-th row-block $\mathbf{H}_o^{k\bullet}$, corresponding to the $k^{\text{th}}$ layer, with $\Delta\boldsymbol{\theta}$ can be written succinctly as, $\mathbf{H}_o^{k\bullet} \cdot \Delta\boldsymbol{\theta} = \text{vec}_r\left(\mathbf{W}^{k+1:L}\cdot\bar{\Delta}\cdot\mathbf{\Sigma}_{\mathbf{xx}}\mathbf{W}^{1:k-1}\right)$, where $\bar{\Delta} := \sum_{l=1}^{L}\mathbf{W}^{L:l+1}\,\Delta\mathbf{W}^l\,\mathbf{W}^{l-1:1} \in \mathbb{R}^{K\times d}$. This essentially follows from the identity, $\text{vec}_r\,\mathbf{A}\mathbf{X}\mathbf{B} = (\mathbf{A}\otimes\mathbf{B}^\top)\,\text{vec}_r\,\mathbf{X}$ (see proof in Appendix S2.5). Note that $\bar{\Delta}$ represents the net change on the prediction map induced by changes to the weight matrices in the forward pass. We see that: (1) $\Delta\boldsymbol{\theta}$ is linearly compressed into a highly interpretable matrix $\bar{\Delta}$ with $Kd$ entries. (2) The same compressed $\bar{\Delta}$ is shared across all result blocks as it is independent of the row block index $k$. Overall, this already hints that there is a significant intrinsic structure in the Hessian that constrains its range.

**Range of the functional Hessian $\mathbf{H}_f$** Here we multiply with $\widehat{\Delta\boldsymbol{\theta}}$, which is similar to $\Delta\boldsymbol{\theta}$ except that we consider $\Delta\mathbf{W}^{l^\top}$ instead of $\Delta\mathbf{W}^l$. Then the product corresponding to $k^{\text{th}}$ layer is, $\mathbf{H}_f^{k\bullet} \cdot \widehat{\Delta\boldsymbol{\theta}} = \text{vec}_r\left(\mathbf{W}^{k+1:L}\,\mathbf{\Omega}\,[\Delta^{<k}]^\top + [\Delta^{>k}]^\top\,\mathbf{\Omega}\,\mathbf{W}^{1:k-1}\right)$, where $\Delta^{<k} := \sum_{l=1}^{k-1}\mathbf{W}^{k-1:l+1}\,\Delta\mathbf{W}^l\,\mathbf{W}^{l-1:1}$ and $\Delta^{>k} := \sum_{l=k+1}^{L}\mathbf{W}^{L:l+1}\,\Delta\mathbf{W}^l\,\mathbf{W}^{l-1:k+1}$. Notice that the range of $\mathbf{H}_f$ is also inherently constrained like in the case of $\mathbf{H}_o$, but there are two important differences. First, the data dependent part is now the covariance $\mathbf{\Omega} = \mathbf{E}\left[\boldsymbol{\delta}_{\mathbf{x},\mathbf{y}}\,\mathbf{x}^\top\right]$. Clearly, if this matrix is low rank (say $s$), this will directly impact the rank of $\mathbf{H}_f$. The other significant difference is that the weight matrix product is split at layer $k$. This reflects the fact that upstream and downstream layers have a different effect. We will see ahead that these factors will neatly gives rise to a dependence on the sum of hidden-layer widths.

# 4 Main result: Analysis of the Hessian rank

**Preliminaries.** Let us denote the rank of the uncentered covariance $\boldsymbol{\Sigma_{xx}} = \mathbf{E}\left[\mathbf{xx}^\top\right]$ by $r$. If $r < d$, then without loss of generality, consider $\boldsymbol{\Sigma_{xx}} := (\boldsymbol{\Sigma_{xx}})_{r \times r}$, which is always possible by pre-processing the input. Thus, $\boldsymbol{\Sigma_{xx}} \succ 0$, always. Also, in such a case, we take $\mathbf{W}^1 \in \mathbb{R}^{M_1 \times r}$. Further, the only assumption we make in our analysis is A1, which is in fact guaranteed at typical initialization almost surely (c.f. Appendix S2.6). In Section 6, we see what happens while training.

**Assumption A1.** *Maximal Rank:* $\forall l \in [L]$, $\mathbf{W}^l \in \mathbb{R}^{M_l \times M_{l-1}}$ *has rank equal to* $\min(M_l, M_{l-1})$.

Lastly, the omitted proofs in the coming subsections are located in Appendices S2, S3, S4 respectively.

## 4.1 Analytical tool

The key idea of our analysis technique is to reduce the rank of involved matrices to the rank of a certain special kind of matrix (or its variant), $\mathbf{Z}$, as shown below:

$$\mathbf{Z} = \begin{pmatrix} \mathbf{I}_q \otimes \mathbf{C} \\ \mathbf{D} \otimes \mathbf{I}_n \end{pmatrix}, \quad \text{with } \mathbf{C} \in \mathbb{R}^{m \times n}, \ \mathbf{D} \in \mathbb{R}^{p \times q}. \tag{6}$$

This row-partitioned matrix has a characteristic structure, where an identity matrix alternates between the two sides of the Kronecker product. Such matrices are in fact omnipresent in the Hessian structure, and importantly for our purpose, they possess additional properties on their rank. Inherent to these properties and our analysis, is the use of generalized inverse [41] and oblique (non-orthogonal) projector matrices. The following Lemma 1 from [42] details such a result:

**Lemma 1.** *Let* $\mathbf{Z}$ *be a matrix as in Eq.* (6)*. Then,* $\mathrm{rk}(\mathbf{Z}) = q \, \mathrm{rk}(\mathbf{C}) + n \, \mathrm{rk}(\mathbf{D}) - \mathrm{rk}(\mathbf{C}) \, \mathrm{rk}(\mathbf{D})$.

## 4.2 Rank of the outer-product Hessian

Consider the following decomposition of $\mathbf{H}_o$, which reveals its 'outer-product' nature:

**Proposition 2.** *For a deep linear network,* $\mathbf{H}_o = \mathbf{A}_o \mathbf{B}_o \mathbf{A}_o^\top$, *where* $\mathbf{B}_o = \mathbf{I}_K \otimes \boldsymbol{\Sigma_{xx}} \in \mathbb{R}^{Kd \times Kd}$, *and* $\mathbf{A}_o^\top = \left( \mathbf{W}^{L:2} \otimes \mathbf{I}_d \quad \cdots \quad \mathbf{W}^{L:l+1} \otimes \mathbf{W}^{1:l-1} \quad \cdots \quad \mathbf{I}_K \otimes \mathbf{W}^{1:L-1} \right) \in \mathbb{R}^{Kd \times p}$,

where $p$ is the number of parameters. A straightforward consequence is that if there is no bottleneck in between, i.e., no hidden-layer with width $M_i < \min(K, d)$, then the matrix $\mathbf{B}_o$ will control the rank of $\mathbf{H}_o$. Hence, as a first upper bound we get, $\mathrm{rk}(\mathbf{H}_o) \leq \min\left(\mathrm{rk}(\mathbf{A}_o), \mathrm{rk}(\mathbf{B}_o)\right) = \mathrm{rk}(\mathbf{B}_o) = \mathrm{rk}(\mathbf{I}_K) \, \mathrm{rk}(\boldsymbol{\Sigma_{xx}}) = Kr$.

Such a decomposition is however not new (see [43]), but this only forms an initial step of our analysis and the current bound can be loose in the bottleneck case (e.g., an auto-encoder). Let us define the minimum dimension to be $q := \min(r, M_1, \cdots, M_{L-1}, K)$. Our main theorem can then be stated as:

**Theorem 3.** *Consider the matrix* $\mathbf{A}_o$ *mentioned in Proposition* 2*. Under the assumption A1,* $\mathrm{rk}(\mathbf{A}_o) = r \, \mathrm{rk}(\mathbf{W}^{2:L}) + K \, \mathrm{rk}(\mathbf{W}^{L-1:1}) - \mathrm{rk}(\mathbf{W}^{2:L}) \, \mathrm{rk}(\mathbf{W}^{L-1:1}) = q \, (r + K - q)$.

Now, from Theorem 3, it is evident that we can simply upper bound the rank of $\mathbf{H}_o$, by the rank of $\mathbf{A}_o$. But actually, it is possible to show an equality (using Lemma 18), as described below:

**Corollary 4.** *Under the setup of Theorem* 3*, the rank of* $\mathbf{H}_o$ *is given by* $\mathrm{rk}(\mathbf{H}_o) = q \, (r + K - q)$.

**Remark.** It should be emphasized that this analysis not only delivers the rank of the outer-product Hessian but also of Neural Tangent Kernel [44] (see Appendix S9.5), Fisher information matrix, network Jacobian, due to their underlying intimate relationship, unaltered through the lens of rank.

## 4.3 Rank of the functional Hessian

For analyzing the rank of the functional Hessian $\mathbf{H}_f$, we will continue operating by having one of the derivatives with respect to a transposed weight matrix (here, $\mathbf{W}^{l^\top}$), since rank does not change with column or row permutations. We denote this modification of the functional Hessian by $\widehat{\mathbf{H}}_f$. Then from Eqs. (4, 5), we can observe that there is a common structure within the blocks contained in the column $l$. Namely, all the weight matrix-chains have either $l - 1$ or $l + 1$ as their right indices. So our approach will be to bound the rank of the individual column blocks, as formalized below:

**Theorem 5.** *For a deep linear network, the rank of $l$-th column-block, $\widehat{\mathbf{H}}_f^{\bullet l}$, of the matrix $\widehat{\mathbf{H}}_f$, under the assumption A1 is given as* $\mathrm{rk}(\widehat{\mathbf{H}}_f^{\bullet l}) = \widehat{q}\, M_{l-1} + \widehat{q}\, M_l - \widehat{q}^2$, *for $l \in [2, \cdots, L-1]$. When $l = 1$, we have* $\mathrm{rk}(\widehat{\mathbf{H}}_f^{\bullet 1}) = \widehat{q}\, M_1 + \widehat{q}\, s - \widehat{q}^2$. *And, when $l = L$, we have* $\mathrm{rk}(\widehat{\mathbf{H}}_f^{\bullet L}) = \widehat{q}\, M_{L-1} + \widehat{q}\, s - \widehat{q}^2$. *Here, $\widehat{q} := \min(r, M_1, \cdots, M_{L-1}, K, s) = \min(q, s)$ and $s := \mathrm{rk}(\boldsymbol{\Omega}) = \mathrm{rk}(\mathbf{E}\,[\boldsymbol{\delta}_{\mathbf{x},\mathbf{y}}\, \mathbf{x}^{\top}])$.*

The upper bound on the rank of $\mathbf{H}_f$ follows by combining the above result over all the columns:

**Corollary 6.** *Under the setup of Theorem 5, the rank of $\mathbf{H}_f$ can be upper bounded as,* $\mathrm{rk}(\mathbf{H}_f) \le 2\,\widehat{q}\, M + 2\,\widehat{q}\, s - L\,\widehat{q}^2$, *where* $M = \sum_{i=1}^{L-1} M_i$.

Although the above result is an upper bound, empirically this is precisely the formula at initialization, and thus we have the *tightest upper bound on the rank of the functional Hessian*. Besides, in general, we find that at initialization $s = \min(r, K)$, hence $\widehat{q} = q$ and *we will just use $q$ hereafter for simplicity*.

**Block-column independence.** A surprising element of the above analysis is that just adding the ranks of the block-columns, corresponding to the respective layers, gives the rank of the entire $\mathbf{H}_f$ which is tight. This phenomenon is quite straightforward to see in a 2-layer network and there Corollary 6 is an equality. However, the more interesting observation is that this holds even for arbitrary depth and also extends to networks *with non-linearities* such as ReLU and Tanh. This implies that the column spaces associated with the layerwise block-columns do not overlap. Thus, $\mathbf{H}_f$ *is similar to a block diagonal matrix*, and it should be possible to uncover the similarity transformation. But this is beyond the current scope, and we leave it as an open question.

### 4.4 Overall bound on the Hessian Rank

Finally, in order to get an upper bound on the rank of the entire Hessian, we simply use $\mathrm{rk}(\mathbf{A} + \mathbf{B}) \le \mathrm{rk}(\mathbf{A}) + \mathrm{rk}(\mathbf{B})$, along with the Corollary 4 and Corollary 6 (with $\widehat{q} = q$ as noted above), to obtain:

$$\mathrm{rk}(\mathbf{H}_{\mathcal{L}}) \le \mathrm{rk}(\mathbf{H}_o) + \mathrm{rk}(\mathbf{H}_f) \le 2\,q\,M - L\,q^2 + 2\,q\,s + q\,(r + K - q).$$

**Fact 7.** *The following equality holds empirically:* $\boxed{\mathrm{rk}(\mathbf{H}_{\mathcal{L}}) = 2q\,M - L\,q^2 + q(r + K)}$.

This implies that our upper bound is off by just a constant additive factor of $2\,q\,s - q^2$, which is another startling finding. Thus, revealing that the *intersection of the $\mathbf{H}_o$, $\mathbf{H}_f$ column spaces has a rather small dimension*. E.g., take the typical case of no bottlenecks, $q = s$, then our upper bound exceeds the true rank by a small constant $q^2$, i.e., minimum dimension squared. This suggests that the direct sum of their column spaces is not too far-fetched as an approximation to the column space of $\mathbf{H}_{\mathcal{L}}$. Previously, [30] empirically noted a similar observation for 1-hidden layer networks and [33] showed a high degree of mutual orthogonality in the asymptotic regime. Our result shows a similar consequence in the finite case.

**Alternative interpretation.** Besides the above result, that the rank of the Hessian is proportional to the sum of widths, there is an alternate way of viewing this. Let us calculate the rank deficiency in the network when (uncentered) input-covariance has rank, i.e., $r = d$. Using Fact (7) this comes out to $\text{rank-deficiency}(\mathbf{H}_{\mathcal{L}}) = \sum_{i=0}^{L-1} (M_i - q)(M_{i+1} - q)$, whereas the number of parameters is equal to $p = \sum_{i=0}^{L-1} M_i\, M_{i+1}$. Hence this lends an elegant interpretation to our formula, whereby the amount of rank deficiency is equal to the number of parameters of a hypothetical network whose all layer-widths have been subtracted by the minimum layer-width of the original network.

### 4.5 Hessian Rank as effective # of parameters

Consider a quadratic optimization problem, $\mathrm{argmin}_{\boldsymbol{\theta} \in \mathbb{R}^p}\, \boldsymbol{\theta}^{\top}\, \mathbf{M}\, \boldsymbol{\theta}$, where assume the matrix $\mathbf{M}$ has a reduced rank $r < p$ and is in rank normal form, $\mathbf{M} = [\mathbf{M}_r; \mathbf{0}]$. As a result, we get an equivalent but reduced problem $\mathrm{argmin}_{\boldsymbol{\theta}_r \in \mathbb{R}^r}\, \boldsymbol{\theta}_r^{\top}\, \mathbf{M}_r\, \boldsymbol{\theta}_r$ of size $r < p$, where $\boldsymbol{\theta}_r$ is the vector containing the first $r$ parameters. More generally, we can reparameterize $\boldsymbol{\theta} \mapsto \mathbf{U}^{\top}\boldsymbol{\theta}$ where $\mathbf{U}$ contains the (top-$r$) eigenvectors of $\mathbf{M}$. Thus, here it is strikingly clear that rank provides the effective number of parameters. Further, we can go to the case of neural networks by simply utilizing the 2$^{\text{nd}}$ order Taylor series expansion of the loss $\mathcal{L}$ — and so the Hessian $\mathbf{H}_{\mathcal{L}}$ plays the role of $\mathbf{M}$. Overall, this suggests that the above results on the Hessian rank can provide insights into the effective # of parameters in neural networks, or in other words, into the inherent complexity of neural networks.

# 5 Empirical Results

## 5.1 Verification of Rank formulas and their behaviour

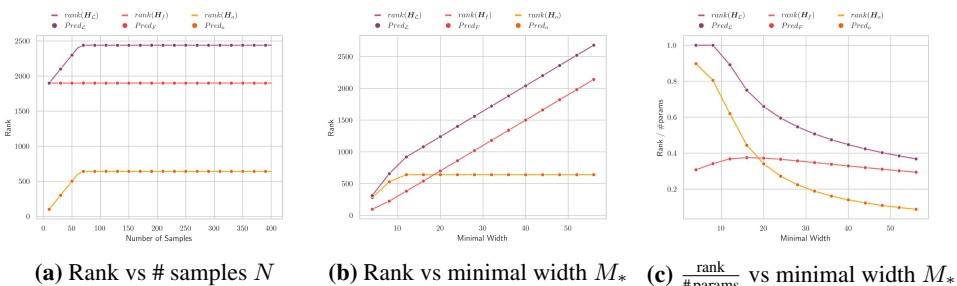

**(a)** Rank vs # samples $N$    **(b)** Rank vs minimal width $M_*$    **(c)** $\frac{\text{rank}}{\#\,\text{params}}$ vs minimal width $M_*$

**Figure 2:** Behaviour of rank and rank/#params on CIFAR10 using MSE, with hidden layers: $50, 20, 20, 20$ (Fig. 2a) and $M_*, M_*$ (Fig. 2b, 2c). The lines indicate the true value and circles denote our formula predictions.

**Setup.** We test our results on a *variety of datasets*: MNIST [45], FashionMNIST [46], CIFAR10 [47]; for *various loss types*: MSE, cross-entropy, cosh; across *several initialization schemes*: Glorot [48], uniform, orthogonal [21]. Our theory extends to all these settings. However, due to space constraints we only show a subset of experiments here, but the rest can be found in the Appendix S9.

**Procedure.** To verify the prediction of our theoretical results, we perform an exact calculation of the rank by computing the full Hessian and the corresponding singular value decomposition (SVD). The available iterative schemes for rank approximation (c.f. [49]), although more memory-efficient, are only effective for well-separated spectra and thus cannot provide useful approximations in the case of neural network Hessians. Besides the exact Hessian computation, we also utilize FLOAT-64 precision to ensure 'true' rank calculation, resulting in increased memory costs. Hence, we downscale the image resolution to $d = 64$ to test on more realistic networks.

**Results.** We study how the rank varies as a function of the sample size $N$ and the network architecture (for varying widths). Fig. 2 shows this for a linear network on CIFAR10 with MSE loss. First, in Fig 2a, we observe that our predictions match the true rank exactly across all sample sizes as the dependence on $N$ is only exhibited in the rank of the empirical covariance $\widehat{\Sigma}_{\mathbf{xx}}$, confirming that rank is largely a distribution-independent quantity. So, for the rest of our experiments, we sufficiently subsample to ensure that the empirical and true covariance have the same rank. Next, in Fig. 2b, we see that *our rank formulas hold for arbitrary-sized network architectures*. To contextualize the growth of rank with increasing width, in Fig. 2c, we normalize it by # params $p$. We notice that rank/# params, which intuitively captures the fraction of effective parameters, saturates down to a small level — *signalling the extent of redundancy present in the network parameterization* (similar plots across depth and simulation of rank/# params on larger settings can be found in S9.1, S9.2).

## 5.2 The case of non-linearities

Although the linear nature of the neural network was crucial to our analysis, in this section we show experimentally that our results also extend to the non-linear setting — as numerical rank [50].

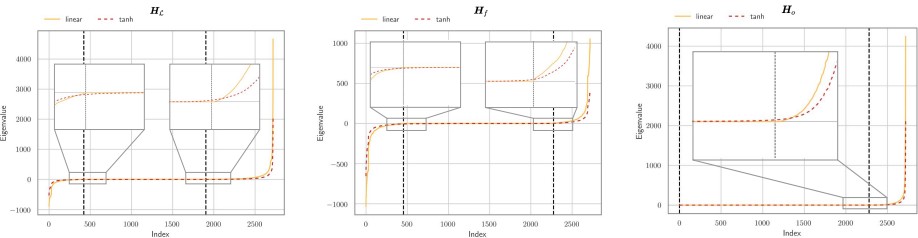

**Figure 3:** Spectrum of the loss Hessian $\mathbf{H}_{\mathcal{L}}$ (left), functional Hessian $\mathbf{H}_f$ (middle) and outer product $\mathbf{H}_o$ (right), for **linear** and **non-linear** networks. Black dashed lines are the predictions of the bulk size via our rank formulas. We use 2 hidden layers of size $30, 20$ with Tanh activation on MNIST.

**Visual comparison of the Hessian spectra.** Let us first understand how non-linearities affect the Hessian spectrum relative to the spectrum of linear networks. Fig. 3 compares the spectra of $\mathbf{H}_{\mathcal{L}}$, $\mathbf{H}_f$, $\mathbf{H}_o$ in these two scenarios (linear vs Tanh), with a zoomed-in inset near the cut-off obtained from rank formulas corresponding to the linear case. We can observe the presence of numerous tiny, but not exactly zero, eigenvalues in the non-linear case. So, if we were to measure the rank with a threshold up to machine precision — as we did in the linear case — this would result in an inflation of the rank measurement for the non-linear scenario. From a practical point of view, tiny but non-zero eigenvalues hold little significance, so a more relevant quantity is the numerical rank [50] that uses a reasonable threshold to weed out such extraneous eigenvalues. The numerical rank, or alternatively the size of bulk around zero, for the non-linear case, indeed seems to be captured by our (linear) rank formulas to high fidelity, as evident from Fig. 3. Similar results for other non-linearities, loss functions, datasets can be found in the Appendix S9.4.

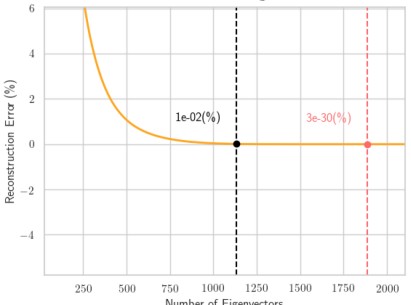

**Quantitative measure of the fidelity of Rank formulas.** To complement the visual grounds presented above, we now quantitatively measure the (in)significance of such spuriously tiny eigenvalues, in terms of the reconstruction error incurred by excluding them. Hence, we perform a low-rank approximation via the SVD and measure the (relative) reconstruction error at the value of rank from our formulas of the linear case. Fig. 4 displays the reconstruction error as a function of the # of top eigenvectors employed, for a ReLU network. As a reference, we consider the empirical rank measurement obtained at machine precision in this case.

*We find that using the linear rank value provides an excellent reconstruction, hence demonstrating the fidelity of our rank formulas to serve as a measure of numerical rank in the non-linear case.* The same observation extends to other non-linearities and losses, which we highlight in the Appendix S9.3. Consequently, iterative Hessian estima-

**Figure 4:** Hessian reconstruction error for a **ReLU** network of hidden layer sizes $30, 20$ with cross-entropy loss, as the rank of the approximation is increased. The dashed vertical lines indicate the cut-off at various values of the rank: **first line** at the prediction based on the linear model, **second line** at the empirical measurement of rank.

tion procedures, e.g. in second-order optimization methods [16, 51], could benefit from the linear rank as a guiding criterion for their design of Hessian approximation. Besides, these experiments also indicate that previous bounds, such as those by Jacot et al. [33], on the rank of outer-product Hessian and functional Hessian are quite coarse to be of much use (i.e., even > # parameters). This is because these bounds have a linear dependence on the product of: # of samples $N$ and # of classes $K$. For the same reason, other works [16, 52] that bound the rank of the outer-product Hessian $\mathbf{H}_o$, trivially, based on the # of samples $N$ are of little use.

## 6 Evolution of Rank during training

The upper bounds on the Hessian rank detailed before inherently depend on the rank of the weight matrices. While initialization guarantees them to be of maximal rank, the rank of weight matrices might possibly decrease during training, thus bringing about a decrease in the Hessian rank. Under some additional assumptions, Lemma 8 shows that this does not happen and the rank remains constant.

**Lemma 8.** *For a deep linear network, consider the gradient flow dynamics* $\dot{\mathbf{W}}_t^l = -\eta\nabla_{\mathbf{W}^l}\mathcal{L}_S(\boldsymbol{\theta})\big|_{\boldsymbol{\theta}=\boldsymbol{\theta}_t}$. *Assume: (a) Centered classes:* $\frac{1}{N}\sum_{i:y_{ic}=1}^{N}\mathbf{x}_i = \mathbf{0}, \ \forall c \in [1,\ldots,K]$. *(b) Balancedness at initialization:* $\mathbf{W}_0^{k+1^\top}\mathbf{W}_0^{k+1} = \mathbf{W}_0^k\mathbf{W}_0^{k^\top}$. *(c) Square weight-matrices:* $\mathbf{W}^l \in \mathbb{R}^{M\times M}, \ \forall l$ *and* $K = d = M$. *Then for all layers* $l$, $\mathrm{rk}(\mathbf{W}_t^l) = \mathrm{rk}(\mathbf{W}_0^l), \quad \forall t < \infty$.

Balancedness is a common assumption that has been used in many previous works, like Arora et al. [53]. Centered classes can easily be enforced via a pre-processing step, however empirically this is not required. While the proof (see Appendix S5) holds for square case, empirically we also find this to be true for non-square matrices (Fig. 5, left) and non-linearities as shown in Fig. S2.

**Consequence.** An implication of this result is that *our upper bounds on the Hessian rank remain valid throughout the training.* Even if the rank of the weight matrices were to decrease, say in the rectangular case, the Hessian rank would only decrease and our bounds would still be valid. The other avenue for a decrease in rank is learning-driven, and as $\mathbf{\Omega} = \mathbf{E}\left[\boldsymbol{\delta}_{\mathbf{x},\mathbf{y}}\,\mathbf{x}^\top\right] \to 0$, the functional Hessian

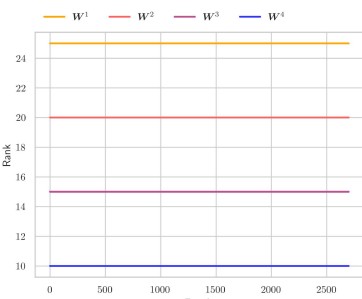 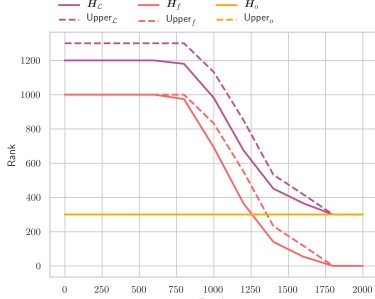

**Figure 5:** Rank dynamics for a linear network with hidden layer sizes 25, 20, 15, trained using a linear teacher. We show the evolution of **(left)** the rank of the weights with usual Glorot initialization (guaranteeing maximal rank) and **(right)** the rank of the Hessians, alongside the upper bounds, as a function of training time.

$\mathbf{H}_f \to \mathbf{0}$ (as it has an explicit dependence on the residual $\boldsymbol{\delta}_{\mathbf{x},\mathbf{y}}$). So, if the learning architecture is powerful enough, the rank of $\mathbf{H}_f$ will approach $0$, as well as leading to a decrease in the rank of $\mathbf{H}_\mathcal{L}$, and so $\mathrm{rk}(\mathbf{H}_\mathcal{L}) \to \mathrm{rk}(\mathbf{H}_o)$. In other words, the loss Hessian is completely captured through the outer product Hessian at convergence. We demonstrate this empirically via Fig. 5 (right) where we train a linear network using a linear teacher and visualize the rank dynamics as well as our upper bounds. We provide more details as well as experiments for the non-linear case in the Appendix S5.

## 7 Further results

### 7.1 Provable Hessian degeneracy with non-linearities

From Section 5.2, it is clear that our upper bounds faithfully estimate the numerical rank in the non-linear case, despite the numerous tiny eigenvalues which cause inflated rank measurements. This might raise the question if it is possible to establish, in this non-linear case, that the Hessian has provable degeneracy and not just approximate? The challenge is that here the data distribution manifests additionally via the activations at each layer, thus a theoretical analysis seems intractable without imposing strong assumptions. For a 1-hidden layer network, $F(\mathbf{x}) = \mathbf{W}^2\sigma(\mathbf{W}^1\mathbf{x})$, we show it is still possible to get a pessimistic yet non-trivial upper bound with the following (mild) assumption:

**Assumption A2.** *For each active hidden neuron $i$, the weighted input covariance has the same rank as the overall input covariance, i.e., $\mathrm{rk}(\mathbf{E}\left[\alpha_{\mathbf{x}}\mathbf{x}\mathbf{x}^\top\right]) = \mathrm{rk}(\mathbf{\Sigma}_{\mathbf{xx}}) = r$, with $\alpha_{\mathbf{x}} = \sigma'(\mathbf{x}^\top\mathbf{W}^1_{i\bullet})^2$.*

**Theorem 9.** *Consider a 1-hidden layer network with non-linearity $\sigma$ such that $\sigma(z) = \sigma'(z)z$ and let $\widetilde{M}$ be the # of active hidden neurons (i.e., probability of activation $> 0$). Then, under assumption A1 and A2, rank of $\mathbf{H}_o$ is given as, $\mathrm{rk}(\mathbf{H}_o) \le r\widetilde{M} + \widetilde{M}K - \widetilde{M}$.*

*Assumption A2 is rather mild*, i.e., in the finite-sample case, it holds as soon as the # of samples $N > 2d$, for typical initialization of parameter weights. Besides, the class of non-linearities which satisfy the above-mentioned condition includes e.g., *ReLU, Leaky-ReLU*. Further, this result extends to $\mathbf{H}_\mathcal{L}$:

**Corollary 10.** *At convergence to the minimum, the rank of the loss Hessian $\mathbf{H}_\mathcal{L}$, for the same setup as Theorem 9, is upper bounded by: $\mathrm{rk}(\mathbf{H}_\mathcal{L}) \le r\widetilde{M} + \widetilde{M}K - \widetilde{M}$.*

Contrast this with the view from [54], who claim the spectrum to be generically non-degenerate. Or, unlike [55], we establish this without any assumptions on a particular kind of overparameterization.

**Fact 11.** *For multiple hidden-layers, the following generalization of Theorem 9 holds empirically, $\mathrm{rk}(\mathbf{H}_o) \le p - M_1(d-r) - \sum_{i=1}^{L-1} M_i$, where $p$ is the # of parameters and assuming no dead neurons.*

While these bounds are likely to be quite loose as noticeable from Section 5.2, *but more importantly they help establish provable degeneracy of the Hessian at the minimum, with the number of 'absolutely-flat' directions (i.e., those in the Hessian null space) in proportion to the sum of hidden-layer sizes.*

### 7.2 Effect of bias on the rank of Hessian

Now, we see how the Hessian rank changes when bias is enabled throughout a deep linear network. We make the following simplifying assumption, which is actually a standard convention in practice.

**Assumption A3.** *The input data has zero mean, i.e., $\mathbf{x} \sim p_{\mathbf{x}}$ is such that $\mathbf{E}\left[\mathbf{x}\right] = 0$.*

> **Theorem 12.** *Under the assumption A1 and A3, for a deep linear network with bias, the rank of $\mathbf{H}_o$ is upper bounded as, $\mathrm{rk}(\mathbf{H}_o) \leq q(r + K - q) + K$, where $q := \min(r, M_1, \cdots, M_{L-1}, K)$.*

The proof can be found in the Appendix S7. Empirically, we do not require the input to be mean zero and our upper bound actually holds with equality. Also, we list rank formulas for functional Hessian and the overall loss Hessian in Appendix S7, in the non-bottleneck case. E.g.,

**Fact 13.** $\mathrm{rk}\left(\mathbf{H}_{\mathcal{L}}\right) = 2q'M + q'(r+K) - Lq'^2 + Lq'$, *where* $q' := \min(r+1, M_1, \cdots, M_{L-1}, K)$.

Here too, rank deficiency has a cleaner interpretation of being equal to the # of parameters in a hypothetical network with bias enabled, but with the minimum dimension $q'$ (that reflects the homogeneous coordinate at input) subtracted: $\mathrm{rank\text{-}deficiency}(\mathbf{H}_{\mathcal{L}}) = \sum_{i=0}^{L-1} (M_i + 1 - q')(M_{i+1} - q')$.

# 8   Conclusion

**Summary.**   Our paper provides a precise understanding of how the neural network structure constrains the Hessian range and the resulting rank deficiency. *In contrast to the number of parameters which are proportional to layer-widths squared, we obtain that rank is proportional to layer-width.* The proof strategy relies on bounding the rank of the two parts of the Hessian separately, i.e., the outer-product Hessian $\mathbf{H}_o$ and the functional Hessian $\mathbf{H}_f$, both of which are replete with the special $\mathbf{Z}$-like structure. The analysis also reveals several striking properties of the Hessian, such as surprisingly small overlap in the column spaces of $\mathbf{H}_o$ and $\mathbf{H}_f$, and independence of the layer-wise column blocks in $\mathbf{H}_f$. While our results were derived assuming linear activations, we demonstrate that, even with non-linearities, our formulas faithfully capture the numerical rank.

All in all, by providing fundamental insights into the rank of the Hessian map — which ultimately point to the sources and extent of redundancy in overparameterized neural networks, as a result of its compositional structure — our work paves the way to exciting avenues for future research.

**Discussion.**   In particular, our results merit discussion on some important aspects of deep learning:

*(i) Overparameterization:* Modern DNNs, with billions of parameters, are in stark contrast to the traditional statistical viewpoint of having # of parameters approximately equal to the # of samples. While several works have argued for measuring model complexity instead through weight norms [56], margins [57], compressibility [58], yet it remains hard to get an interpretable ballpark on the model complexity of neural networks. Since rank intuitively captures the notion of effective parameters, it could be a *possible alternative to benchmark overparameterization*, e.g. for double descent [38].

*(ii) Flatness:* A growing number of works [59–61] correlate the choice of regularizers, optimizers, or hyperparameters, with the additional flatness brought about by them at the minimum. However, the significant rank degeneracy of the Hessian, which we have provably established, also points to *another source of flatness* — that exists as a virtue of the compositional model structure —from the initialization itself. Thus, a prospective avenue of future work would be to compare different architectures based on this inherent kind of flatness.

*(iii) Generalization:* An interesting observation available from our work is that factors such as width, depth, enabling bias — commonly observed to improve generalization — *also result in decreasing the rank/# parameters ratio*, see Fig. S8b, S8c, S3. In a similar vein, recent work of [62] has provided a lower bound to the generalization error of statistical estimators in terms of the rank of the Fisher (which is intimately related to the Hessian) divided by # of parameters. Practically, one could use a further relaxation of rank as nuclear norm normalized by the spectral norm, in scenarios with spurious rank inflation. Overall, this suggests the relevance of studying the link between rank and generalization.

**Limitations.**   Although empirically our (linear) rank results faithfully capture the behaviour of Hessian rank with non-linearities, we still lack a rigorous theoretical characterization of the same (which will likely require incorporating additional distribution-specific assumptions). Besides, our analysis is limited to the case of fully-connected networks, and it would be interesting to extend this to other architectures such as convolutional networks, residual networks, etc. Lastly, it still needs to be shown theoretically that our upper-bound on the rank of the functional Hessian holds with equality, despite that this is what we observe in practice.

## Acknowledgments and Disclosure of Funding

We would like to thank Nicolò Ruggeri for reviewing a first draft of the paper. Further, we would like to thank the members of DA lab and Bernhard Schölkopf for useful comments. Sidak Pal Singh would also like to acknowledge the financial support from Max Planck ETH Center for Learning Systems.

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
