# Supplementary Material

## Table of Contents

# S1 Backpropagation in matrix-derivatives for the general case

In the general case, we can represent the gradient in analogy as to Eq. (1),

$$\frac{\partial F}{\partial \mathbf{W}^k} = \mathbf{J}^{L:k+1}\mathbf{\Lambda}^k \otimes \left[F^{k-1:1}(\mathbf{x})\right]^\top, \quad \text{where } F^{k-1:1} = F^{k-1} \circ \cdots \circ F^1 . \tag{7}$$

And, further $\mathbf{J}$ denotes the Jacobian map across the indexed layers, which is itself a composition of elementary Jacobians.

$$\mathbf{J}^k = \mathbf{\Lambda}^k \mathbf{W}^k, \quad \mathbf{\Lambda}^k = \mathrm{diag}(\dot{\sigma}^{(k)}), \quad \mathbf{J}^{k+1:L} = \mathbf{J}^{L:k+1\top} . \tag{8}$$

The Jacobian maps depend on functions of the input as each $\mathbf{\Lambda}^k$ depends on the pre-activation $\mathbf{W}^k \mathbf{x}_{k-1}$. By the usual chain rule (backpropagation) one has for a linear DNN, at a sample $(\mathbf{x}, \mathbf{y})$:

$$\frac{\partial \ell}{\partial \mathbf{W}^k} := \frac{\partial \ell_{\mathbf{x},\mathbf{y}}}{\partial \mathbf{W}^k} = \underbrace{\left[\mathbf{W}^{k+1:L}\boldsymbol{\delta}_{\mathbf{x},\mathbf{y}}\right]}_{\text{backward} \in \mathbb{R}^{M_k}} \cdot \underbrace{\left[\mathbf{W}^{k-1:1}\mathbf{x}\right]^\top}_{\text{forward} \in \mathbb{R}^{M_{k-1}}} = \mathbf{W}^{k+1:L}[\mathbf{W}^{L:1}\mathbf{x}\mathbf{x}^\top - \mathbf{y}\mathbf{x}^\top]\mathbf{W}^{1:k-1} . \tag{9}$$

The gradient with regard to $\mathbf{W}^k$ is first order in $\mathbf{W}^k$ and second order in the other matrices. In the general case, we get

$$\frac{\partial \ell}{\partial \mathbf{W}^k} = \left[\mathbf{J}^{k+1:L}\boldsymbol{\delta}_{\mathbf{x},\mathbf{y}}\right] \cdot \left[F^{k-1:1}(\mathbf{x})\right]^\top \tag{10}$$

Clearly, the partial forward maps are non-linear, whereas the backward maps are linearized at an argument determined by the current input.

## S1.1 Equivalence with Gauss-Newton decomposition

Remember, $\boldsymbol{\theta} \in \mathbb{R}^p$ denotes the (vectorized) parameters of the neural network map $F$, which then feeds into the loss $\ell$. Then, the Hessian of the composition of $\ell$ and $F$ with respect to $\boldsymbol{\theta}$ (computed over a sample $(\mathbf{x}, \mathbf{y})$, but we omit specifying it for brevity) is,

$$\nabla_{\boldsymbol{\theta}}^2(\ell \circ F) = \nabla_{\boldsymbol{\theta}} F^\top [\partial^2 \ell] \nabla_{\boldsymbol{\theta}} F + \sum_{c=1}^K [\partial \ell]_c \nabla_{\boldsymbol{\theta}}^2 F_c .$$

where $\partial\ell$ and $\partial^2\ell$ are respectively the gradient and Hessian of the loss $\ell$ with respect to the network function, $F$. Also, $\nabla_{\boldsymbol{\theta}} F \in \mathbb{R}^{K \times p}$ is the Jacobian map of the network function $F(\mathbf{x})$ with respect to the parameters $\boldsymbol{\theta}$. Let us employ the shorthand $\nabla_k F$ to denote the Jacobian of the network function with respect to the weight matrix $\mathbf{W}^k$ in the numerator-layout style as mentioned earlier. Similarly, let $\nabla_{kl}^2 F_c$ be the Hessian of $c$-th component of the network function with respect to weight matrices $\mathbf{W}^k, \mathbf{W}^l$. Hence, we obtain,

$$\partial^2 \ell = \mathbf{I}_K , \quad \nabla_k F = \mathbf{W}^{L:k+1} \otimes \mathbf{x}^\top \mathbf{W}^{1:k-1}, \partial^2 \ell = \mathbf{I}_K ,$$

$$\nabla_k F = \mathbf{W}^{L:k+1} \otimes \mathbf{x}^\top \mathbf{W}^{1:k-1} ,$$

$$\nabla_k F^\top \mathbf{I}_K \nabla_l F = \mathbf{W}^{k+1:L}\mathbf{W}^{L:l+1} \otimes \mathbf{W}^{k-1:1}\mathbf{x}\mathbf{x}^\top \mathbf{W}^{1:l-1} .$$

Now, the other term has a reduction (or contraction) with the components of the residual, i.e., $\partial\ell = \boldsymbol{\delta}_{\mathbf{x},\mathbf{y}}$. Importantly, we notice that it has a block-hollow structure since, $\nabla_{kk}^2 F_c = \mathbf{0}, \ \forall c \in [1 \cdots K]$. Thus all the diagonal blocks come from the first term in the Gauss-Newton decomposition. Now, comparing the above expressions with the Eqns. (3, 4, 5), it is evident that the two approaches yield the same structure of the Hessian.

As a side-remark, note that in Eqns. (3, 4, 5) we express the $kl$-th block as $\dfrac{\partial^2 \mathcal{L}}{\partial \mathbf{W}^l \partial \mathbf{W}^k}$ instead of $\dfrac{\partial^2 \mathcal{L}}{\partial \mathbf{W}^k \partial \mathbf{W}^l}$, only to ensure consistent shape as per matrix derivative convention (but corresponding entries are ofcourse equal).

## S2  Tools for the analysis

### S2.1  General notation

We employ the shorthand notation, $\mathbf{W}^{k:l}$, to refer to the matrix product chain $\mathbf{W}^k \cdots \mathbf{W}^l$, when $k > l$. When $k < l$, $\mathbf{W}^{k:l}$ will stand for the transposed product chain $\mathbf{W}^{k^\top} \cdots \mathbf{W}^{l^\top}$. In the edge case $k = l$, this will imply either $\mathbf{W}^k$ or $\mathbf{W}^{k^\top}$ depending on the context. Although, we will make the notation explicit on occasions where it might not be evident. Besides, we use the $\otimes$ to denote the Kronecker product of two matrices, $\mathrm{vec}_c$ and $\mathrm{vec}_r$ to denote the column-wise and row-wise vectorization of matrices respectively. $\mathbf{I}_k$ denotes the identity matrix of size $k$, while $\mathbb{1}_k$ denotes the all ones vector of length $k$, $\mathbf{0}_k$ denotes an all-zeros matrix of size $k$. The generalized inverse [41] [S1] of a matrix $\mathbf{A}$ is given by a matrix $\mathbf{A}^-$ which obeys $\mathbf{A}\mathbf{A}^-\mathbf{A} = \mathbf{A}$. The notation $\mathbf{A}^{\bullet i}$ or $\mathbf{A}_{\bullet i}$ denotes the $i$-th column of the matrix $\mathbf{A}$, while $\mathbf{A}^{i\bullet}$ or $\mathbf{A}_{i\bullet}$ refers to its $i$-th row. We place the column and row indices into the subscript or superscript depending on the context they are used.

### S2.2  Helper Lemmas

**Lemma 14.** *Let* $\mathbf{A} \in \mathbb{R}^{m \times n}$ *and* $\mathbf{B} \in \mathbb{R}^{p \times q}$. *Then the row-partitioned matrix* $\begin{bmatrix} \mathbf{I}_q \otimes \mathbf{A} \\ \mathbf{B} \otimes \mathbf{I}_n \end{bmatrix}$ *has the rank,*

$$\mathrm{rk}\begin{bmatrix} \mathbf{I}_q \otimes \mathbf{A} \\ \mathbf{B} \otimes \mathbf{I}_n \end{bmatrix} = q\,\mathrm{rk}(\mathbf{A}) + n\,\mathrm{rk}(\mathbf{B}) - \mathrm{rk}(\mathbf{A})\,\mathrm{rk}(\mathbf{B})$$

*Proof.* The proof relies on the following rank formula due to [63], and are based on using the generalized inverse of a matrix. Besides, the following proof closely follows Chuai and Tian [42].

$$\mathrm{rk}\begin{bmatrix} \mathbf{A} \\ \mathbf{C} \end{bmatrix} = \mathrm{rk}(\mathbf{A}) + \mathrm{rk}\left(\mathbf{C} - \mathbf{C}\mathbf{A}^-\mathbf{A}\right) = \mathrm{rk}(\mathbf{C}) + \mathrm{rk}\left(\mathbf{A} - \mathbf{A}\mathbf{C}^-\mathbf{C}\right)$$

Here, $\mathbf{C}^-$ denotes the weak (generalized) inverse of $\mathbf{C}$, i.e., any solution such that $\mathbf{C}\mathbf{C}^-\mathbf{C} = \mathbf{C}$. Then, we have that,

$$\begin{aligned}
\mathrm{rk}\begin{bmatrix} \mathbf{I}_q \otimes \mathbf{A} \\ \mathbf{B} \otimes \mathbf{I}_n \end{bmatrix} &= \mathrm{rk}(\mathbf{I}_q \otimes \mathbf{A}) + \mathrm{rk}\left((\mathbf{B} \otimes \mathbf{I}_n) - (\mathbf{B} \otimes \mathbf{I}_n)(\mathbf{I}_q \otimes \mathbf{A})^-(\mathbf{I}_q \otimes \mathbf{A})\right) \\
&= \mathrm{rk}(\mathbf{I}_q)\,\mathrm{rk}(\mathbf{A}) + \mathrm{rk}\left((\mathbf{B} \otimes \mathbf{I}_n) - (\mathbf{B} \otimes \mathbf{I}_n)(\mathbf{I}_q \otimes \mathbf{A}^-)(\mathbf{I}_q \otimes \mathbf{A})\right) \\
&= q\,\mathrm{rk}(\mathbf{A}) + \mathrm{rk}\left((\mathbf{B} \otimes \mathbf{I}_n) - \left(\mathbf{B} \otimes A^-A\right)\right) \\
&= q\,\mathrm{rk}(\mathbf{A}) + \mathrm{rk}\left(\mathbf{B} \otimes \left(\mathbf{I}_n - A^-A\right)\right) \\
&= q\,\mathrm{rk}(\mathbf{A}) + \mathrm{rk}(\mathbf{B})\,\mathrm{rk}\left(\mathbf{I}_n - A^-A\right) \\
&\overset{(a)}{=} q\,\mathrm{rk}(\mathbf{A}) + \mathrm{rk}(\mathbf{B})(n - \mathrm{rk}(\mathbf{A})) \\
&= q\,\mathrm{rk}(\mathbf{A}) + n\,\mathrm{rk}(\mathbf{B}) - \mathrm{rk}(\mathbf{A})\,\mathrm{rk}(\mathbf{B})
\end{aligned}$$

where, in (a) we have used that $\mathrm{rk}\left(\mathbf{I}_n - A^-A\right) = n - \mathrm{rk}(\mathbf{A})$, which follows from the fact that column space of the matrix $\mathbf{I}_n - A^-A$, satisfies $\mathcal{C}\left(\mathbf{I}_n - A^-A\right) \subset \mathcal{N}(A)$ (because the null space $\mathcal{N}(A)$ is the set of column vectors $\alpha$ for which $A\alpha = 0$ and since any $x = (\mathbf{I}_n - A^-A)\,y \implies Ax = 0$) which means $\mathrm{rk}\left(\mathbf{I}_n - A^-A\right) \leq \dim\mathcal{N}(A)$ and $\mathrm{rk}\left(\mathbf{I}_n - A^-A\right) \geq \mathrm{rk}\left(\mathbf{I}_n\right) - \mathrm{rk}\left(A^-A\right) = n - \mathrm{rk}(A) = \dim\mathcal{N}(A)$. $\square$

**Lemma 15.** *Let* $\mathbf{A}_1 \in \mathbb{R}^{m_1 \times n_1}$, $\mathbf{A}_2 \in \mathbb{R}^{m_2 \times n_2}$, $\mathbf{B} \in \mathbb{R}^{p \times n_1}$. *Then the column block matrix* $\begin{bmatrix} \mathbf{A}_1 \otimes \mathbf{A}_2 \\ \mathbf{B} \otimes \mathbf{I}_{n_2} \end{bmatrix}$ *has the rank,*

$$\mathrm{rk}\begin{bmatrix} \mathbf{A}_1 \otimes \mathbf{A}_2 \\ \mathbf{B} \otimes \mathbf{I}_{n_2} \end{bmatrix} = \mathrm{rk}(\mathbf{A}_2)\left(\mathrm{rk}\begin{bmatrix} \mathbf{A}_1 \\ \mathbf{B} \end{bmatrix} - \mathrm{rk}(\mathbf{B})\right) + n_2\,\mathrm{rk}(\mathbf{B})$$

---

[S1]This is like a general version of pseudoinverse which only satisfies the first Moore-Penrose condition.

*Proof.*

$$\mathrm{rk}\begin{bmatrix} \mathbf{A}_1 \otimes \mathbf{A}_2 \\ \mathbf{B} \otimes I_{n_2} \end{bmatrix} \overset{(a)}{=} \mathrm{rk}(\mathbf{B} \otimes \mathbf{I}_{n_2}) + \mathrm{rk}\left((\mathbf{A}_1 \otimes \mathbf{A}_2) - (\mathbf{A}_1 \otimes \mathbf{A}_2)(\mathbf{B} \otimes \mathbf{I}_{n_2})^{-}(\mathbf{B} \otimes \mathbf{I}_{n_2})\right)$$

$$= n_2\, \mathrm{rk}(\mathbf{B}) + \mathrm{rk}\left((\mathbf{A}_1 \otimes \mathbf{A}_2) - (\mathbf{A}_1 \otimes \mathbf{A}_2)(\mathbf{B}^{-}\mathbf{B} \otimes \mathbf{I}_{n_2})\right)$$

$$= n_2\, \mathrm{rk}(\mathbf{B}) + \mathrm{rk}\left((\mathbf{A}_1 - \mathbf{A}_1\mathbf{B}^{-}\mathbf{B}) \otimes \mathbf{A}_2)\right)$$

$$= n_2\, \mathrm{rk}(\mathbf{B}) + \mathrm{rk}(\mathbf{A}_1 - \mathbf{A}_1\mathbf{B}^{-}\mathbf{B})\, \mathrm{rk}(\mathbf{A}_2)$$

$$\overset{(b)}{=} n_2\, \mathrm{rk}(\mathbf{B}) + \left(\mathrm{rk}\begin{bmatrix} \mathbf{A}_1 \\ \mathbf{B} \end{bmatrix} - \mathrm{rk}(\mathbf{B})\right)\mathrm{rk}(\mathbf{A}_2)$$

$$= \mathrm{rk}(\mathbf{A}_2)\left(\mathrm{rk}\begin{bmatrix} \mathbf{A}_1 \\ \mathbf{B} \end{bmatrix} - \mathrm{rk}(\mathbf{B})\right) + n_2\, \mathrm{rk}(\mathbf{B})$$

Step (a) and (b) are due to the rank formula for the column block matrix [63].

$$\mathrm{rk}\begin{bmatrix} \mathbf{X} \\ \mathbf{Y} \end{bmatrix} = \mathrm{rk}(\mathbf{Y}) + \mathrm{rk}\left(\mathbf{X} - \mathbf{X}\mathbf{Y}^{-}\mathbf{Y}\right)$$

And, the other steps follow from the basic properties of Kronecker product.

$\square$

**Lemma 16.** *Let $\mathbf{A} \in \mathbb{R}^{p \times q}$, $\mathbf{B} \in \mathbb{R}^{m \times p}$, then we have that:*

$$\mathrm{rk}\begin{bmatrix} \mathbf{A} \\ \mathbf{B}\mathbf{A} \end{bmatrix} = \mathrm{rk}(\mathbf{A})$$

*Proof.* This follows simply from the definition of generalized inverse ($\mathbf{A}\mathbf{A}^{-}\mathbf{A} = \mathbf{A}$) and the rank formula for column-block matrix.

$$\mathrm{rk}\begin{bmatrix} \mathbf{A} \\ \mathbf{B}\mathbf{A} \end{bmatrix} = \mathrm{rk}(\mathbf{A}) + \mathrm{rk}\left((\mathbf{B}\mathbf{A}) - (\mathbf{B}\mathbf{A})\mathbf{A}^{-}\mathbf{A}\right)$$

$$= \mathrm{rk}(\mathbf{A}) + \mathrm{rk}(\mathbf{B}\mathbf{A} - \mathbf{B}\mathbf{A}) = \mathrm{rk}(\mathbf{A})$$

$\square$

**Corollary 17.** *Let $\mathbf{B} \in \mathbb{R}^{m \times p}$, then we get the elementary identity:*

$$\mathrm{rk}\begin{bmatrix} \mathbf{I}_p \\ \mathbf{B} \end{bmatrix} = \mathrm{rk}(\mathbf{I}_p) = p\,.$$

**Lemma 18.** *Given a matrix $\mathbf{M} := \mathbf{A}\mathbf{B}\mathbf{A}^{T}$, with $\mathbf{B} \succ 0$ symmetric, then $\mathrm{rk}(\mathbf{M}) = \mathrm{rk}(\mathbf{A})$.*

*Proof.* Since $\mathbf{B} \succ 0$, we can write $\mathbf{M} = (\mathbf{A}\mathbf{B}^{\frac{1}{2}})(\mathbf{A}\mathbf{B}^{\frac{1}{2}})^{\top}$, where $\mathbf{B}^{\frac{1}{2}}$ denotes the matrix square root of $\mathbf{B}$. This implies $\mathrm{rk}(\mathbf{M}) = \mathrm{rk}(\mathbf{A}\mathbf{B}^{\frac{1}{2}})$, since the null space of $\mathbf{X}^{\top}\mathbf{X}$ is the same as the null space of any arbitrary matrix $\mathbf{X}$, i.e., $\mathcal{N}(\mathbf{X}^{\top}\mathbf{X}) = \mathcal{N}(\mathbf{X})$, and additionally using $\mathrm{rk}(\mathbf{X}^{\top}) = \mathrm{rk}(\mathbf{X})$. Next, as $\mathbf{B} \succ 0$, we have $\mathbf{B}^{\frac{1}{2}} \succ 0$, which further implies $\mathbf{B}^{\frac{1}{2}}$ is full rank. Hence $\mathrm{rk}(\mathbf{A}\mathbf{B}^{\frac{1}{2}}) = \mathrm{rk}(\mathbf{A})$, which at last gives, $\mathrm{rk}(\mathbf{M}) = \mathrm{rk}(\mathbf{A})$. $\square$

## S2.3 {Left, Right, Pseudo}- Inverses

For a matrix $\mathbf{A}$ with full column rank, the left inverse is defined to be a matrix $\mathbf{A}^{-L}$ such that $\mathbf{A}^{-L}\mathbf{A} = \mathbf{I}$. Likewise, when the matrix $\mathbf{A}$ has full rank, we can define a right inverse which is a matrix $\mathbf{A}^{-R}$ such that $\mathbf{A}\mathbf{A}^{-R} = \mathbf{I}$. The left and right inverses need not be unique. But often a nice or convenient choice for the left inverse is $\mathbf{A}^{-L} = (\mathbf{A}^{\top}\mathbf{A})^{-1}\mathbf{A}^{\top}$, while that for the right inverse is $\mathbf{A}^{-R} = \mathbf{A}^{\top}(\mathbf{A}\mathbf{A}^{\top})^{-1}$.

The (Moore-Penrose) pseudoinverse $\mathbf{A}^\dagger$ of a matrix $\mathbf{A} \in \mathbb{R}^{m \times n}$ is a unique matrix that satisfies the following properties:

$$\mathbf{A}\mathbf{A}^\dagger\mathbf{A} = \mathbf{A}$$
$$\mathbf{A}^\dagger\mathbf{A}\mathbf{A}^\dagger = \mathbf{A}^\dagger$$
$$\left(\mathbf{A}\mathbf{A}^\dagger\right)^\top = \mathbf{A}\mathbf{A}^\dagger$$
$$\left(\mathbf{A}^\dagger\mathbf{A}\right)^\top = \mathbf{A}^\dagger\mathbf{A}$$

When the matrix has full column rank or full row rank, then the pseudoinverse agrees with the particular choice of left and right inverse we mentioned above. In such scenarios of full column or row rank, when we want to refer to this choice of left or right inverse, we will simply denote them by $\mathbf{A}^\dagger$.

## S2.4 Block row and column operations

In our proofs, we make use of row and column operations, taken altogether on blocks of matrices rather than just individual rows or columns. So, here we clarify what we actually mean by such block row and column operations, and how their usage does not affect the rank of matrix to which these are applied. Essentially, we will look at the corresponding "elementary matrices" that get formed and argue that they multiplying with them does not change rank.

Let us consider that we have with us the following matrix, $\mathbf{P}$, with its row blocks labeled as $R_1, \cdots, R_L$.

$$\mathbf{P} = \begin{array}{c} R_1 \\ \vdots \\ R_i \\ \vdots \\ R_L \end{array}\begin{pmatrix} \mathbf{\Omega}^1 \otimes \mathbf{N}^1 \\ \vdots \\ \mathbf{\Omega}^i \otimes \mathbf{N}^i \\ \vdots \\ \mathbf{\Omega}^L \otimes \mathbf{N}^L \end{pmatrix}. \tag{11}$$

### S2.4.1 Factoring-out block operations

Assume that the matrix $\mathbf{\Omega}^i$ has full column rank and then $\mathbf{\Omega}^i \otimes \mathbf{I}$ also has full column rank and is left-invertible. That means we can write the following decomposition $\mathbf{P} = \mathbf{P}_1\mathbf{P}_2$.

$$\mathbf{P} = \underbrace{\begin{pmatrix} \mathbf{I} & \cdots & \mathbf{0} & \cdots & \mathbf{0} \\ \vdots & \ddots & \vdots & \ddots & \vdots \\ \mathbf{0} & \cdots & \mathbf{\Omega}^i \otimes \mathbf{I} & \cdots & \mathbf{0} \\ \vdots & \ddots & \vdots & \ddots & \vdots \\ \mathbf{0} & \cdots & \mathbf{0} & \cdots & \mathbf{I} \end{pmatrix}}_{\mathbf{P}_1}\underbrace{\begin{pmatrix} \mathbf{\Omega}^1 \otimes \mathbf{N}^1 \\ \vdots \\ \mathbf{I} \otimes \mathbf{N}^i \\ \vdots \\ \mathbf{\Omega}^L \otimes \mathbf{N}^L \end{pmatrix}}_{\mathbf{P}_2}. \tag{12}$$

Since even $\mathbf{P}_1$ is left-invertible (identity matrix in all diagonals except for $\mathbf{\Omega}^i \otimes \mathbf{I}$ but which is left-invertible), we have that,

$$\mathrm{rk}(\mathbf{P}) = \mathrm{rk}(\mathbf{P}_1\mathbf{P}_2) = \mathrm{rk}(\mathbf{P}_2).$$

This is what we actually mean when applying the row operation,

$$R_i \leftarrow (\mathbf{\Omega}^i \otimes \mathbf{I})^\dagger R_i.$$

Subsequently, we start working with the matrix $\mathbf{P}_2$, although we may not explicitly update the name of the matrix.

**Remark.** Likewise, we could have instead assumed $N^i$ to be left invertible, and factored out a corresponding $\mathbf{P}_1$ matrix with $\mathbf{I} \otimes N^i$ as one of its diagonal blocks. Going further, in a similar manner, we can define the column operation analogue of this by factoring out from the right a matrix which is right-invertible, and will thus preserve rank.

### S2.4.2 Deletion block operations

The block operations in this section are nothing but the analogue of the usual row operations, except they are carried out at the level of blocks. Say that we have done the above factoring-out operation in Eq. (14). Now, we are given that $N^1 = N^i$, i.e, we have the matrix:

$$\mathbf{P} = \begin{pmatrix} \mathbf{\Omega}^1 \otimes \mathbf{N}^1 \\ \vdots \\ \mathbf{I} \otimes \mathbf{N}^1 \\ \vdots \\ \mathbf{\Omega}^L \otimes \mathbf{N}^L \end{pmatrix}. \tag{13}$$

Now, consider we multiply from the left with the matrix $\mathbf{Q}$ to yield $\mathbf{P}'$:

$$\mathbf{P}' = \underbrace{\begin{pmatrix} \mathbf{I} & \cdots & -\mathbf{\Omega}^1 \otimes \mathbf{I} & \cdots & \mathbf{0} \\ \vdots & \ddots & \vdots & \ddots & \vdots \\ \mathbf{0} & \cdots & \mathbf{I} & \cdots & \mathbf{0} \\ \vdots & \ddots & \vdots & \ddots & \vdots \\ \mathbf{0} & \cdots & \mathbf{0} & \cdots & \mathbf{I} \end{pmatrix}}_{\mathbf{Q}} \begin{pmatrix} \mathbf{\Omega}^1 \otimes \mathbf{N}^1 \\ \vdots \\ \mathbf{I} \otimes \mathbf{N}^1 \\ \vdots \\ \mathbf{\Omega}^L \otimes \mathbf{N}^L \end{pmatrix} = \begin{pmatrix} \mathbf{0} \\ \vdots \\ \mathbf{I} \otimes \mathbf{N}^1 \\ \vdots \\ \mathbf{\Omega}^L \otimes \mathbf{N}^L \end{pmatrix}. \tag{14}$$

Since $\mathbf{Q}$ is a upper-triangular matrix with ones on the diagonal, it is invertible. As a result, rank of $\mathbf{P}$ does not change when multiplied by $\mathbf{Q}$ from the left.

This is what we actually mean when applying the row operation,

$$R_1 \leftarrow R_1 - (\mathbf{\Omega}^1 \otimes \mathbf{I}) R_i.$$

Subsequently, we start working with the matrix $\mathbf{P}'$, although we may not explicitly update the name of the matrix.

**Remark.** Notice, we could have done a similar thing on other side of Kronecker factors. Going further, in a similar manner, we can define the column operation analogue of this by multiplying on the right such a matrix which is invertible, and which will thus preserve rank.

**Aliter.** One can perhaps intuit these block operations from the point of view of inclusion of subspaces, but here we are being a bit pedantic.

## S2.5    Matrix derivatives

Let us start by discussing some simple facts on vectorization. Consider a matrix $\mathbf{A} \in \mathbb{R}^{m \times n}$, and recall that $\mathrm{vec}_r$ and $\mathrm{vec}_c$ denote rowwise and columnwise vectorization respectively. Then firstly we have the following simple relation between them:

$$\mathrm{vec}_c(\mathbf{A}) = \mathrm{vec}_r(\mathbf{A}^\top). \tag{15}$$

Now, we give the proof of the commonly-used identity, $\mathrm{vec}_c(\mathbf{A}\mathbf{X}\mathbf{B}) = \left(\mathbf{B}^\top \otimes \mathbf{A}\right) \mathrm{vec}_c(\mathbf{X})$, where $\mathbf{A} \in \mathbb{R}^{m \times n}$, $\mathbf{X} \in \mathbb{R}^{n \times p}$, $\mathbf{B} \in \mathbb{R}^{p \times q}$. For more details on this, refer to [39, 64].

The main idea is to write $\mathbf{X}$ in the form of canonical basis vectors $\mathbf{e}_i$, i.e., $\mathbf{X} = \sum_{i=1}^{p} \mathbf{X}_{\bullet i}\, \mathbf{e}_i^\top$, where $\mathbf{X}_{\bullet i}$ denotes the $i$-th column of $\mathbf{X}$.

$$
\begin{aligned}
\mathrm{vec}_c\left(\mathbf{A} \sum_{i=1}^{p} \mathbf{X}_{\bullet i}\, \mathbf{e}_i^\top \mathbf{B}\right) &= \mathrm{vec}_c\left(\sum_{i=1}^{p} (\mathbf{A}\mathbf{X}_{\bullet i})\left(\mathbf{B}^\top \mathbf{e}_i\right)\right) \\
&\overset{(a)}{=} \sum_{i=1}^{p} \left(\mathbf{B}^\top \mathbf{e}_i\right) \otimes (\mathbf{A}\mathbf{X}_{\bullet i}) \\
&\overset{(b)}{=} \left(\mathbf{B}^\top \otimes \mathbf{A}\right) \sum_{i=1}^{p} \mathbf{e}_i \otimes \mathbf{X}_{\bullet i} = \left(\mathbf{B}^\top \otimes \mathbf{A}\right) \mathrm{vec}_c(\mathbf{X}) \quad \square
\end{aligned}
$$

In step (a), we have used that for two vectors $\mathbf{a}, \mathbf{b}$, the following basic fact $\mathrm{vec}_c(\mathbf{a}\mathbf{b}^\top) = \mathbf{b} \otimes \mathbf{a}$ holds. And, in (b) we have employed the mixed-product property of Kronecker products.

Since, we utilize row-wise vectorization in our paper, let use find the equivalent relation in terms of that:

$$\mathrm{vec}_r\left(\mathbf{A}\mathbf{X}\mathbf{B}\right) \overset{\text{Eq. (15)}}{=} \mathrm{vec}_c\left(\mathbf{B}^\top \mathbf{X}^\top \mathbf{A}^\top\right) = \left(\mathbf{A} \otimes \mathbf{B}^\top\right) \mathrm{vec}_c\left(\mathbf{X}^\top\right) \overset{\text{Eq. (15)}}{=} \left(\mathbf{A} \otimes \mathbf{B}^\top\right) \mathrm{vec}_r(\mathbf{X}). \tag{16}$$

Recall, we use the numerator (Jacobian) layout to express matrix-by-matrix derivatives, i.e.,

$$\frac{\partial \mathbf{Y}}{\partial \mathbf{X}} := \frac{\partial\, \mathrm{vec}_r(\mathbf{Y})}{\partial\, \mathrm{vec}_r(\mathbf{X})^\top}.$$

Thus, when $\mathbf{Y} = \mathbf{A}\mathbf{X}\mathbf{B}$, we use the above property along the first identification theorem of vector calculus [39] as mentioned below:

**Theorem 19.** *(first identification theorem):*

$$\mathrm{d}f = A(\mathbf{x})\,\mathrm{d}\mathbf{x} \iff \frac{\partial f(\mathbf{x})}{\partial \mathbf{x}^\top} = A(\mathbf{x}),$$

*where,* $\mathrm{d}$ *denotes the differential.*

Finally, this yields that:

$$\frac{\partial \mathbf{A}\mathbf{X}\mathbf{B}}{\partial \mathbf{X}} = \mathbf{A} \otimes \mathbf{B}^\top.$$

**Remark.**    If we were using the column vectorization, we would have instead obtained $\mathbf{B}^\top \otimes \mathbf{A}$.

## S2.6    Rank of weight matrices at initialization

Here we study the rank of random matrices to understand how the weight matrices of a neural network at initialization influence the rank.

**Lemma 20.** *Consider a random matrix* $\mathbf{W} \in \mathbb{R}^{m \times n}$ *for* $m, n \in \mathbb{N}$ *where each entry is sampled i.i.d. w.r.t. to some continuous (i.e. not discrete) probability distribution* $p$, *i.e.* $W_{ij} \sim p$. *Then it holds that*

$$\mathrm{rank}(\mathbf{W}) = \min(m, n)\ \ a.s.$$

*Proof.* Assume w.l.o.g. that $n \leq m$ (otherwise consider the transposed matrix) and enumerate the columns of $\mathbf{W}$ as $\mathbf{W}_1, \ldots, \mathbf{W}_n \in \mathbb{R}^m$. We need to show the linear independence of $\{\mathbf{W}_1, \ldots, \mathbf{W}_n\}$ over $\mathbb{R}^m$. Let us show this inductively. First $\mathbf{W}_1 \neq \mathbf{0}$ with probability 1 since $p$ is continuous. Consider now $\mathbf{W}_2$. Conditioned on the previously sampled column, $\mathbf{W}_1$, due to the continuous nature of the distribution, the probability of $\mathbf{W}_2$ being contained in the span of $\mathbf{W}_1$ is zero:

$$\mathbb{P}\left(\mathbf{W}_2 \in \text{span}(\mathbf{W}_1)\right) = \int_{\mathbb{R}} \underbrace{\mathbb{P}\left(\mathbf{W}_2 \in \text{span}(\mathbf{W}_1) | \mathbf{W}_1\right)}_{=0} p(\mathbf{W}_1) d\mathbf{W}_1 = 0$$

Finally, consider $\mathbf{W}_m$ and condition on the previously sampled vectors $\mathbf{W}_1, \ldots, \mathbf{W}_{m-1}$. By the induction hypotheses, they span an $m-1$-dimensional space. Again, due to the independence of $\mathbf{W}_m$ from the previous vectors and the fact that an $m-1$-dimensional subspace has Lebesgue measure 0 in $\mathbb{R}^m$, it holds in a similar fashion that

$$\mathbb{P}\left(\mathbf{W}_m \in \text{span}(\mathbf{W}_1, \ldots, \mathbf{W}_{m-1})\right) = 0 \tag{17}$$

and the matrix hence has full rank. $\qquad\square$

It turns out that we can apply a similar argument for the case of the product of two random matrices:

**Lemma 21.** *Consider random matrices $\mathbf{V} \in \mathbb{R}^{m \times n}$ and $\mathbf{W} \in \mathbb{R}^{n \times k}$, both drawn with i.i.d. entries according to some continuous probability distribution $p$. Define $\mathbf{Z} = VW \in \mathbb{R}^{m \times k}$. Then it holds that*

$$\text{rank}(\mathbf{Z}) = \min(m, n, k)$$

*Proof.* First, notice that by Lemma 20, both matrices have full rank, i.e. $\text{rank}(\mathbf{V}) = \min(m, n)$ and $\text{rank}(\mathbf{W}) = \min(n, k)$. By standard linear algebra results (not involving the fact that we have random matrices), we get that for $n \leq m$, $\text{rank}(\mathbf{Z}) = \text{rank}(\mathbf{W}) = \min(m, k)$ and for $n \leq k$, $\text{rank}(\mathbf{Z}) = \text{rank}(\mathbf{V}) = \min(m, n)$.

Thus it remains to show the case where $n \geq k, m$, i.e. the contracting dimension is the biggest. Assume w.l.o.g. that $k \leq m$ (otherwise study the transposed matrix). The columns of $\mathbf{Z}$ are given by $\mathbf{z}_i = \mathbf{V}\mathbf{W}_i$ for $i = 1, \ldots, k$. It thus suffices to show the linear independence of $\{\mathbf{z}_1, \ldots, \mathbf{z}_k\}$.

Notice that $\text{rank}(\mathbf{V}) = m$ from the assumptions, thus $\{\mathbf{V}\mathbf{x} : \mathbf{x} \in \mathbb{R}^n\}$ is a $m$-dimensional subspace of $\mathbb{R}^n$. We will apply a similar argument as in Lemma 20. Consider $\mathbf{z}_1 = \mathbf{V}\mathbf{W}_1$. Due to the independence and the fact that $\text{rank}(\mathbf{V}) = m > 0$, $\mathbf{z}_1 \neq 0$ a.s. Assume that $\mathbf{z}_1, \ldots, \mathbf{z}_{k-1}$ are linearly independent, they thus span a $k-1$ dimensional space.

Conditioned on $\mathbf{z}_1, , \ldots, \mathbf{z}_{k-1}$, $\mathbf{z}_k = \mathbf{V}\mathbf{W}_k$ is a random vector in $\text{Im}(\mathbf{V})$ (think of $\mathbf{V}$ as a fixed linear map since we are conditioning on it). Since $\text{span}(\mathbf{z}_1, \ldots, \mathbf{z}_{m-1})$ forms a $k-1$ dimensional subspace of $\text{Im}(\mathbf{V})$, which has dimension $m$, the subspace again has Lebesgue measure zero and we conclude that

$$\mathbb{P}\left(\mathbf{z}_k \in \text{span}(\mathbf{z}_1, \ldots, \mathbf{z}_{k-1})\right) = \int_{\mathbb{R}} \underbrace{\mathbb{P}\left(\mathbf{z}_k \in \text{span}(\mathbf{z}_1, \ldots, \mathbf{z}_{k-1}) | \mathbf{z}_1, \ldots, \mathbf{z}_{k-1}\right)}_{=0} p(\mathbf{z}_k) d\mathbf{z}_k$$

$\qquad\square$

We can now easily use this result for an arbitrary sized matrix product:

**Corollary 22.** *Consider random matrices $\mathbf{W}^i \in \mathbb{R}^{m_i \times m_{i+1}}$ for $i = 1, \ldots, n$ where each entry is initialized i.i.d. w.r.t. a continuous distribution $p$. Define the product matrix $\mathbf{W} = \mathbf{W}^1 \ldots \mathbf{W}^n$. Then it holds that*

$$\text{rank}(\mathbf{W}) = \min(m_1, \ldots, m_n)$$

*Proof.* Apply Lemma 21 recursively, i.e. for $\mathbf{W} = \mathbf{W}^1 \mathbf{W}^{2:n}$, then for $\mathbf{W}^{2:n} = \mathbf{W}^2 \mathbf{W}^{3:n}$ up until $\mathbf{W}^n$. $\qquad\square$

## S3 Rank of the outer-product term

We begin by discussing the proof of Proposition 2. Then, we briefly discuss the example of two-layer networks to motivate the proof, and after that we discuss the proof of the Theorem 3. Subsequently, we present the proof of Corollary 4.

### S3.1 Proof of Proposition 2

**Proposition 2.** *For a deep linear network,* $\mathbf{H}_o = \mathbf{A}_o \mathbf{B}_o \mathbf{A}_o^\top$, *where* $\mathbf{B}_o = \mathbf{I}_K \otimes \mathbf{\Sigma_{xx}} \in \mathbb{R}^{Kd \times Kd}$,

$$\text{and } \mathbf{A}_o^\top = \begin{pmatrix} \mathbf{W}^{L:2} \otimes \mathbf{I}_d & \cdots & \mathbf{W}^{L:l+1} \otimes \mathbf{W}^{1:l-1} & \cdots & \mathbf{I}_K \otimes \mathbf{W}^{1:L-1} \end{pmatrix} \in \mathbb{R}^{Kd \times p},$$

*Proof.* From Eq. (3) we can notice that any block, say $kl$-th, of $\mathbf{H}_o$ can be re-written as,

$$\mathbf{H}_o^{kl} = \begin{pmatrix} \mathbf{W}^{k+1:L} \otimes \mathbf{W}^{k-1:1} \end{pmatrix} \begin{pmatrix} \mathbf{I}_K \otimes \mathbf{\Sigma_{xx}} \end{pmatrix} \begin{pmatrix} \mathbf{W}^{L:l+1} \otimes \mathbf{W}^{1:l-1} \end{pmatrix},$$

where, we have used the mixed-product property of Kronecker products i.e., $\mathbf{AB} \otimes \mathbf{CD} = (\mathbf{A} \otimes \mathbf{C})(\mathbf{B} \otimes \mathbf{D})$. Now, it is clear from looking at the terms which are on the left and right of $\mathbf{I}_K \otimes \mathbf{\Sigma_{xx}}$, that we get the required decomposition.

$\square$

**Remark R1.** *As mentioned in the preliminaries, we consider, without loss of generality, that when the (uncentered) input covariance* $\mathbf{\Sigma_{xx}}$ *has rank* $r < d$, *then we take it to be*

$$\mathbf{\Sigma_{xx}} = \begin{pmatrix} (\mathbf{\Sigma_{xx}})_{r \times r} & \mathbf{0}_{r \times (d-r)} \\ \mathbf{0}_{(d-r) \times r} & \mathbf{0}_{(d-r) \times (d-r)} \end{pmatrix}.$$

*which is always possible by pre-processing the input (although this is not needed in practice). Thus, in such a scenario we can equivalently work with* $\mathbf{\Sigma_{xx}} := (\mathbf{\Sigma_{xx}})_{r \times r}$ *and just consider the first* $r$ *columns of* $\mathbf{W}^1$ *(whose shape will then be* $\mathbb{R}^{M_1 \times r}$*). Otherwise, if* $r = d$, *then we just continue with* $\mathbf{\Sigma_{xx}}$ *and* $\mathbf{W}^1 \in \mathbb{R}^{M_1 \times d}$ *as usual. To simplify our discussion ahead, we will always write* $r$ *in place of* $d$, *however the meaning of it should be clear from this remark.*

Thus, in our presentation of the proof of Theorem 3, we will make a similar adaptation to Proposition 2, and so

$$\mathbf{A}_o^\top = \begin{pmatrix} \mathbf{W}^{L:2} \otimes \mathbf{I}_r & \cdots & \mathbf{W}^{L:l+1} \otimes \mathbf{W}^{1:l-1} & \cdots & \mathbf{I}_K \otimes \mathbf{W}^{1:L-1} \end{pmatrix} \in \mathbb{R}^{Kr \times p'},$$

where $p' = p + K(r - d)$.

### S3.2 Example for two-layer networks

Let us first illustrate Theorem 3 via the example of $L = 2$, i.e., we have a 2-layer network $F_{\boldsymbol{\theta}}(\mathbf{x}) = \mathbf{W}^2 \mathbf{W}^1 \mathbf{x}$, with weight matrices $\mathbf{W}^2 \in \mathbb{R}^{K \times M_1}$ and $\mathbf{W}^1 \in \mathbb{R}^{M_1 \times r}$. Applying Proposition (2), we obtain $\mathbf{A}_o$ with the familiar structure:

$$\mathbf{A}_o = \begin{pmatrix} \mathbf{W}^{2^\top} \otimes \mathbf{I}_r \\ \mathbf{I}_K \otimes \mathbf{W}^1 \end{pmatrix}$$

Applying Lemma 1 on $\mathbf{A}_o$ thus yields: $\text{rk}(\mathbf{A}_o) = r\,\text{rk}(\mathbf{W}^{2^\top}) + K\,\text{rk}(\mathbf{W}^1) - \text{rk}(\mathbf{W}^{2^\top})\,\text{rk}(\mathbf{W}^1)$. If we assume that the hidden layer is the bottleneck, i.e., $q := \min(r, M_1, K) = M_1$, then we get $\text{rk}(\mathbf{A}_o) = r\,M_1 + K\,M_1 - M_1^2$, keeping in mind the Assumption A1.

While here the special $\mathbf{Z}$-like structure is apparent at the outset, the general case of $L$-layers is more involved and requires additional work to reduce to this structure, as illustrated in our proof ahead.

### S3.3 Proof of Theorem 3

Let us restate the Theorem 3 from the main text,

**Theorem 3.** *Consider the matrix $\mathbf{A}_o$ mentioned in Proposition 2. Under the assumption A1,*

$$\mathrm{rk}(\mathbf{A}_o) = r\,\mathrm{rk}(\mathbf{W}^{2:L}) + K\,\mathrm{rk}(\mathbf{W}^{L-1:1}) - \mathrm{rk}(\mathbf{W}^{2:L})\,\mathrm{rk}(\mathbf{W}^{L-1:1}) = q\,(r + K - q).$$

*Proof.* The proof is divided into two parts: (1) Bottleneck case and (2) Non-bottleneck case. For more details about the block-row operations that we employ here, please refer to the Section S2.4.

**Part 1: Bottleneck case.** We assume without loss of generality that the layer $\ell - 1$ has the minimum layer width out of all hidden layers, besides what is known that $M_{\ell-1} < \min(d, K)$. We will therefore have that $\mathbf{W}^\ell \in \mathbb{R}^{M_\ell \times M_{\ell-1}}$ will have full column rank and will be left-invertible. Due to random initialization of weight matrices (see Section S2.6) we will also have $\mathbf{W}^{k:\ell}$, $k \geq \ell$ to also have a left inverse.

Then let us write the block matrix $\mathbf{A}_o$ in the column manner and label the row blocks corresponding to layer $\ell$ as $R_\ell$:

$$\mathbf{A}_o = \begin{array}{c} R_1 \\ \vdots \\ R_{\ell-1} \\ R_\ell \\ R_{\ell+1} \\ \vdots \\ R_L \end{array} \begin{pmatrix} \mathbf{W}^{2:L} \otimes \mathbf{I}_r \\ \vdots \\ \mathbf{W}^{\ell:L} \otimes \mathbf{W}^{\ell-2:1} \\ \mathbf{W}^{\ell+1:L} \otimes \mathbf{W}^{\ell-1:1} \\ \mathbf{W}^{\ell+2:L} \otimes \mathbf{W}^{\ell:1} \\ \vdots \\ \mathbf{I}_K \otimes \mathbf{W}^{L-1:1} \end{pmatrix}. \tag{18}$$

Consider the following (block) row operations:

$$R_k \leftarrow \left(\boldsymbol{I}_{M_k} \otimes \mathbf{W}^{k-1:\ell}\right)^\dagger R_k, \quad \forall k \in \ell+1, \cdots, L$$

These row-operations are valid as the pre-factor has full-column rank, and are rank preserving, as discussed in Section S2.4. In this way, we have

$$\mathbf{A}_o = \begin{array}{c} R_1 \\ \vdots \\ R_{\ell-2} \\ R_{\ell-1} \\ R_\ell \\ R_{\ell+1} \\ \vdots \\ R_L \end{array} \begin{pmatrix} \mathbf{W}^{2:L} \otimes \mathbf{I}_r \\ \vdots \\ \mathbf{W}^{\ell-1:L} \otimes \mathbf{W}^{\ell-3:1} \\ \mathbf{W}^{\ell:L} \otimes \mathbf{W}^{\ell-2:1} \\ \mathbf{W}^{\ell+1:L} \otimes \mathbf{W}^{\ell-1:1} \\ \mathbf{W}^{\ell+2:L} \otimes \mathbf{W}^{\ell-1:1} \\ \vdots \\ \mathbf{I}_K \otimes \mathbf{W}^{\ell-1:1} \end{pmatrix}. \tag{19}$$

Similarly, $\mathbf{W}^{\ell-1^\top} \in \mathbb{R}^{M_{\ell-2} \times M_{\ell-1}}$ as well as $\mathbf{W}^{k:\ell-1}$ for $k \leq \ell-1$ is also full-column rank and thus left-invertible. Then apply the following row operations,

$$R_k \leftarrow \left(\mathbf{W}^{k+1:\ell-1} \otimes I_{M_{k-1}}\right)^\dagger R_k, \quad \forall k \in \ell-2, \cdots, 1$$

And we get,

$$
\mathbf{A}_o =
\begin{array}{c}
R_1 \\ \vdots \\ R_{\ell-2} \\ R_{\ell-1} \\ R_\ell \\ R_{\ell+1} \\ \vdots \\ R_L
\end{array}
\left(
\begin{array}{c}
\mathbf{W}^{\ell:L} \otimes \mathbf{I}_r \\
\vdots \\
\mathbf{W}^{\ell:L} \otimes \mathbf{W}^{\ell-3:1} \\
\mathbf{W}^{\ell:L} \otimes \mathbf{W}^{\ell-2:1} \\
\hline
\mathbf{W}^{\ell+1:L} \otimes \mathbf{W}^{\ell-1:1} \\
\mathbf{W}^{\ell+2:L} \otimes \mathbf{W}^{\ell-1:1} \\
\vdots \\
\mathbf{I}_K \otimes \mathbf{W}^{\ell-1:1}
\end{array}
\right). \tag{20}
$$

Now using $R_1$, we can apply the deletion block operations to remove $\{R_2, \cdots, R_{\ell-1}\}$ as the left term in the Kronecker is identical. Next, via $R_L$ we can apply the deletion block operations to get rid of $\{R_\ell, \cdots, R_L\}$ as now the right term in the Kronecker is identical. We are left with:

$$
\mathbf{A}_o =
\begin{array}{c} R_1 \\ R_L \end{array}
\left(
\begin{array}{c}
\mathbf{W}^{\ell:L} \otimes \mathbf{I}_r \\
\mathbf{I}_K \otimes \mathbf{W}^{\ell-1:1}
\end{array}
\right). \tag{21}
$$

Finally, we can apply Lemma 1 to obtain that, when $\mathbf{W}^\ell$ has full column rank or $M_{\ell-1}$ is the minimum layer-width (i.e., the bottleneck dimension):

$$
\begin{aligned}
\mathrm{rk}(\mathbf{A}_o) &= r \, \mathrm{rk}(\mathbf{W}^{\ell:L}) + K \, \mathrm{rk}(\mathbf{W}^{\ell-1:1}) - \mathrm{rk}(\mathbf{W}^{\ell:L}) \, \mathrm{rk}(\mathbf{W}^{\ell-1:1}) \\
&= r M_{\ell-1} + K M_{\ell-1} - M_{\ell-1}^2 \\
&= q \, (r + K - q),
\end{aligned}
$$

where, in the last step, we have used the definition of $q := \min(r, M_1, \cdots, M_{L-1}, K) = M_{\ell-1}$ which gives rise to the equivalent expression.

**Part 2: Non-bottleneck case.** This part is very similar and we will see that it uses just one set of row operations (like for the layers $> \ell$ and $< \ell$). In the non-bottleneck case, there are further two possibilities:

**When $K$ is the minimum:** This means that $\mathbf{W}^{L^\top}$ has full column-rank as $M_L = K$ is the minimum of all layer widths and input dimensionality. We start from the same $\mathbf{A}_o$ matrix as in Eq. (18). Now consider the following factoring-out operations:

$$
R_k \leftarrow \left(\mathbf{W}^{k+1:L} \otimes \boldsymbol{I}_{M_{k-1}}\right)^\dagger R_k \quad \forall k \in 1, \cdots, L-1
$$

These row-operations are valid (i.e., rank-preserving) as the pre-factor of (block-)row $R_k$ has full column rank, as discussed in Section S2.4. This results in,

$$
\mathbf{A}_o =
\begin{array}{c}
R_1 \\ \vdots \\ R_{\ell-1} \\ R_\ell \\ R_{\ell+1} \\ \vdots \\ R_L
\end{array}
\left(
\begin{array}{c}
\mathbf{I}_K \otimes \mathbf{I}_r \\
\vdots \\
\mathbf{I}_K \otimes \mathbf{W}^{\ell-2:1} \\
\mathbf{I}_K \otimes \mathbf{W}^{\ell-1:1} \\
\mathbf{I}_K \otimes \mathbf{W}^{\ell:1} \\
\vdots \\
\mathbf{I}_K \otimes \mathbf{W}^{L:1}
\end{array}
\right).
\tag{22}
$$

This is then followed by,

$$
R_k \leftarrow R_k - \left(\boldsymbol{I}_K \otimes \mathbf{W}^{k-1:1}\right) R_1, \quad \forall\, k \in 2, \cdots, L\,.
$$

This results in,

$$
\mathbf{A}_o = R_1 \left(\mathbf{I}_K \otimes \mathbf{I}_r\right).
\tag{23}
$$

**When $r$ is the minimum:** This means that $\mathbf{W}^1$ has full column-rank as $M_0 = r$ is the minimum of all layer widths, input and output dimensionality. We start from the same $\mathbf{A}_o$ matrix as in Eq. (18). Now consider the following factoring-out operations:

$$
R_k \leftarrow \left(\boldsymbol{I}_{M_k} \otimes \mathbf{W}^{k-1:1}\right)^{\dagger} R_k \quad \forall\, k \in 2, \cdots, L
$$

As before, these row-operations are valid (i.e., rank-preserving) as the pre-factor of (block-)row $R_k$ has full column rank, as discussed in Section S2.4. This results in,

$$
\mathbf{A}_o =
\begin{array}{c}
R_1 \\ \vdots \\ R_{\ell-1} \\ R_\ell \\ R_{\ell+1} \\ \vdots \\ R_L
\end{array}
\left(
\begin{array}{c}
\mathbf{W}^{2:L} \otimes \mathbf{I}_r \\
\vdots \\
\mathbf{W}^{\ell:L} \otimes \mathbf{I}_r \\
\mathbf{W}^{\ell+1:L} \otimes \mathbf{I}_r \\
\mathbf{W}^{\ell+2:L} \otimes \mathbf{I}_r \\
\vdots \\
\mathbf{I}_K \otimes \mathbf{I}_r
\end{array}
\right).
\tag{24}
$$

This is then followed by,

$$
R_k \leftarrow R_k - \left(\mathbf{W}^{k+1:L} \otimes \boldsymbol{I}_r\right) R_L, \quad \forall\, k \in 1, \cdots, L-1\,.
$$

This results in,

$$
\mathbf{A}_o = R_L \left(\mathbf{I}_K \otimes \mathbf{I}_r\right).
\tag{25}
$$

**Resulting rank:** Thus, for either scenario of $K$ or $r$ being the minimum we get,

$$
\mathrm{rk}(\mathbf{A}_o) = \mathrm{rk}(\boldsymbol{I}_K \otimes \mathbf{I}_r) = Kr.
$$

**Final note.** It is easy to check that for both the first and second case, we can summarize the obtained rank in the form of the following equality:

$$\mathrm{rk}(\mathbf{A}_o) = r\,\mathrm{rk}(\mathbf{W}^{L:2}) + K\,\mathrm{rk}(\mathbf{W}^{L-1:1}) - \mathrm{rk}(\mathbf{W}^{L:2})\,\mathrm{rk}(\mathbf{W}^{L-1:1}) = q\,(r + K - q).$$

$\square$

### S3.4 Proof of Corollary 4

Let us first recall the Corollary,

**Corollary 4.** *Under the setup of Theorem 3, the rank of* $\mathbf{H}_o$ *is given by*

$$\mathrm{rk}(\mathbf{H}_o) = q\,(r + K - q).$$

*Proof.* It is quite evident that we can use the rank of $\mathbf{A}_o$ to bound the rank of $\mathbf{H}_o$ due to the decomposition from Proposition 2. However, as mentioned in the main text, we can show an equality using the Lemma 18 since $\mathbf{H}_o$ is also of the form $\mathbf{A}\mathbf{B}\mathbf{A}^\top$ with $\mathbf{B} = (\mathbf{I}_K \otimes \boldsymbol{\Sigma}_{\mathbf{xx}}) \succ \mathbf{0}$.

$$\mathrm{rk}(\mathbf{H}_o) \overset{\text{Prop. 2}}{=} \mathrm{rk}(\mathbf{A}_o\mathbf{B}\mathbf{A}_o^\top) \overset{\text{Lemma 18}}{=} \mathrm{rk}(\mathbf{A}_o)$$
$$\overset{\text{Thm. 3}}{=} q\,(r + K - q).$$

$\square$

## S4 Rank of the functional Hessian term

### S4.1 Proof of Theorem 5

Similar to the outer-product case, here as well the proof is divided into two cases. However, there is a subtle difference in that the input-residual covariance matrix ($\boldsymbol{\Omega} = \mathbf{E}\left[\boldsymbol{\delta}_{\mathbf{x},\mathbf{y}}\mathbf{x}^\top\right]$) can also dictate the rank, besides the weight matrix with minimum dimension. Especially since during training, as the residual approaches zero, the matrix $\boldsymbol{\Omega} \to 0$. To abstract this, we will consider a new definition $\widehat{q}$ of $q$ which includes $s = \mathrm{rk}(\boldsymbol{\Omega})$ in the minimum, and thus we get

$$\widehat{q} = \min(r, M_1, \cdots, M_{L-1}, K, s).$$

Notice the additional $s$ as the last argument in the minimum. Further, since $\boldsymbol{\Omega} \in \mathbb{R}^{K \times r}$, we have that $s \leq \min(K, r)$.

Given these considerations, we split our analysis to the case where $\widehat{q} = s$ (referred to as the non-bottleneck case) and where $\widehat{q} = \min(M_1, \cdots, M_{L-1})$ (referred to as the bottleneck case). In the bottleneck case, we can also just substitute $q$ in place of $\widehat{q}$, but we will retain $\widehat{q}$ for uniformity. Note, if $\widehat{q} = r$ or $\widehat{q} = K$, then these cases are already subsumed by $\widehat{q} = s$ case, since $s \leq \min(K, r)$.

Further, in each of the two cases, we will analyse the rank of the block-columns of the functional Hessian formed with respect to a transposed weight matrix, i.e., $\widehat{\mathbf{H}}_f^{\bullet \ell}$. This makes it easier to deal with underlying structure of the actual block-columns $\mathbf{H}_f^{\bullet \ell}$, and does not affect the rank as it is invariant to row or column permutations. Finally, before we proceed into the details of the two parts, let us recollect the theorem statement,

**Theorem 5.** *For a deep linear network, the rank of l-th column-block, $\widehat{\mathbf{H}}_f^{\bullet l}$, of the matrix $\widehat{\mathbf{H}}_f$, under the assumption A1 is given as $\mathrm{rk}(\widehat{\mathbf{H}}_f^{\bullet l}) = \widehat{q}\,M_{l-1} + \widehat{q}\,M_l - \widehat{q}^2$, for $l \in [2, \cdots, L-1]$. When $l = 1$, we have $\mathrm{rk}(\widehat{\mathbf{H}}_f^{\bullet 1}) = \widehat{q}\,M_1 + \widehat{q}\,s - \widehat{q}^2$. And, when $l = L$, we have $\mathrm{rk}(\widehat{\mathbf{H}}_f^{\bullet L}) = \widehat{q}\,M_{L-1} + \widehat{q}\,s - \widehat{q}^2$. Here, $\widehat{q} := \min(r, M_1, \cdots, M_{L-1}, K, s) = \min(q, s)$ and $s := \mathrm{rk}(\boldsymbol{\Omega}) = \mathrm{rk}(\mathbf{E}\left[\boldsymbol{\delta}_{\mathbf{x},\mathbf{y}}\mathbf{x}^\top\right])$.*

#### S4.1.1 Non-bottleneck case ($\widehat{q} = s$)

In this case, the Theorem boils down to showing the following:

1. The rank of the $\ell^{\text{th}}$ column ($\ell \in [2, \cdots, L-1]$) of the functional Hessian, i.e. $\widehat{\mathbf{H}}_f^{\bullet \ell}$, is given by:

$$\mathrm{rk}(\widehat{\mathbf{H}}_f^{\bullet \ell}) = s\,M_{\ell-1} + s\,M_\ell - s^2$$

2. The rank of the first column of the functional Hessian, i.e. $\widehat{\mathbf{H}}_f^{\bullet 1}$, is given by:

$$\mathrm{rk}(\widehat{\mathbf{H}}_f^{\bullet 1}) = s\,M_1$$

3. The rank of the last column of the functional Hessian, i.e. $\widehat{\mathbf{H}}_f^{\bullet L}$, is given by:

$$\mathrm{rk}(\widehat{\mathbf{H}}_f^{\bullet L}) = s\,M_{L-1}$$

*Proof.* In this case, $\boldsymbol{\Omega}$, which shows up in every single block will dictate the rank. As before, for more details about the factoring-out and deletion block-row operations that we employ here, please refer to the Section S2.4. Let us now look at each of the parts:

**Part 1.** Let us first consider the case of the inner-columns, with $\ell > 1$ and $\ell < L$. Now, the expression for $\widehat{\mathbf{H}}_f^{\bullet \ell}$ is given by,

$$
\widehat{\mathbf{H}}_f^{\bullet\ell} =
\begin{array}{c}
\text{vec}_r(\mathbf{W}^1) \\
\vdots \\
\text{vec}_r(\mathbf{W}^j) \\
\vdots \\
\text{vec}_r(\mathbf{W}^{\ell-1}) \\
\text{vec}_r(\mathbf{W}^\ell) \\
\text{vec}_r(\mathbf{W}^{\ell+1}) \\
\vdots \\
\text{vec}_r(\mathbf{W}^k) \\
\vdots \\
\text{vec}_r(\mathbf{W}^L)
\end{array}
\overset{\text{vec}_r(\mathbf{W}^{\ell^\top})}{
\begin{pmatrix}
\mathbf{W}^{2:\ell-1} \otimes \boldsymbol{\Omega}^\top \mathbf{W}^{L:\ell+1} \\
\vdots \\
\mathbf{W}^{j+1:\ell-1} \otimes \mathbf{W}^{j-1:1} \boldsymbol{\Omega}^\top \mathbf{W}^{L:\ell+1} \\
\vdots \\
\mathbf{I}_{M_{\ell-1}} \otimes \mathbf{W}^{\ell-2:1} \boldsymbol{\Omega}^\top \mathbf{W}^{L:\ell+1} \\
\hline
\mathbf{0} \\
\hline
\mathbf{W}^{\ell+2:L} \boldsymbol{\Omega} \mathbf{W}^{1:\ell-1} \otimes \mathbf{I}_{M_\ell} \\
\vdots \\
\mathbf{W}^{k+1:L} \boldsymbol{\Omega} \mathbf{W}^{1:\ell-1} \otimes \mathbf{W}^{k-1:\ell+1} \\
\vdots \\
\boldsymbol{\Omega} \mathbf{W}^{1:\ell-1} \otimes \mathbf{W}^{L-1:\ell+1}
\end{pmatrix}}
$$

Now, since $\text{rk}(\boldsymbol{\Omega}) = s$, we can express it as $\boldsymbol{\Omega} = \mathbf{CD}$, where $\mathbf{C} \in \mathbb{R}^{K \times s}$ and $\mathbf{D} \in \mathbb{R}^{s \times d}$. Then consider the following factoring-out block operations:

$$
R_j \leftarrow \left( \mathbf{I}_{m_j} \otimes \mathbf{W}^{j-1:1} \mathbf{D}^\top \right)^\dagger R_j, \quad \forall j \in 1, \cdots, \ell-1
$$

where, $^\dagger$ denotes the left-inverse. When $j = 1$, $\mathbf{W}^{j-1:1} := \mathbf{I}_r$. These row-operations are valid (i.e., rank-preserving) as $\mathbf{W}^{j-1}\mathbf{D}^\top$ has full column rank and is left-invertible. Similarly, consider the following factoring-out block operations:

$$
R_k \leftarrow \left( \mathbf{W}^{k+1:L} \mathbf{C} \otimes \mathbf{I}_{M_{k-1}} \right)^\dagger R_k, \quad \forall k \in \ell+1, \cdots, L
$$

Note, when $k = L$, $\mathbf{W}^{k+1:L} := \mathbf{I}_K$. These row-operations are also valid (i.e., rank-preserving) as $\mathbf{W}^{k+1:L}\mathbf{C}$ has full column rank and is left-invertible. Then we can express the $\widehat{\mathbf{H}}_f^{\bullet\ell}$ as follows:

$$
\widehat{\mathbf{H}}_f^{\bullet\ell} =
\begin{array}{c}
\text{vec}_r(\mathbf{W}^1) \\
\vdots \\
\text{vec}_r(\mathbf{W}^j) \\
\vdots \\
\text{vec}_r(\mathbf{W}^{\ell-1}) \\
\text{vec}_r(\mathbf{W}^\ell) \\
\text{vec}_r(\mathbf{W}^{\ell+1}) \\
\vdots \\
\text{vec}_r(\mathbf{W}^k) \\
\vdots \\
\text{vec}_r(\mathbf{W}^L)
\end{array}
\overset{\text{vec}_r(\mathbf{W}^{\ell^\top})}{
\begin{pmatrix}
\mathbf{W}^{2:\ell-1} \otimes \mathbf{C}^\top \mathbf{W}^{L:\ell+1} \\
\vdots \\
\mathbf{W}^{j+1:\ell-1} \otimes \mathbf{C}^\top \mathbf{W}^{L:\ell+1} \\
\vdots \\
\mathbf{I}_{M_{\ell-1}} \otimes \mathbf{C}^\top \mathbf{W}^{L:\ell+1} \\
\hline
\mathbf{0} \\
\hline
\mathbf{D} \mathbf{W}^{1:\ell-1} \otimes \mathbf{I}_{M_\ell} \\
\vdots \\
\mathbf{D} \mathbf{W}^{1:\ell-1} \otimes \mathbf{W}^{k-1:\ell+1} \\
\vdots \\
\mathbf{D} \mathbf{W}^{1:\ell-1} \otimes \mathbf{W}^{L-1:\ell+1}
\end{pmatrix}}
$$

Now, it is clear that we can use row block $\ell - 1$ to eliminate all row blocks prior to it and row block $\ell + 1$ to eliminate all row blocks after it by the following deletion block operations:

$$R_j \leftarrow R_j - \left(\mathbf{W}^{j+1:\ell-1} \otimes \mathbf{I}_q\right) R_{\ell-1}, \quad \forall j \in 1, \cdots, \ell - 1$$

$$R_k \leftarrow R_k - \left(\mathbf{I}_q \otimes \mathbf{W}^{k-1:\ell+1}\right) R_{\ell+1}, \quad \forall k \in \ell + 1, \cdots, L$$

Hence, we have that

$$\mathrm{rk}(\widehat{\mathbf{H}}_f^{\bullet\ell}) = \mathrm{rk}\left(\begin{array}{c} \mathbf{I}_{M_{\ell-1}} \otimes \mathbf{C}^\top \mathbf{W}^{L:\ell+1} \\ \mathbf{D}\mathbf{W}^{1:\ell-1} \otimes \mathbf{I}_{M_\ell} \end{array}\right) = s\,M_{\ell-1} + s\,M_\ell - s^2$$

where, we used the Lemma from [42] in the last step.

**Part 2.** The procedure for this part will follow Part 1 procedure for blocks **after** the zero block.

$$\mathrm{rk}(\widehat{\mathbf{H}}_f^{\bullet\ell}) = \min(s\,M_1, d\,M_1) = s\,M_1$$

**Part 3.** The procedure for this part will follow Part 1 procedure for blocks **before** the zero block.

$$\mathrm{rk}(\widehat{\mathbf{H}}_f^{\bullet\ell}) = \min(s\,M_{L-1}, K\,M_{L-1}) = s\,M_{L-1}$$

$\square$

### S4.1.2 Bottleneck case ($\widehat{q} \neq s$)

Here, we need to prove the following:

1. The rank of the $\ell^{\text{th}}$ column ($\ell \in [2, \cdots, L-1]$) of the functional Hessian, i.e. $\widehat{\mathbf{H}}_f^{\bullet\ell}$, is given by:

$$\mathrm{rk}(\widehat{\mathbf{H}}_f^{\bullet\ell}) = \widehat{q}\,M_{\ell-1} + \widehat{q}\,M_\ell - \widehat{q}^2$$

2. The rank of the first column of the functional Hessian, i.e. $\widehat{\mathbf{H}}_f^{\bullet 1}$, is given by:

$$\mathrm{rk}(\widehat{\mathbf{H}}_f^{\bullet 1}) = \widehat{q}\,M_1 + \widehat{q}\,s - \widehat{q}^2$$

3. The rank of the last column of the functional Hessian, i.e. $\widehat{\mathbf{H}}_f^{\bullet L}$, is given by:

$$\mathrm{rk}(\widehat{\mathbf{H}}_f^{\bullet L}) = \widehat{q}\,M_{L-1} + \widehat{q}\,s - \widehat{q}^2$$

*Proof.* Let us assume that $M_k$ is the bottleneck width, so it will "dictate" the rank now. Like in previous parts, for more details about the block-row operations that we employ here, please refer to the Section S2.4. Let us now look at each of the parts:

**Part 1.** Let us first consider the case of the inner-columns, with $\ell > 1$ and $\ell < L$. Further, let us take $k > \ell$, and the procedure in the other scenario of $k < \ell$ is similar.

We have that $\widehat{\mathbf{H}}_f^{\bullet\ell}$ is as follows:

$$\widehat{\mathbf{H}}_f^{\bullet\ell} = \begin{array}{c} \\ \text{vec}_r(\mathbf{W}^1) \\ \vdots \\ \text{vec}_r(\mathbf{W}^j) \\ \vdots \\ \text{vec}_r(\mathbf{W}^{\ell-1}) \\ \text{vec}_r(\mathbf{W}^{\ell}) \\ \text{vec}_r(\mathbf{W}^{\ell+1}) \\ \vdots \\ \text{vec}_r(\mathbf{W}^{k}) \\ \vdots \\ \text{vec}_r(\mathbf{W}^{L}) \end{array} \overset{\text{vec}_r(\mathbf{W}^{\ell\top})}{\left(\begin{array}{c} \mathbf{W}^{2:\ell-1} \otimes \mathbf{\Omega}^\top \mathbf{W}^{L:\ell+1} \\ \vdots \\ \mathbf{W}^{j+1:\ell-1} \otimes \mathbf{W}^{j-1:1} \mathbf{\Omega}^\top \mathbf{W}^{L:\ell+1} \\ \vdots \\ \mathbf{I}_{M_{\ell-1}} \otimes \mathbf{W}^{\ell-2:1} \mathbf{\Omega}^\top \mathbf{W}^{L:\ell+1} \\ \hline \mathbf{0} \\ \hline \mathbf{W}^{\ell+2:L} \mathbf{\Omega} \mathbf{W}^{1:\ell-1} \otimes \mathbf{I}_{M_\ell} \\ \vdots \\ \mathbf{W}^{k+1:L} \mathbf{\Omega} \mathbf{W}^{1:\ell-1} \otimes \mathbf{W}^{k-1:\ell+1} \\ \vdots \\ \mathbf{\Omega} \mathbf{W}^{1:\ell-1} \otimes \mathbf{W}^{L-1:\ell+1} \end{array}\right)}$$

Notice, we can write $\mathbf{W}^{i+1:L} = \mathbf{W}^{i+1:k} \mathbf{W}^{k+1:L}$, $\forall i \leq k-1$. The matrix $\mathbf{W}^{i+1:k}$ is left-invertible as $M_k$ is the bottleneck width. Then, we consider the following factoring-out block operations (see Section S2.4) for the below zero part:

$$R_i \leftarrow \left(\mathbf{W}^{i+1:k} \otimes \mathbf{I}_{M_{i-1}}\right)^\dagger R_i, \quad \forall i \in \ell+1, \cdots, k-1$$

where, $^\dagger$ denotes the left-inverse. Now, for a layer $i$ between $\{k+2, \cdots, L\}$, we notice that $\mathbf{W}^{i-1:\ell+1} = \mathbf{W}^{i-1:k+1} \mathbf{W}^{k:\ell+1}$ with $\mathbf{W}^{i-1:k+1}$ being left-invertible. Hence consider the factoring-out operations,

$$R_i \leftarrow \left(\mathbf{I}_{M_i} \otimes \mathbf{W}^{i-1:k+1}\right)^\dagger R_i, \quad \forall i \in k+2, \cdots, L$$

Now, for the row-blocks in the part above zero, we have that $\mathbf{W}^{j-1:1} \mathbf{\Omega}^\top \mathbf{W}^{L:\ell+1} = \mathbf{W}^{j-1:1} \mathbf{\Omega}^\top \mathbf{W}^{L:k+1} \mathbf{W}^{k:\ell+1}$, $\forall j \leq \ell-1$. Notice, that $\mathbf{W}^{j-1:1} \mathbf{\Omega}^\top \mathbf{W}^{L:k+1}$ is left invertible due to the bottleneck $M_k$. Hence, consider the following block-row operations:

$$R_j \leftarrow \left(\mathbf{I}_{m_j} \otimes \mathbf{W}^{j-1:1} \mathbf{\Omega}^\top \mathbf{W}^{L:k+1}\right)^\dagger R_j, \quad \forall j \in 1, \cdots, \ell-1$$

Here, it does not matter to us what the actual value of $s$ is, since we know that $M_k$ is the minimum width which guarantees that the above expression containing $\mathbf{\Omega}^\top$ is left invertible.

Overall, we can thus express the $\widehat{\mathbf{H}}_f^{\bullet\ell}$ as follows:

$$
\widehat{\mathbf{H}}_f^{\bullet\ell} = 
\begin{array}{c}
\\
\text{vec}_r(\mathbf{W}^1) \\
\vdots \\
\text{vec}_r(\mathbf{W}^j) \\
\vdots \\
\text{vec}_r(\mathbf{W}^{\ell-1}) \\
\\
\text{vec}_r(\mathbf{W}^\ell) \\
\text{vec}_r(\mathbf{W}^{\ell+1}) \\
\vdots \\
\text{vec}_r(\mathbf{W}^k) \\
\vdots \\
\text{vec}_r(\mathbf{W}^L)
\end{array}
\begin{pmatrix}
\mathbf{W}^{2:\ell-1} \otimes \mathbf{W}^{k:\ell+1} \\
\vdots \\
\mathbf{W}^{j+1:\ell-1} \otimes \mathbf{W}^{k:\ell+1} \\
\vdots \\
\mathbf{I}_{M_{\ell-1}} \otimes \mathbf{W}^{k:\ell+1} \\
\hline
\mathbf{0} \\
\hline
\mathbf{W}^{k+1:L}\boldsymbol{\Omega}\mathbf{W}^{1:\ell-1} \otimes \mathbf{I}_{M_\ell} \\
\vdots \\
\mathbf{W}^{k+1:L}\boldsymbol{\Omega}\mathbf{W}^{1:\ell-1} \otimes \mathbf{W}^{k-1:\ell+1} \\
\vdots \\
\boldsymbol{\Omega}\mathbf{W}^{1:\ell-1} \otimes \mathbf{W}^{k:\ell+1}
\end{pmatrix}
\overset{\text{vec}_r(\mathbf{W}^{\ell\top})}{}
$$

Now, it is clear that we can use row $\ell - 1$ to eliminate all rows prior to it as well as the rows from $k + 1$ to $L$. While the row $\ell + 1$ can be used to eliminate all rows from $\ell + 2$ until $k$. Hence, we have that,

$$
\text{rk}(\widehat{\mathbf{H}}_f^{\bullet\ell}) = \text{rk}\begin{pmatrix} \mathbf{I}_{M_{\ell-1}} \otimes \mathbf{W}^{k:\ell+1} \\ \mathbf{W}^{k+1:L}\boldsymbol{\Omega}\mathbf{W}^{1:\ell-1} \otimes \mathbf{I}_{M_\ell} \end{pmatrix} = \widehat{q} M_{\ell-1} + \widehat{q} M_\ell - \widehat{q}^2
$$

where, we used the Lemma from [42] in the last step.

**Part 2.** Now we deal with the column block corresponding to first layer, which is:

$$
\widehat{\mathbf{H}}_f^1 = 
\begin{array}{c}
\\
\text{vec}_r(\mathbf{W}^1) \\
\text{vec}_r(\mathbf{W}^2) \\
\vdots \\
\text{vec}_r(\mathbf{W}^{k-1}) \\
\text{vec}_r(\mathbf{W}^k) \\
\text{vec}_r(\mathbf{W}^{k+1}) \\
\vdots \\
\text{vec}_r(\mathbf{W}^L)
\end{array}
\begin{pmatrix}
\mathbf{0} \\
\mathbf{W}^{3:L}\boldsymbol{\Omega} \otimes \mathbf{I}_{M_1} \\
\vdots \\
\mathbf{W}^{k:L}\boldsymbol{\Omega} \otimes \mathbf{W}^{k-2:2} \\
\mathbf{W}^{k+1:L}\boldsymbol{\Omega} \otimes \mathbf{W}^{k-1:2} \\
\mathbf{W}^{k+2:L}\boldsymbol{\Omega} \otimes \mathbf{W}^{k:2} \\
\vdots \\
\boldsymbol{\Omega} \otimes \mathbf{W}^{L-1:2}
\end{pmatrix}
\overset{\text{vec}_r(\mathbf{W}^{1\top})}{}
$$

Basically, we have to follow the same procedure for row blocks before $k$ and for the ones after $k$ as done in the part 1. In other words, for the row blocks prior to $k$, we can write the $\mathbf{W}^{i+1:L} = \mathbf{W}^{i+1:k}\mathbf{W}^{k+1:L}$. Since $\mathbf{W}^{i+1:k}$ is left-invertible due to the bottleneck, we consider the following factoring-out operations:

$$
R_i \leftarrow \left(\mathbf{W}^{i+1:k} \otimes \mathbf{I}_{M_{i-1}}\right)^\dagger R_i, \quad \forall i \in 2, \cdots, k-1
$$

Now, for a layer $i$ between $\{k+2, \cdots, L\}$, we notice that $\mathbf{W}^{i-1:2} = \mathbf{W}^{i-1:k+1}\mathbf{W}^{k:2}$ with $\mathbf{W}^{i-1:k+1}$ being left-invertible. Hence the factoring-out operations will be,

$$R_i \leftarrow \left( \mathbf{I}_{M_i} \otimes \mathbf{W}^{i-1:k+1} \right)^{\dagger} R_i, \quad \forall i \in k+2, \cdots, L$$

This resulting $\widehat{\mathbf{H}}_f^1$ is as follows:

$$\widehat{\mathbf{H}}_f^1 = \begin{array}{c} \\ \text{vec}_r(\mathbf{W}^1) \\ \text{vec}_r(\mathbf{W}^2) \\ \vdots \\ \text{vec}_r(\mathbf{W}^{k-1}) \\ \text{vec}_r(\mathbf{W}^k) \\ \text{vec}_r(\mathbf{W}^{k+1}) \\ \vdots \\ \text{vec}_r(\mathbf{W}^L) \end{array} \begin{pmatrix} \text{vec}_r(\mathbf{W}^{1\top}) \\ \mathbf{0} \\ \mathbf{W}^{k+1:L}\mathbf{\Omega} \otimes \mathbf{I}_{M_1} \\ \vdots \\ \mathbf{W}^{k+1:L}\mathbf{\Omega} \otimes \mathbf{W}^{k-2:2} \\ \mathbf{W}^{k+1:L}\mathbf{\Omega} \otimes \mathbf{W}^{k-1:2} \\ \mathbf{W}^{k+2:L}\mathbf{\Omega} \otimes \mathbf{W}^{k:2} \\ \vdots \\ \mathbf{\Omega} \otimes \mathbf{W}^{k:2} \end{pmatrix}$$

Now, we can use row 2 to eliminate rows 3 until $k$, and similarly we can use row $L$ to eliminate rows $k+1$ to $L-1$. Thus, we have that:

$$\text{rk}(\widehat{\mathbf{H}}_f^1) = \text{rk} \begin{pmatrix} \mathbf{W}^{k+1:L}\mathbf{\Omega} \otimes \mathbf{I}_{M_1} \\ \mathbf{\Omega} \otimes \mathbf{W}^{k:2} \end{pmatrix} = \widehat{q}\,M_1 + \widehat{q}\,s - \widehat{q}^2$$

Different from the previous analysis, in the above matrix we did not have a "naked" Identity matrix on the left, so we could not directly use the Lemma 1. But, we used its generalized version contained in Lemma 15 along with the Lemma 16 to obtain the rank in the final step.

**Part 3.** Finally, we have the last column:

$$\widehat{\mathbf{H}}_f^{\bullet L} = \begin{array}{c} \\ \text{vec}_r(\mathbf{W}^1) \\ \vdots \\ \text{vec}_r(\mathbf{W}^{k-1}) \\ \text{vec}_r(\mathbf{W}^k) \\ \text{vec}_r(\mathbf{W}^{k+1}) \\ \vdots \\ \text{vec}_r(\mathbf{W}^{L-1}) \\ \text{vec}_r(\mathbf{W}^L) \end{array} \begin{pmatrix} \text{vec}_r(\mathbf{W}^{L\top}) \\ \mathbf{W}^{2:L-1} \otimes \mathbf{\Omega}^{\top} \\ \vdots \\ \mathbf{W}^{k:L-1} \otimes \mathbf{W}^{k-2:1}\mathbf{\Omega}^{\top} \\ \mathbf{W}^{k+1:L-1} \otimes \mathbf{W}^{k-1:1}\mathbf{\Omega}^{\top} \\ \mathbf{W}^{k+2:L-1} \otimes \mathbf{W}^{k:1}\mathbf{\Omega}^{\top} \\ \vdots \\ \mathbf{I}_{M_{L-1}} \otimes \mathbf{W}^{L-2:1}\mathbf{\Omega}^{\top} \\ \mathbf{0} \end{pmatrix}$$

We can follow a similar strategy as carried out in Part 2 to get,

$$\text{rk}(\widehat{\mathbf{H}}_f^{\bullet L}) = \text{rk} \begin{pmatrix} \mathbf{W}^{k+1:L-1} \otimes \mathbf{\Omega}^{\top} \\ \mathbf{I}_{M_{L-1}} \otimes \mathbf{W}^{k:1}\mathbf{\Omega}^{\top} \end{pmatrix} = \widehat{q}\,M_{L-1} + \widehat{q}\,s - \widehat{q}^2$$

$\square$

## S4.2 Proof of Corollary 6

Let us remember the Corollary from the main text,

**Corollary 6.** *Under the setup of Theorem 5, the rank of $\mathbf{H}_f$ can be upper bounded as,*

$$\mathrm{rk}(\mathbf{H}_f) \leq 2\,\widehat{q}\,M + 2\,\widehat{q}\,s - L\,\widehat{q}^2\,, \quad \textit{where} \quad M = \sum_{\ell=1}^{L-1} M_\ell\,.$$

*Proof.* By using the above Theorem 5 and applying the fact that $\mathrm{rk}([\mathbf{A}\ \mathbf{B}]) \leq \mathrm{rk}(\mathbf{A}) + \mathrm{rk}(\mathbf{B})$ on the column-blocks of $\widehat{\mathbf{H}}_f$, we get the desired upper bound on the rank of entire functional Hessian. ,

$$\mathrm{rk}(\mathbf{H}_f) = \mathrm{rk}(\widehat{\mathbf{H}}_f) \ \leq\ \sum_{\ell=1}^{L} \mathrm{rk}(\widehat{\mathbf{H}}_f^{\bullet\ell})$$

$$= \widehat{q}\,M_1 + \widehat{q}\,s - \widehat{q}^2 + \sum_{\ell=2}^{L-1}\left(\widehat{q}\,M_{\ell-1} + \widehat{q}\,M_\ell - \widehat{q}^2\right) + \widehat{q}\,M_{L-1} + \widehat{q}\,s - \widehat{q}^2$$

$$= 2\,\widehat{q}\left(\sum_{\ell=1}^{L-1} M_\ell\right) + 2\,\widehat{q}\,s - L\,\widehat{q}^2\ =\ 2\,\widehat{q}\,M + 2\,\widehat{q}\,s - L\,\widehat{q}^2$$

$\square$

## S5 Evolution of Rank during training

Here we prove Lemma 6, stating that the rank of the individual weights remains invariant under gradient flow dynamics. For sake of readability, let us restate Lemma 8:

**Lemma 8.** *For a deep linear network, consider the gradient flow dynamics* $\dot{\mathbf{W}}_t^l = -\eta \nabla_{\mathbf{W}^l} \mathcal{L}_S(\boldsymbol{\theta})\big|_{\boldsymbol{\theta}=\boldsymbol{\theta}_t}$. *Assume: (a) Centered classes:* $\frac{1}{N}\sum_{i:y_{ic}=1}^N \mathbf{x}_i = \mathbf{0}, \quad \forall c \in [1,\ldots,K]$. *(b) Balancedness at initialization:* $\mathbf{W}_0^{k+1\top}\mathbf{W}_0^{k+1} = \mathbf{W}_0^k$ $Wm_0^{k\top}$. *(c) Square weight-matrices:* $\mathbf{W}^l \in \mathbb{R}^{M\times M}, \forall l$ *and* $K = d = M$. *Then for all layers* $l$, $\mathrm{rk}(\mathbf{W}_t^l) = \mathrm{rk}(\mathbf{W}_0^l), \quad \forall t < \infty$.

To prove Lemma 8, we first need some helper lemmas. We start with a well-known result:

**Lemma 23.** *Under assumption b) in Lemma 8 and gradient flow dynamics, it holds that*

$$\left(\mathbf{W}_t^{k+1}\right)^T \mathbf{W}_t^{k+1} = \mathbf{W}_t^k \left(\mathbf{W}_t^k\right)^T$$

*Proof.* We refer to [65] (Theorem 1) for a proof. □

This essentially guarantees that balancedness is preserved throughout training if we guarantee it at initialization. Let us assume that the weights $\mathbf{W}^l$ are full rank, which for the squared matrix case is equivalent to $\det\left(\mathbf{W}^l\right) \neq 0$. The Jacobi formula allows us to extend the dynamics of $\mathbf{W}^l$ to the determinant:

**Lemma 24.** *Given a dynamic matrix* $t \mapsto \mathbf{A}(t)$, *the determinant follows:*

$$\frac{d}{dt}\det\left(\mathbf{A}(t)\right) = \mathrm{tr}\left(\mathrm{adj}\left(\mathbf{A}(t)\right)\frac{d\mathbf{A}(t)}{dt}\right)$$

*where* $\mathrm{adj}\left(\mathbf{A}(t)\right)$ *is the adjugate matrix satisfying*

$$\mathbf{A}(t)\,\mathrm{adj}\left(\mathbf{A}(t)\right) = \mathrm{adj}\left(\mathbf{A}(t)\right)\mathbf{A}(t) = \det(\mathbf{A}(t))\boldsymbol{I}$$

*Proof.* For a proof of this standard result we refer to Magnus and Neudecker [66] for instance. □

We are now ready to proof the main claim:

**Proof of Lemma 8:** Let us first simplify the right-hand side of the gradient flow equation using the assumptions, before applying Lemma 24 :

$$\nabla_{\mathbf{W}^l}\mathcal{L}(\mathbf{W})\big|_{\mathbf{W}=-\eta\mathbf{W}_t} = -\eta\mathbf{W}_t^{l+1:L}\boldsymbol{\Omega}\mathbf{W}_t^{1:l-1}$$
$$= -\eta\mathbf{W}_t^{l+1:L}\mathbf{W}_t^{L:l+1}\mathbf{W}_t^l\mathbf{W}_t^{l-1:1}\boldsymbol{\Sigma}\mathbf{W}_t^{1:l-1}$$
$$\overset{L1}{=} -\eta\left(\mathbf{W}_t^l\mathbf{W}_t^{lT}\right)^{L-l}\mathbf{W}_t^l\mathbf{W}_t^{l-1:1}\boldsymbol{\Sigma}\mathbf{W}_t^{1:l-1}$$

Applying Lemma 24, gives

$$\frac{d}{dt}\det\left(\mathbf{W}_t^l\right) = \mathrm{tr}\left(\mathrm{adj}\left(\mathbf{W}_t^l\right)\dot{\mathbf{W}}_t^l\right)$$
$$= -\eta\,\mathrm{tr}\left(\mathrm{adj}\left(\mathbf{W}_t^l\right)\left(\mathbf{W}_t^l\mathbf{W}_t^{lT}\right)^{L-l}\mathbf{W}_t^l\mathbf{W}_t^{l-1:1}\boldsymbol{\Sigma}\mathbf{W}_t^{1:l-1}\right)$$
$$= -\eta\,\mathrm{tr}\left(\det\left(\mathbf{W}_t^l\right)\left(\mathbf{W}_t^{lT}\mathbf{W}_t^l\right)^{L-l}\mathbf{W}_t^{l-1:1}\boldsymbol{\Sigma}\mathbf{W}_t^{1:l-1}\right)$$
$$= -\eta\det\left(\mathbf{W}_t^l\right)\mathrm{tr}\left(\left(\mathbf{W}_t^{lT}\mathbf{W}_t^l\right)^{L-l}\mathbf{W}_t^{l-1:1}\boldsymbol{\Sigma}\mathbf{W}_t^{1:l-1}\right)$$
$$= -\eta\det\left(\mathbf{W}_t^l\right)\mathrm{tr}(\mathbf{B}(t))$$

We can write the solution of this differential equation as

$$\det\left(\mathbf{W}_t^l\right) = e^{-\eta\int_0^t \mathrm{tr}(\mathbf{B}(s))ds}\det\left(\mathbf{W}_0^l\right)$$

Of course, we have no idea how to solve the integral in the exponential, but the exact solution does not matter as it is always positive and thus not zero. Thus, as long as $\det\left(\mathbf{W}_0^l\right) \neq 0$, also

$\det \left( \mathbf{W}_t^l \right) \neq 0$ holds, at least for a finite time horizon $t < \infty$. $\qquad\qquad\square$

We illustrated this theoretical finding through linear networks in Fig. 5. We give more context here. In Fig. 5 we consider a linear teacher, i.e. we start with a random Gaussian vector $\boldsymbol{x} \sim \mathcal{N}(0, \mathbf{1}_{30 \times 30})$ and we obtain targets through a teacher $\boldsymbol{W} \in \mathbb{R}^{30 \times 10}$ via $\boldsymbol{y} = \boldsymbol{W}\boldsymbol{x}$. We then train a linear network with hidden layer sizes $25, 20, 15$ on this dataset using SGD with learning rate $\eta = 0.00009$. We observe in Fig. 5 how the weights remain full-rank throughout training and as a consequence, the rank of the Hessians remains constant initially. Due to the linear nature of the teacher, an exact training error of zero is achievable. As soon as the error reaches a certain threshold (around $10^{-30}$), the contribution of the functional Hessian starts to vanish as its eigenvalues, one-by-one become too small to count towards the rank. As a consequence, the rank of the loss Hessian also decreases as it is composed of both the functional and the outer Hessian. Also observe how the outer Hessian remains constant throughout the entire optimization, as it does not depend on error. As expected, at the end of training, the loss Hessian collapses onto the outer Hessian.

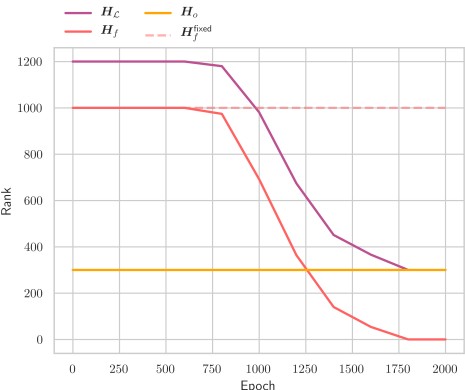

**Figure S1:** Rank dynamics for a linear network with hidden layer sizes $25, 20, 15$, trained using a linear teacher. We show the rank of the Hessians as a function of training time. The dashed line shows how the rank of $\mathbf{H}_f$ evolves when $\boldsymbol{\Omega}$ is kept fixed (denoted as $\mathbf{H}_f^{\text{fixed}}$).

We underline the fact that the decrease in rank comes solely from the vanishing error by calculating the rank of $\mathbf{H}_f$ when $\boldsymbol{\Omega}$ is kept fixed (we call this $\mathbf{H}_f^{\text{fixed}}$. We visualize the result in Fig. S1. Indeed we see that the corresponding rank of $\mathbf{H}_f^{\text{fixed}}$ does not evolve at all but remains constant.

To complete the picture we also show a non-linear network with its corresponding rank dynamics for weights and Hessians in Fig. S2. We use a subset of MNIST, down-scaled to $d = 25$ and hidden layer sizes $25, 20, 15$. We train the model for 2000 epochs with SGD and a learning rate $\eta = 0.09$. We observe that the weights also remain constant througout training when a non-linearity is employed. Moreover, the we have a similar decrease in rank as for the linear case due to the network achieving very small error, making the contribution of $\mathbf{H}_f$ zero eventually. Similarly, we also show the corresponding rank dynamics for fixed $\boldsymbol{\Omega}$ and again observe that the reduction in rank largely comes from the residual tending to zero.

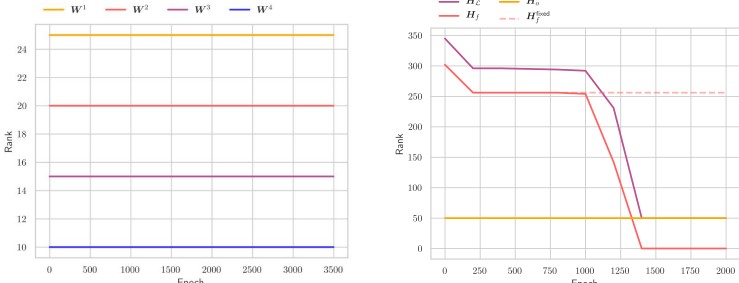

**Figure S2:** Rank dynamics for a ReLU network with hidden layer sizes 25, 20, 15, trained on a subset of MNIST. We show the evolution of the rank of the weights (left) with usual Glorot initialization (guaranteeing maximal rank) and the rank of the Hessians as a function of training time (right). The dashed line shows how the rank of $\mathbf{H}_f$ evolves when $\mathbf{\Omega}$ is kept fixed (denoted as $\mathbf{H}_f^{\text{fixed}}$).

# S6 Pessimistic bounds in the non-linear case

## S6.1 Proof strategy for the $\mathbf{H}_o$ upper bound

The key idea behind our upcoming proof strategy is to do the analysis à la super-position of "unit-networks", i.e., networks with one-hidden neuron. This is because we can reformulate the network function $F_{\boldsymbol{\theta}}$ as sum of network functions of $M$ unit-networks, one per each hidden neuron. Mathematically,

$$F_{\boldsymbol{\theta}}(\mathbf{x}) = \sum_{i=1}^{M} F_{\theta_i}(\mathbf{x}), \quad \text{where,} \quad F_{\theta_i}(\mathbf{x}) = \mathbf{W}^2_{\bullet\,i}\, \sigma(\mathbf{W}^{1^\top}_{i\,\bullet}\, \mathbf{x}).$$

where, $\mathbf{W}^1_{i\,\bullet} \in \mathbb{R}^r$ is the $i$-th row of $\mathbf{W}^1$ and $\mathbf{W}^2_{\bullet\,i} \in \mathbb{R}^K$ is the $i$-th column of $\mathbf{W}^2$. To ease notation, we will henceforth use the shorthand: $\mathbf{V} := \mathbf{W}^1$, and $\mathbf{W} := \mathbf{W}^2$. The column and row indexing follow the convention as before.

Our approach will be to analyze the rank of the outer-product Hessian $\mathbf{H}_o$ in the case of empirical loss, or in other words with finitely many samples. Then, we will do a limiting argument to extend it to case of population loss. The benefit of such an approach is that we can bound the rank of $\mathbf{H}_o$ via the rank of the Jacobian of the function, $\nabla_{\boldsymbol{\theta}} F_{\boldsymbol{\theta}}(\mathbf{X})$ formed over the entire data matrix $\mathbf{X} \in \mathbb{R}^{d \times N}$. We will, in turn, bound the rank of such a Jacobian matrix $\nabla_{\boldsymbol{\theta}} F_{\boldsymbol{\theta}}(\mathbf{X})$ via the ranks of the Jacobian matrices for the unit-network functions $\nabla_{\boldsymbol{\theta}} F_{\theta_i}(\mathbf{X})$.

**Remark R2.** *Similar to Remark R1, when the rank $r$ of the data matrix (or alternatively the empirical (uncentered) input covariance) is less than $d$, we will consider without loss of generality that $\mathbf{X} \in \mathbb{R}^{r \times N}$.*

In other words, for the sake of analysis, we consider that the data matrix has been pre-processed to take into account rank lower than input dimension.

## S6.2 Lemmas on the structure and rank of Jacobian matrix of the unit-network

As a first step, the following Lemma 25 shows the structure of the Jacobian matrix for the unit-network, formed over the entire data matrix.

**Lemma 25.** *Consider the unit-network $F_{\theta_i}(\mathbf{x}) = \mathbf{W}_{\bullet\,i}\, \sigma(\mathbf{V}^\top_{i\,\bullet}\, \mathbf{x})$ corresponding to $i$-th neuron, with the non-linearity $\sigma$ such that $\sigma(z) = \sigma'(z) z$. Let the data-matrix be denoted by $\mathbf{X} \in \mathbb{R}^{r \times N}$. Further, let us denote the matrix which has on its diagonal — the activation derivatives $\sigma'(\mathbf{V}^\top_{i\,\bullet}\, \mathbf{x})$ over all the samples $\mathbf{x}$, and zero elsewhere, by $\mathbf{\Lambda}^i \in \mathbb{R}^{N \times N}$. Then the Jacobian matrix $\nabla_{\boldsymbol{\theta}} F_{\theta_i}(\mathbf{X})$ is given (in transposed form) by,*

$$\nabla_{\boldsymbol{\theta}} F_{\theta_i}(\mathbf{X})^\top = \begin{matrix} \vdots \\ \mathbf{V}_{i\,\bullet} \\ \mathbf{W}_{\bullet\,i} \\ \vdots \end{matrix} \begin{pmatrix} \mathbf{0} \\ \mathbf{X}\,\mathbf{\Lambda}^i \otimes \mathbf{W}^\top_{\bullet\,i} \\ \mathbf{V}^\top_{i\,\bullet}\mathbf{X}\,\mathbf{\Lambda}^i \otimes \mathbf{I}_K \\ \mathbf{0} \end{pmatrix}$$

*Proof.* Let us start simple by computing the gradient with respect to $k$-th component of the function, i.e. $F^k$, which comes out as follows:

$$\frac{\partial F^k}{\partial w_{ij}} = \mathbb{1}\{k = i\}\, \sigma(\mathbf{V}\mathbf{x})_j$$

$$\frac{\partial F^k}{\partial v_{ij}} = w_{ki}\, \sigma'(\mathbf{V}^\top_{i\,\bullet}\, \mathbf{x})\, x_j$$

where $\sigma(\mathbf{V}\mathbf{x})_j = \sigma(\mathbf{V}^\top_{j\,\bullet}\, \mathbf{x})$, with $\mathbf{V}_{j\,\bullet}$ being the $j$-th row of $\mathbf{V}$, *in a column vector format* as mentioned above. And, for example in case of ReLU, $\sigma'(z) = \partial\,\sigma(z)/\partial z = \mathbb{1}\{z > 0\}$.

For a fixed sample $\mathbf{x}$, consider that the activation derivatives for all hidden-neurons are stored in a diagonal matrix $\mathbf{\Lambda}^{\mathbf{x}} \in \mathbb{R}^{M \times M}$, i.e., $\mathbf{\Lambda}^{\mathbf{x}}_{jj} = \sigma'(\mathbf{V}^{\top}_{j \bullet} \mathbf{x})$. Then we can rewrite $\sigma(\mathbf{V}\mathbf{x}) = \mathbf{\Lambda}^{\mathbf{x}} \mathbf{V}\mathbf{x}$ for all non-linearities that satisfy $\sigma(z) = \sigma'(z)z$. So, we have that the Jacobian of the function with respect to the parameters comes out to be,

$$
\nabla_{\boldsymbol{\theta}} F_{\boldsymbol{\theta}}(\mathbf{x})^{\top} =
\begin{array}{c}
\\
\mathbf{V}_{1\bullet} \\
\vdots \\
\mathbf{V}_{i\bullet} \\
\vdots \\
\mathbf{V}_{M\bullet} \\
\mathbf{W}_{1\bullet} \\
\vdots \\
\mathbf{W}_{k\bullet} \\
\vdots \\
\mathbf{W}_{K\bullet}
\end{array}
\begin{array}{ccccc}
\boldsymbol{F}^{1} & \cdots & \boldsymbol{F}^{k} & \cdots & \boldsymbol{F}^{K}
\end{array}
\left(
\begin{array}{ccccc}
w_{11}\,\mathbf{\Lambda}^{\mathbf{x}}_{11}\,\mathbf{x} & \cdots & w_{k1}\,\mathbf{\Lambda}^{\mathbf{x}}_{11}\,\mathbf{x} & & w_{K1}\,\mathbf{\Lambda}^{\mathbf{x}}_{11}\,\mathbf{x} \\
\vdots & & \vdots & & \vdots \\
w_{1i}\,\mathbf{\Lambda}^{\mathbf{x}}_{ii}\,\mathbf{x} & \cdots & w_{ki}\,\mathbf{\Lambda}^{\mathbf{x}}_{ii}\,\mathbf{x} & \cdots & w_{Ki}\,\mathbf{\Lambda}^{\mathbf{x}}_{ii}\,\mathbf{x} \\
\vdots & & \vdots & & \vdots \\
w_{1M}\,\mathbf{\Lambda}^{\mathbf{x}}_{MM}\,\mathbf{x} & \cdots & w_{kM}\,\mathbf{\Lambda}^{\mathbf{x}}_{MM}\,\mathbf{x} & \cdots & w_{KM}\,\mathbf{\Lambda}^{\mathbf{x}}_{MM}\,\mathbf{x} \\
\hline
\mathbf{\Lambda}^{\mathbf{x}}\mathbf{V}\mathbf{x} & \cdots & \mathbf{0} & \cdots & \mathbf{0} \\
\vdots & \ddots & & \ddots & \vdots \\
\mathbf{0} & & \mathbf{\Lambda}^{\mathbf{x}}\mathbf{V}\mathbf{x} & & \mathbf{0} \\
\vdots & \ddots & & \ddots & \vdots \\
\mathbf{0} & \cdots & \mathbf{0} & \cdots & \mathbf{\Lambda}^{\mathbf{x}}\mathbf{V}\mathbf{x}
\end{array}
\right)
$$

Hence, from above we can we write the Jacobian of the $i$-th unit-network function $\mathbf{W}_{\bullet i}\,\sigma(\mathbf{V}^{\top}_{i \bullet} \mathbf{x})$ with respect to the entire set of parameters and at a given input $\mathbf{x}$, as follows:

$$
\nabla_{\boldsymbol{\theta}} F_{\theta_i}(\mathbf{x})^{\top} = \begin{array}{c} \vdots \\ \mathbf{V}_{i\bullet} \\ \mathbf{W}_{\bullet i} \\ \vdots \end{array}
\left(
\begin{array}{c}
\mathbf{0} \\
\mathbf{W}^{\top}_{\bullet i} \otimes \mathbf{\Lambda}^{\mathbf{x}}_{ii}\,\mathbf{x} \\
\mathbf{I}_K \otimes \mathbf{V}^{\top}_{i\bullet}\big(\mathbf{\Lambda}^{\mathbf{x}}_{ii}\,\mathbf{x}\big) \\
\mathbf{0}
\end{array}
\right)
\overset{(a)}{=}
\begin{array}{c} \vdots \\ \mathbf{V}_{i\bullet} \\ \mathbf{W}_{\bullet i} \\ \vdots \end{array}
\left(
\begin{array}{c}
\mathbf{0} \\
\mathbf{\Lambda}^{\mathbf{x}}_{ii}\,\mathbf{x} \otimes \mathbf{W}^{\top}_{\bullet i} \\
\mathbf{V}^{\top}_{i\bullet}\big(\mathbf{\Lambda}^{\mathbf{x}}_{ii}\,\mathbf{x}\big) \otimes \mathbf{I}_K \\
\mathbf{0}
\end{array}
\right),
$$

where, in (a) we have used the fact that for vectors $\mathbf{a}$, $\mathbf{b}$ we have that $\mathbf{a}^{\top} \otimes \mathbf{b} = \mathbf{b} \otimes \mathbf{a}^{\top} = \mathbf{b}\,\mathbf{a}^{\top}$, as well as the fact that $\mathbf{V}^{\top}_{i\bullet}\big(\mathbf{\Lambda}^{\mathbf{x}}_{ii}\,\mathbf{x}\big)$ is a scalar which allows us to commute the factors in the corresponding Kronecker product.

Finally, we can express the above Jacobian across all the samples in the data matrix, as stated in the lemma:

$$
\nabla_{\boldsymbol{\theta}} F_{\theta_i}(\mathbf{X})^{\top} = \begin{array}{c} \vdots \\ \mathbf{V}_{i\bullet} \\ \mathbf{W}_{\bullet i} \\ \vdots \end{array}
\left(
\begin{array}{c}
\mathbf{0} \\
\mathbf{X}\,\mathbf{\Lambda}^{i} \otimes \mathbf{W}^{\top}_{\bullet i} \\
\mathbf{V}^{\top}_{i\bullet}\mathbf{X}\,\mathbf{\Lambda}^{i} \otimes \mathbf{I}_K \\
\mathbf{0}
\end{array}
\right).
$$

Here, we utilized that $\mathbf{A} \otimes \mathbf{B} = [\mathbf{A}_{\bullet 1} \otimes \mathbf{B}, \cdots, \mathbf{A}_{\bullet n} \otimes \mathbf{B}]$ for some arbitrary matrix $\mathbf{A}$ containing $n$ columns. Besides, we have collected the activation derivatives for the $i$-th neuron, i.e., $\mathbf{\Lambda}^{\mathbf{x}}_{ii} = \sigma'(\mathbf{V}^{\top}_{i\bullet}\,\mathbf{x})$, across all samples $\mathbf{x}$, into the diagonal matrix $\mathbf{\Lambda}^{i} \in \mathbb{R}^{N \times N}$. $\qquad\square$

From the above Lemma, we can also see that the benefit of analyzing via the unit-networks is that we only have to deal with the activation derivatives of a single neuron at a time. Besides, now that we know the structure of the unit-network Jacobian, we will analyze its rank. But before, let's recall the assumption A2 from the main text, in our current notation:

**Assumption A2.** *For each active hidden neuron $i$, the weighted input covariance has the same rank as the overall input covariance, i.e.,* $\mathrm{rk}(\mathbf{E}\,[\alpha_{\mathbf{x}}\,\mathbf{x}\mathbf{x}^{\top}]) = \mathrm{rk}(\mathbf{\Sigma}_{\mathbf{xx}}) = r$, *with* $\alpha_{\mathbf{x}} = \sigma'(\mathbf{x}^{\top}\,\mathbf{V}_{i\bullet})^2$.

This assumption can be translated into finite-sample case as follows. First, note that the (uncentered) input covariance $\mathbf{\Sigma}_{\mathbf{xx}}$ corresponds to $\frac{1}{N}\mathbf{X}\mathbf{X}^{\top}$, while the weighted covariance $\mathbf{E}\,[\alpha_{\mathbf{x}}\,\mathbf{x}\mathbf{x}^{\top}]$ corresponds to the matrix $\frac{1}{N}\mathbf{X}\mathbf{\Lambda}^{i}\mathbf{\Lambda}^{i}\mathbf{X}^{\top}$. This is straightforward to check, and notice $\alpha_{\mathbf{x}} = (\mathbf{\Lambda}^{\mathbf{x}}_{ii})^2$.

Then, the equivalent assumption is to require $\mathrm{rk}\left(\mathbf{X}\boldsymbol{\Lambda}^i\boldsymbol{\Lambda}^i\mathbf{X}^\top\right) = \mathrm{rk}\left(\mathbf{X}\mathbf{X}^\top\right) = r$, ignoring the constant $\frac{1}{N}$ which does not affect rank. Further, since for any arbitrary matrix $\mathbf{A}$, we have that $\mathrm{rk}(\mathbf{A}\mathbf{A}^\top) = \mathrm{rk}(\mathbf{A})$. Thus, our equivalent assumption can be simplified to as follows:

**Assumption A2′.** *(finite-sample equivalent) For each active hidden neuron $i$, assume that* $\mathrm{rk}\left(\mathbf{X}\boldsymbol{\Lambda}^i\right) = \mathrm{rk}\left(\mathbf{X}\right) = r$, *where $\boldsymbol{\Lambda}^i$, as detailed before, contains the activation derivatives across all samples for this neuron $i$.*

**Lemma 26.** *Under the same setup as Lemma 25 and Assumptions A1, A2 (or equivalently A2′), the rank of the Jacobian matrix, $\nabla_{\boldsymbol{\theta}}F_{\boldsymbol{\theta}_i}(\mathbf{X})$, of the $i$-th unit-network is given by:*

$$\mathrm{rk}(\nabla_{\boldsymbol{\theta}}F_{\boldsymbol{\theta}_i}(\mathbf{X})) = r + K - 1\,.$$

*Proof.* From Lemma 25, the Jacobian matrix is given by (ignoring the zero blocks which do not matter for the analysis of rank),

$$\nabla_{\boldsymbol{\theta}}F_{\boldsymbol{\theta}_i}(\mathbf{X})^\top = \begin{pmatrix} \mathbf{X}\,\boldsymbol{\Lambda}^i \otimes \mathbf{W}_{\bullet i}^\top \\ \mathbf{V}_{i\bullet}^\top\mathbf{X}\,\boldsymbol{\Lambda}^i \otimes \mathbf{I}_K \end{pmatrix} = \underbrace{\begin{pmatrix} \mathbf{I}_r \otimes \mathbf{W}_{\bullet i}^\top \\ \mathbf{V}_{i\bullet}^\top \otimes \mathbf{I}_K \end{pmatrix}}_{\mathbf{A}_i \in \mathbb{R}^{(r+K)\times Kr}} \underbrace{\left(\mathbf{X}\,\boldsymbol{\Lambda}^i \otimes \mathbf{I}_K\right)}_{\in \mathbb{R}^{Kr \times KN}}$$

Now this factorization reveals the familiar $\mathbf{Z}$-like structure, and so the matrix labelled $\mathbf{A}_i$ in the above factorization has rank equal to $r + K - 1$ by Lemma 1. And, $\mathrm{rk}(\mathbf{X}\,\boldsymbol{\Lambda}^i \otimes \mathbf{I}_K) = K\,\mathrm{rk}(\mathbf{X}\,\boldsymbol{\Lambda}^i) = Kr$, by employing assumption A2′. Thus, this matrix $\mathbf{X}\,\boldsymbol{\Lambda}^i \otimes \mathbf{I}_K$ is right invertible. Hence, we have:

$$\mathrm{rk}(\nabla_{\boldsymbol{\theta}}F_{\boldsymbol{\theta}_i}(\mathbf{X})) = \mathrm{rk}(\nabla_{\boldsymbol{\theta}}F_{\boldsymbol{\theta}_i}(\mathbf{X})^\top) = \mathrm{rk}(\mathbf{A}_i) = r + K - 1\,.$$

$\square$

### S6.3 Proof of Theorem 9

Now, that we are equipped to prove the Theorem, and let us recall its statement from the main text:

**Theorem 9.** *Consider a 1-hidden layer network with non-linearity $\sigma$ such that $\sigma(z) = \sigma'(z)z$ and let $\widetilde{M}$ be the # of active hidden neurons (i.e., probability of activation $> 0$). Then, under assumption A1 and A2, rank of $\mathbf{H}_o$ is given as, $\mathrm{rk}(\mathbf{H}_o) \leq r\widetilde{M} + \widetilde{M}K - \widetilde{M}$.*

*Proof.* In the case of empirical loss (i.e., finite-sample case), we can express the outer-product Hessian as $\mathbf{H}_o^N = \frac{1}{N}\nabla_{\boldsymbol{\theta}}F_{\boldsymbol{\theta}}(\mathbf{X})^\top \nabla_{\boldsymbol{\theta}}F_{\boldsymbol{\theta}}(\mathbf{X}) = \frac{1}{N}\sum_{i=1}^N \nabla_{\boldsymbol{\theta}}F_{\boldsymbol{\theta}}(\mathbf{x}^i)^\top \nabla_{\boldsymbol{\theta}}F_{\boldsymbol{\theta}}(\mathbf{x}^i)$. It is clear that rank of $\mathbf{H}_o^N$ is the same as the rank of $\nabla_{\boldsymbol{\theta}}F_{\boldsymbol{\theta}}(\mathbf{X})$ as $\mathrm{rk}(\mathbf{A}^\top\mathbf{A}) = \mathrm{rk}(\mathbf{A})$ for any arbitrary matrix $\mathbf{A}$. Thus we have that,

$$\mathrm{rk}(\mathbf{H}_o^N) = \mathrm{rk}(\nabla_{\boldsymbol{\theta}}F_{\boldsymbol{\theta}}(\mathbf{X})) = \mathrm{rk}\big(\sum_{i=1}^M \nabla_{\boldsymbol{\theta}}F_{\boldsymbol{\theta}_i}(\mathbf{X})\big) \leq \sum_{i=1}^M \mathrm{rk}(\nabla_{\boldsymbol{\theta}}F_{\boldsymbol{\theta}_i}(\mathbf{X}))$$

$$\overset{\text{Lemma 26}}{\leq} \sum_{i=1}^{\widetilde{M}} r + K - 1 = r\widetilde{M} + \widetilde{M}K - \widetilde{M}\,.$$

The first inequality is because of subadditivity of rank, i.e., $\mathrm{rk}(\mathbf{A} + \mathbf{B}) \leq \mathrm{rk}(\mathbf{A}) + \mathrm{rk}(\mathbf{B})$. Next, here we only sum over the active hidden neurons, whose count is $\widetilde{M}$. Because, for dead neurons $\nabla_{\boldsymbol{\theta}}F_{\boldsymbol{\theta}}(\mathbf{X}) = \mathbf{0}$, and thus rank of Jacobian for dead unit-networks will be zero.

Now, in order to extend this to case of population loss, we essentially have to consider the limit of $N \to \infty$. As we can see from the analysis so far, the rank of the outer-product Hessian $\mathbf{H}_o^N$ is always bounded by $r\widetilde{M} + \widetilde{M}K - \widetilde{M}$ for any finite $N \geq N_0$, where $N_0$ is the minimum number of samples that are needed for the assumption A2 to hold.

Thus, we have a sequence of matrices $\{\mathbf{H}_o^N\}_{N \geq N_0}$, each of which has rank bounded above by $r\widetilde{M} + \widetilde{M}K - \widetilde{M}$. Because, matrix rank is a lower semi-continuous function, the above sequence will converge to a matrix, $\mathbf{H}_o$ of the population loss, with rank at most $r\widetilde{M} + \widetilde{M}K - \widetilde{M}$. Therefore,

$$\mathrm{rk}(\mathbf{H}_o) \leq r\widetilde{M} + \widetilde{M}K - \widetilde{M}\,.$$

$\square$

### S6.4 Note on the assumption

The assumption $A2'$ that $\mathrm{rk}(\mathbf{X}\boldsymbol{\Lambda}^i) = r$, holds as soon as $\mathrm{rk}(\boldsymbol{\Lambda}^i) \geq r$ in expectation. This is something that depends on the data distribution but only mildly. For instance, one such scenario is when we use the typical form of initialization $v_{ij} \overset{i.i.d.}{\sim} \mathcal{N}(0,1)$, then conditioned on a fixed example $\mathbf{x}$, we have $\mathbf{V}_{i\bullet}^\top \mathbf{x} \sim \mathcal{N}(0, \|\mathbf{x}\|^2)$. To further consolidate this point, let us consider $\sigma(z) = \mathrm{ReLU}(z) = \max(z, 0)$. Then, for instance if the underlying *data distribution is symmetric*, the entries of $\boldsymbol{\Lambda}^i$ — which are nothing but $\sigma'(\mathbf{V}_{i\bullet}^\top \mathbf{x}) = \mathbb{1}\{\mathbf{V}_{i\bullet}^\top \mathbf{x} > 0\}$ — will be non-zero with probability $\frac{1}{2}$. The rank of the diagonal matrix $\boldsymbol{\Lambda}^i$ just amounts to the number of non-zero entries. Hence, in expectation, as soon as we have at least $2r$ examples, or more simply $2d$ examples since $d \geq r$, we should be fine.

# S7 Rank of the Hessian with bias

## S7.1 Proof of Theorem 12

We consider the case where each layer implements an affine mapping instead of a linear. So now we have additional parameter vectors for these bias terms, $\mathbf{b}^1, \cdots, \mathbf{b}^L$, and we can write the network function as:

$$F(\mathbf{x}) = \mathbf{W}^L \left( \cdots \left( \mathbf{W}^2 \left( \mathbf{W}^1 \mathbf{x} + \mathbf{b}^1 \right) + \mathbf{b}^2 \right) \cdots \right) + \mathbf{b}^L$$

In terms of a recursive expansion, it can also be written in the following manner:

$$F(\mathbf{x}) := F^{L:1}(\mathbf{x}) = \mathbf{W}^L F^{L-1:1}(\mathbf{x}) + \mathbf{b}^L, \quad \text{where} \quad F^0(\mathbf{x}) = \mathbf{x}. \tag{26}$$

We will also use the notation $F^{1:l}$ to mean $F^{l:1\top}$. Let us recall the assumption and the theorem stated in the main text:

**Assumption A3.** *The input data has zero mean, i.e., $\mathbf{x} \sim p_{\mathbf{x}}$ is such that $\mathbf{E}[\mathbf{x}] = 0$.*

In other words, we assume that the input data has zero mean, which is actually a standard practical convention.

**Theorem 12.** *Under the assumption A1 and A3, for a deep linear network with bias, the rank of $\mathbf{H}_o$ is upper bounded as, $\mathrm{rk}(\mathbf{H}_o) \leq q(r + K - q) + K$, where $q := \min(r, M_1, \cdots, M_{L-1}, K)$.*

*Proof.* Since the above function, Eq. (26), is of a similar form as the one in Eq. (1), we use the matrix-derivative rule in order to obtain the following expression of the *(transposed)* Jacobian at a point $(\mathbf{x}, \mathbf{y})$:

$$\nabla F(\mathbf{x})^\top = \begin{array}{c} \mathrm{vec}_r(\mathbf{W}^1) \\ \vdots \\ \mathrm{vec}_r(\mathbf{W}^l) \\ \vdots \\ \mathrm{vec}_r(\mathbf{W}^L) \\ \mathbf{b}^1 \\ \vdots \\ \mathbf{b}^l \\ \vdots \\ \mathbf{b}^L \end{array} \left( \begin{array}{c} \mathbf{W}^{2:L} \otimes F^0(\mathbf{x}) \\ \vdots \\ \mathbf{W}^{l+1:L} \otimes F^{l-1:1}(\mathbf{x}) \\ \vdots \\ \mathbf{I}_K \otimes F^{L-1:1}(\mathbf{x}) \\ \mathbf{W}^{2:L} \\ \vdots \\ \mathbf{W}^{l+1:L} \\ \vdots \\ \mathbf{I}_K \end{array} \right) \tag{27}$$

*Comment about the Hessian indexing:* We will assume that the blocks from $[1, \cdots, L]$ index the weight matrices and those from $[L+1, \cdots, 2L]$ index the bias parameters.

Recall the outer-product Hessian $\mathbf{H}_o$ in the case of mean-squared loss is given by

$$\mathbf{H}_o = \mathbf{E} \left[ \nabla F(\mathbf{x})^\top \nabla F(\mathbf{x}) \right].$$

Let us look at the expression for the $kl$-th block, for $k, l \in [L]$ (i.e., from the sub-matrix corresponding to *weight-weight Hessian*):

$$\mathbf{H}_o^{kl} = \mathbf{E} \left[ \mathbf{W}^{k+1:L} \mathbf{W}^{L:l+1} \otimes F^{k-1:1}(\mathbf{x}) F^{1:l-1}(\mathbf{x}) \right] \tag{28}$$

$$= \mathbf{W}^{k+1:L} \mathbf{W}^{L:l+1} \otimes \mathbf{E} \left[ F^{k-1:1}(\mathbf{x}) F^{1:l-1}(\mathbf{x}) \right] \tag{29}$$

Now, let us make use of the assumption A3. Once we have applied this, the dependence on input is only via the uncentered covariance of input (or the second moment matrix). Alongside we have terms corresponding to $F^{l-1:1}(\mathbf{0})$, which is the output of the network when $\mathbf{0}$ is passed as the input. Overall, using the zero-mean assumption in Eq. (29) yields:

$$\mathbf{H}_o^{kl} = \underbrace{\mathbf{W}^{k+1:L}\mathbf{W}^{L:l+1} \otimes \mathbf{W}^{k-1:1}\boldsymbol{\Sigma}_{\mathbf{xx}}\mathbf{W}^{1:l-1}}_{\text{Expression in the linear, non-bias, case}} + \underbrace{\mathbf{W}^{k+1:L}\mathbf{W}^{L:l+1} \otimes F^{k-1:1}(\mathbf{0})F^{1:l-1}(\mathbf{0})}_{\text{New terms containing bias}}$$

(30)

We see that first part of the expression is identical to the linear case without bias, and it is only the second part that contains the bias terms.

Similarly, for the *bias-bias Hessian* blocks $\mathbf{H}_o^{kl}$ such that $k, l \in [L \cdots 2L]$, there is no dependence on input at all and contains only bias terms. Likewise, the *weight-bias Hessian* blocks has no dependence on the input.

Hence, it seems quite natural to separately analyze the terms without bias and with bias. So, the rank of the first non-bias part comes directly from our previous analysis of Theorem 3 and is equal to $q(r + K - q)$.

The analysis for the left-over bias part is not too hard either. This can be simply decomposed as the product $\mathbf{B}_o\mathbf{B}_o^\top$, where $\mathbf{B}_o$ is given by:

$$\mathbf{B}_o = \begin{matrix} \text{vec}_r(\mathbf{W}^2) \\ \vdots \\ \text{vec}_r(\mathbf{W}^l) \\ \vdots \\ \text{vec}_r(\mathbf{W}^L) \\ \mathbf{b}^1 \\ \vdots \\ \mathbf{b}^l \\ \vdots \\ \mathbf{b}^L \end{matrix} \begin{pmatrix} \mathbf{W}^{3:L} \otimes F^1(\mathbf{0}) \\ \vdots \\ \mathbf{W}^{l+1:L} \otimes F^{l-1:1}(\mathbf{0}) \\ \vdots \\ \mathbf{I}_K \otimes F^{L-1:1}(\mathbf{0}) \\ \mathbf{W}^{2:L} \\ \vdots \\ \mathbf{W}^{l+1:L} \\ \vdots \\ \mathbf{I}_K \end{pmatrix}$$

(31)

If we compare this expression to that in Eq. 27, we see that there is no block corresponding to the first row there, as $F^0(\mathbf{0}) = \mathbf{0}$. Then, one simply has to notice that the last block in $\mathbf{B}_o$, which essentially corresponds to the parameter $\mathbf{b}^L$, is the $K \times K$ identity matrix $\mathbf{I}_K$. Hence, the matrix $\mathbf{B}_o$ which itself has $K$ columns, has rank equal to $K$, using Lemma 16. Then, the rank of the bias part is equal to that of $\text{rk}(\mathbf{B}_o) = K$, since we know that $\text{rk}(\mathbf{A}\mathbf{A}^\top) = \text{rk}(\mathbf{A})$.

Finally, we use the subadditivity of rank, i.e., $\text{rk}(\mathbf{A} + \mathbf{B}) \leq \text{rk}(\mathbf{A}) + \text{rk}(\mathbf{B})$, on this decomposition of the outer-product Hessian into outer-product Hessian for non-bias and the new terms containing the bias parameters. Thus, we obtain that:

$$\text{rk}(\mathbf{H}_o) \leq q(r + K - q) + K \,.$$

$\square$

## S7.2 Formulas for two layer networks

For two layer (1-hidden layer) networks with linear activation and $M_1$ hidden units, $d$ dimensional input and $K$ classes, empirical evidence seems to suggest the following. Define $s = \min(r, K)$ and $q = \min(r, M_1, K)$. Let us define $s' := \min(r + 1, K)$. Then we find:

- $\mathrm{rk}(\mathbf{H}_o) = q\,(r + K - q) + K$
- $\mathrm{rk}\,(\mathbf{H}_f) = 2sM_1 + \mathbb{1}\{K > r\}\,2M_1 = 2\min(K, r + 1)\,M_1 = 2s'M_1$
- $\mathrm{rk}\,(\mathbf{H}_{\mathcal{L}}) = 2\min(K, r+1)\,(M_1-q) + q\,(K+r+1) + K = 2s'(M_1-q) + q\,(K+r+1) + K$

If we compare the upper-bounds for the scenario without bias to the one with bias, we find that change in the rank of $\mathbf{H}_f$ is due to changing $r \to r + 1$ in the formula, which makes sense as bias can be understood as adding a homogeneous coordinate in the input. For $\mathbf{H}_o$, the rank formula now includes an additive term of $K$. And both these changes together affect the change in rank for $\mathbf{H}_{\mathcal{L}}$.

## S7.3 Formulas for $L$-layer networks

The upper-bound for $\mathbf{H}_o$ that we noted in the previous section also holds for the general case, as evident from our proof in Section S7.1. Empirically as well, we obtain $\mathrm{rk}(\mathbf{H}_o) = q\,(r+K-q)+K$ as the exact formula.

For the functional and overall Hessian, we list formulas that seem to hold *empirically* for the non-bottleneck case. Here, the input size has to take into account the homogeneous coordinate, so by non-bottleneck it is meant that $M_i \geq \min(r + 1, K)$, $\forall\, i \in [1, \cdots, L - 1])$.

Define $q' = \min(r+1, M_1, \cdots, M_{L-1}, K) = \min(r+1, K)$, which because of our non-bottleneck assumption comes out to be same as the $s'$ in the previous section.

- $\mathrm{rk}\,(\mathbf{H}_f) = 2q'\left(\sum_{i=1}^{L-1} M_i\right) + 2q's' - Lq'^2 + (L-2)q'$
- $\mathrm{rk}\,(\mathbf{H}_{\mathcal{L}}) = 2q'\left(\sum_{i=1}^{L-1} M_i\right) + q'(r+K) - Lq'^2 + Lq' = 2q'M + q'(r+K) - Lq'^2 + Lq'$

Let us compare the above bound to the rank of Hessian $\mathbf{H}_{\mathcal{L}}$ in the linear case with bias by assuming that the output layer has the smallest size, i.e., $q' = K$.

Then for linear case **without bias**:

$$\mathrm{rk}(\mathbf{H}_{\mathcal{L}}) = 2K\,M - L\,K^2 + K(r + K)$$

While for linear case **with bias**:

$$\mathrm{rk}(\mathbf{H}_{\mathcal{L}}) = 2K\,M - L\,K^2 + K(r + K) + LK$$

Basically, we just have an additional term of $LK$ in the rank, whereas the additional number of parameters are,

$$\sum_{i=1}^{L} M_i \geq LK\,.$$

## S7.4 Effect of bias on $\frac{\text{rank}}{\#params}$

In Fig. S3 we showcase the resulting effect of enabling bias in the network on the Hessian rank, by simulating the ratio $\frac{\text{rank}}{\#params}$ across increasing depth, for the loss Hessian $\mathbf{H}_{\mathcal{L}}$ with and without bias. We find that in both cases the $\frac{\text{rank}}{\#params}$ curve saturates to a small threshold. But interestingly, we see that enabling bias further results in a decrease in this ratio.

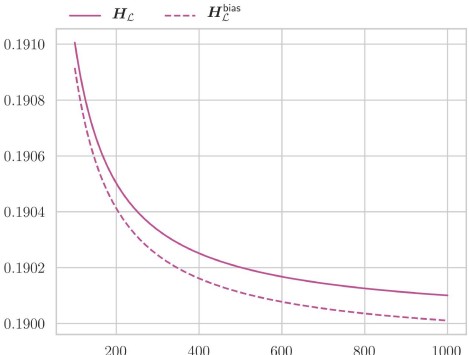

**Figure S3:** We simulate $\frac{\text{rank}}{\#\text{params}}$ over depth for $\mathbf{H}_{\mathcal{L}}$, $\mathbf{H}_{\mathcal{L}}^{\text{bias}}$ — in other words, with bias disabled and bias enabled. For this, we use $M = 1000$, $d = r = 784$ and $K = 100$.

# S8 Properties of the Hessian Spectrum

## S8.1 Spectrum of outer-product Hessian

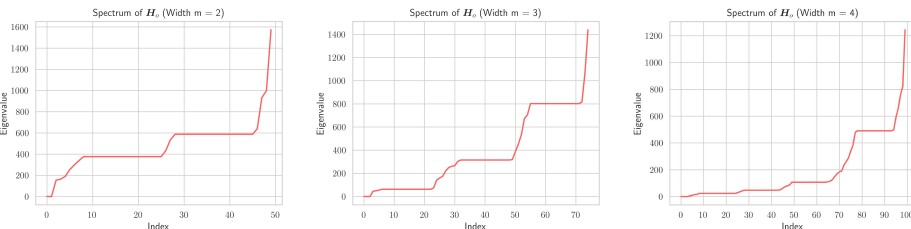

**Figure S4:** *Spectrum of outer product.* $\mathbf{H}_o$ spectrum has $q$ plateaus of size $K - M_{L-1}$ located at the eigenvalues of $\mathbf{E}\left[F^{L-1:1}(\mathbf{x})\,F^{1:L-1}(\mathbf{x})\right]$, even with non-linearities and for any $L$. Here, $K = 20$, and $q = M_{L-1} = 2, 3, 4$ in each of the sub-figures respectively. We use Gaussian mixture data of dimension 5.

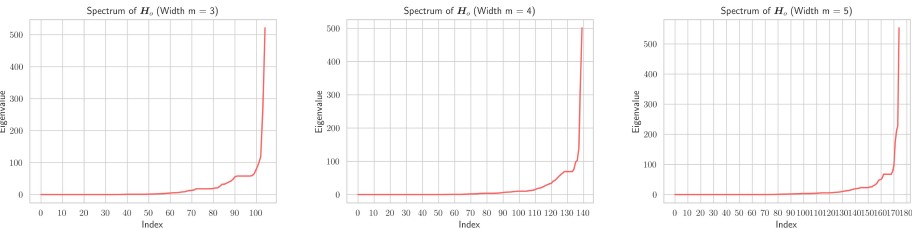

**Figure S5:** *Spectrum of outer product.* $\mathbf{H}_o$ spectrum has $q$ plateaus of size $K - M_{L-1}$ located at the eigenvalues of $\mathbf{E}\left[F^{L-1:1}(\mathbf{x})\,F^{1:L-1}(\mathbf{x})\right]$, even with non-linearities and for any $L$. Here, $K = 10$, and $q = M_{L-1} = 3, 4, 5$ in each of the sub-figures respectively. We use down scaled MNIST $d = 25$.

The eigenvalues of the outer-product term of the Hessian, which is the one that dominates the spectrum near the end of training, can be written in closed-form for fully-connected neural networks, with linear activations.

Recall from Proposition. (2) that the outer-product Hessian can be decomposed as follows, $\mathbf{H}_o = \mathbf{A}_o(\mathbf{I}_K \otimes \boldsymbol{\Sigma}_{\mathbf{xx}})\mathbf{A}_o^\top$, where $\mathbf{A}_o \in \mathbb{R}^{p \times Kd}$ is as follows:

$$
\mathbf{A}_o = \begin{pmatrix} \mathbf{W}^{2:L} \otimes \mathbf{I}_d \\ \vdots \\ \mathbf{W}^{\ell+1:L} \otimes \mathbf{W}^{\ell-1:1} \\ \vdots \\ \mathbf{I}_K \otimes \mathbf{W}^{L-1:1} \end{pmatrix}
$$

Since $\mathbf{AB}$ and $\mathbf{BA}$ have the same non-zero eigenvalues, we have that eigenvalues of $\mathbf{H}_o$ are the same as $\tilde{\mathbf{H}}_o = \mathbf{A}_o^\top \mathbf{A}_o(\mathbf{I}_K \otimes \boldsymbol{\Sigma}_{\mathbf{xx}})$, and notice $\mathbf{A}_o^\top \mathbf{A}_o \in \mathbb{R}^{Kd \times Kd}$ and comes out to be,

$$
\mathbf{A}_o^\top \mathbf{A}_o = \sum_{\ell=1}^{L} \mathbf{W}^{L:\ell+1} \mathbf{W}^{\ell+1:L} \otimes \mathbf{W}^{1:\ell-1} \mathbf{W}^{\ell-1:1}
$$

This is nothing but the diagonal-blocks of the Hessian-outer product added in the "transposed" fashion. Hence we have the result on the eigenvalues (evals) that,

$$
\text{evals}(\mathbf{H}_o) = \text{evals}(\mathbf{A}_o^\top \mathbf{A}_o\,(\mathbf{I}_K \otimes \boldsymbol{\Sigma}_{\mathbf{xx}})) = \text{evals}\left(\sum_{\ell=1}^{L} \mathbf{W}^{L:\ell+1} \mathbf{W}^{\ell+1:L} \otimes \mathbf{W}^{1:\ell-1} \mathbf{W}^{\ell-1:1} \boldsymbol{\Sigma}_{\mathbf{xx}}\right)
$$

**Repeated eigenvalues.** A consequence of this is that a plateau of repeated eigenvalue exists, whenever the last layer is strictly bigger than the penultimate layer. In fact, this plateau phenomenon also holds for non-linear networks, since even for such networks the last layer is not usually followed by non-linearities.

Notice, that when $K > M_{L-1}$, the matrix $\mathbf{W}^{L:\ell+1}\mathbf{W}^{\ell+1:L}$ in the left part of the Kronecker product will have rank at most $M_{L-1}$, except for the case when $\ell = L$. There, for $\ell = L$, we obtain a identity matrix, $\mathbf{I}_K$, in the left part of the Kronecker product, whose rank is of course $K$. Thus when all terms are added up together, $K - M_{L-1}$ times the eigenvalues of $\mathbf{W}^{1:L-1}\mathbf{W}^{L-1:1}\mathbf{\Sigma_{xx}}$ will show up for the overall Hessian $\mathbf{H}_o$ as well. Obviously, since Kronecker product with identity implies eigenvalues of the other matrix are multiplied by 1. This results in the plateaued behaviour of the eigenvalue spectrum. We illustrate this finding in Figure S4 for a ReLU network on Gaussian mixture data. We also show the results for MNIST in Figure S5. Due to the spectrum being not as cleanly separated as for the Gaussian case, the results are not as clearly visible but still hold exactly as verified experimentally.

## S8.2  Spectrum of Functional Hessian

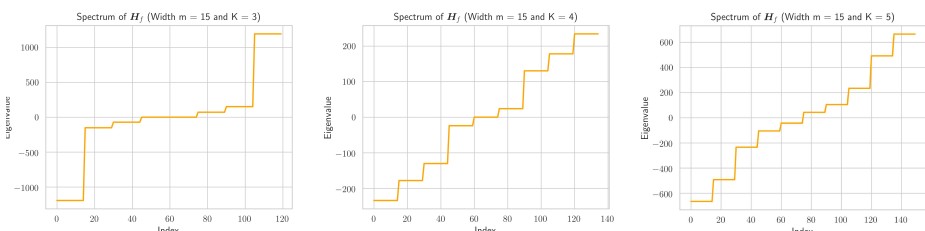

**Figure S6:** *Spectrum of Functional Hessian.* We use a Gaussian mixture of dimension $d = 5$ and a linear model with one hidden layer of size $M = 15$. We vary the number of classes $K = 3, 4, 5$ in each of the sub-figures respectively. Notice that we have $2K$ plateaus of width $M = 15$.

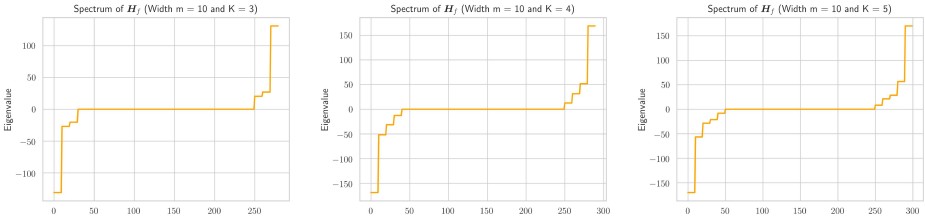

**Figure S7:** *Spectrum of Functional Hessian.* We use down-sampled MNIST of dimension $d = 25$ and a linear model with one hidden layer of size $M = 10$. We vary the number of classes $K = 3, 4, 5$ in each of the sub-figures respectively. Notice that we have $2K$ plateaus of width $M = 10$.

Consider the case of 1-hidden layer network with $M$ hidden neurons. We notice that the functional Hessian in the linear case has a interesting step-like structure in the spectrum, while in the non-linear case empirically appears to interpolate or pass through it.

Here, the functional Hessian part is given as follows:

$$\mathbf{H}_f = \begin{pmatrix} \mathbf{0}_{KM} & \mathbf{\Omega} \otimes \mathbf{I}_M \\ \mathbf{\Omega}^\top \otimes \mathbf{I}_M & \mathbf{0}_{dM} \end{pmatrix},$$

where, $\mathbf{\Omega} = \mathbf{E}[\boldsymbol{\delta}_{\mathbf{x},\mathbf{y}}\, \mathbf{x}^\top]$ as before. Now since eigenvalues $\lambda$ are given by the solution to the characteristic polynomial, $\rho(\lambda) = det(\mathbf{H}_f - \lambda\mathbf{I}_p)$, where $p = dM + KM$ denotes the total number of parameters. We can further write it as,

$$\mathbf{H}_f = \begin{pmatrix} -\lambda \mathbf{I}_{KM} & \mathbf{\Omega} \otimes \mathbf{I}_M \\ \mathbf{\Omega}^\top \otimes \mathbf{I}_M & -\lambda \mathbf{I}_{dM} \end{pmatrix}$$

Now, we consider the determinant formula through the Schur complement assuming the block matrix $\mathbf{A}$ is invertible, i.e.,

$$det(\mathbf{M}) = det \begin{pmatrix} \mathbf{A} & \mathbf{B} \\ \mathbf{C} & \mathbf{D} \end{pmatrix} = det(\mathbf{A}) \, det(\mathbf{D} - \mathbf{C}\mathbf{A}^{-1}\mathbf{B})$$

Hence, in our case we obtain:

$$\rho(\lambda) = (-\lambda)^{KM} \, det \left( -\lambda^2 \, \mathbf{I}_{dM} + (\mathbf{\Omega}^\top \mathbf{\Omega} \otimes \mathbf{I}_M) \right)$$

Where we can see that $det \left( -\lambda^2 \, \mathbf{I}_{dM} + (\mathbf{\Omega}^\top \mathbf{\Omega} \otimes \mathbf{I}_M) \right)$ corresponds to the characteristic polynomial of the matrix $\mathbf{Z} = (\mathbf{\Omega}^\top \mathbf{\Omega}) \otimes \mathbf{I}_M$ and with each eigenvalue of $\mathbf{Z}$ occurring with both as positive and negative signs as eigenvalues of $\mathbf{H}_f$, repeated $M$ times. See Figure S6 for Gaussian mixture data and Figure S7 for down-sampled MNIST.

## S9 Detailed Empirical Results

Here we collect the variety of experiments omitted from the main text due to space constraints. We begin by providing further evidence for the validity of our rank predictions for linear networks by varying the dataset and the loss function employed in the calculation of the Hessian. We then present more experiments for the non-linear case, showing more spectral plots and reconstruction errors for more non-linearities. Finally, also show how our rank predictions also extend to the neural tangent kernel.

Experiments were implemented in the JAX framework[S2] and performed on CPU (AMD EPYC 7H12) with 256 GB memory.

### S9.1 Verification of Rank Predictions for Linear Networks

We verify our formulas for MNIST [45], CIFAR10 [47] and FASHION-MNIST [46]. Moreover we employ diverse losses such as mean-squared error, cross entropy loss and cosh loss. We show the dynamics of rank as a function of sample size, minimal width and depth.

For all the considered settings, we observe exact matches across all datasets and all losses. We structure the experiments as follows. We group by loss functions, starting with MSE, then cross entropy and then cosh loss. For each loss function, we vary the sample size, width and depth of the architecture for the three datasets. Finally, we vary the initialization scheme and study the effect of sample size, width and depth for MSE loss on CIFAR10. Finally, we verify the predictions for architectures that use bias, again using MSE loss on CIFAR10. For all the experiments, we down-sample the corresponding dataset to dimensionality $d = 64$ for width and sample size plots, while for depth plots, in order to be able to use deeper models, we down-sample to $d = 16$. We use $N = 300$ number of samples for all experiments.

#### S9.1.1 Mean Squared Error (MSE)

Here we perform more experiments in the spirit of Figure 2c. We also show the rank dynamics with varying depth and only present the normalized plots for both width and depth. Figure S8 shows the results for CIFAR10, S9 for FashionMNIST and S10 for MNIST. We observe a perfect match for all the datasets.

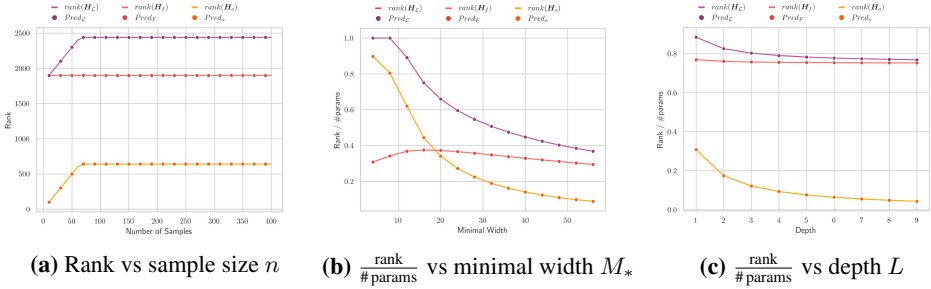

**(a)** Rank vs sample size $n$     **(b)** $\frac{\text{rank}}{\#\,\text{params}}$ vs minimal width $M_*$     **(c)** $\frac{\text{rank}}{\#\,\text{params}}$ vs depth $L$

**Figure S8:** Behaviour of rank and rank/#params on CIFAR10 using MSE, with hidden layers: $50, 20, 20, 20$ (Fig. S8a), $M_*, M_*$ (Fig. S8b) and $L$ layers of width $M = 25$ (Fig. S8c). The lines indicate the true value and circles denote our formula predictions.

---

[S2]https://github.com/google/jax

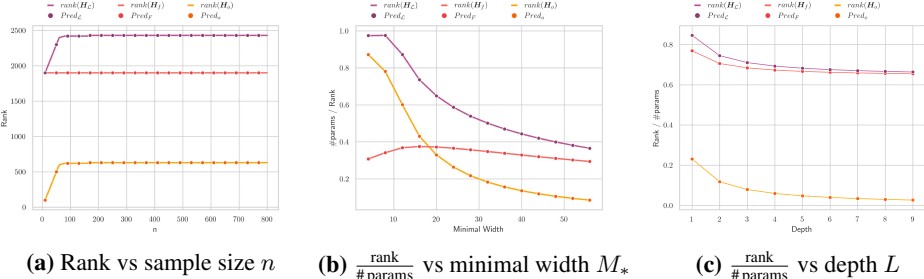

**(a)** Rank vs sample size $n$     **(b)** $\frac{\text{rank}}{\#\,\text{params}}$ vs minimal width $M_*$     **(c)** $\frac{\text{rank}}{\#\,\text{params}}$ vs depth $L$

**Figure S9:** Behaviour of rank and rank/#params on FashionMNIST using MSE, with hidden layers: $50, 20, 20, 20$ (Fig. S9a), $M_*, M_*$ (Fig. S9b) and $L$ layers of width $M = 25$ (Fig. S9c). The lines indicate the true value and circles denote our formula predictions.

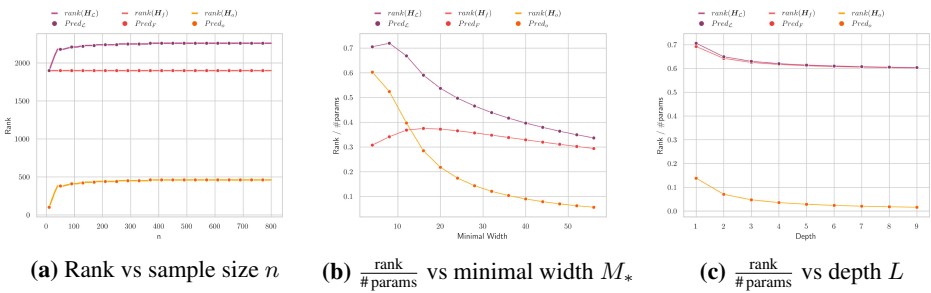

**(a)** Rank vs sample size $n$     **(b)** $\frac{\text{rank}}{\#\,\text{params}}$ vs minimal width $M_*$     **(c)** $\frac{\text{rank}}{\#\,\text{params}}$ vs depth $L$

**Figure S10:** Behaviour of rank and rank/#params on MNIST using MSE, with hidden layers: $50, 20, 20, 20$ (Fig. S10a), $M_*, M_*$ (Fig. S10b) and $L$ layers of width $M = 25$ (Fig. S10c). The lines indicate the true value and circles denote our formula predictions.

### S9.1.2   Cross Entropy

Here we consider another popular loss function, namely cross entropy, which is defined as

$$\ell_{\text{cp}}(\boldsymbol{\theta}) = -\sum_{i=1}^{N}\sum_{k=1}^{K} \log\left(\text{softmax}_k\left(F_{\boldsymbol{\theta}}(\mathbf{x}_i)\right) y_{ik}\right.$$

where $y_{ik} = \begin{cases} 1 & \text{if "$k$" is the label} \\ 0 & \text{otherwise} \end{cases}$   and $\text{softmax}_k(\boldsymbol{z}) = \frac{e^{z_k}}{\sum_{l=1}^{K} e^{z_l}}$. Observe that cross entropy is combined with a softmax operation at the output layer, constraining the final vector to sum to $1$, i.e. $\sum_{l=1}^{K} \text{softmax}_k(\boldsymbol{z}) = 1$. This induces, by construction a linear dependence at the output, thus instead of having $K$ free outputs, we only have $K-1$ independent outputs. We reflect this in our rank formulas by replacing every occurrence of $K$ by $K-1$.

Figure S11 shows the results for CIFAR10, Figure S12 for FashionMNIST and Figure S13 for MNIST. We observe a perfect match for all the datasets.

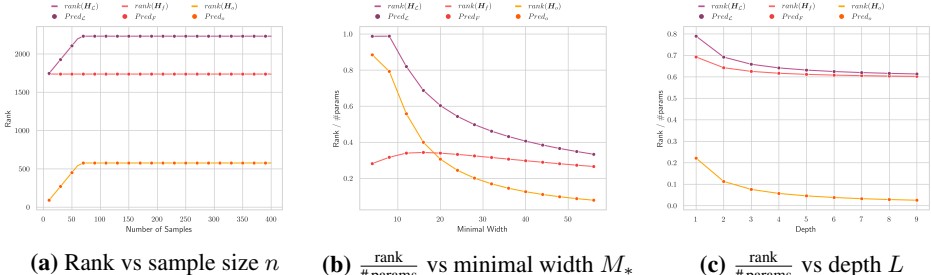

**(a)** Rank vs sample size $n$     **(b)** $\frac{\text{rank}}{\#\,\text{params}}$ vs minimal width $M_*$     **(c)** $\frac{\text{rank}}{\#\,\text{params}}$ vs depth $L$

**Figure S11:** Behaviour of rank and rank/#params on CIFAR10 using cross entropy, with hidden layers: $50, 20, 20, 20$ (Fig. S11a), $M_*, M_*$ (Fig. S11b) and $L$ layers of width $M = 25$ (Fig. S11c). The lines indicate the true value and circles denote our formula predictions.

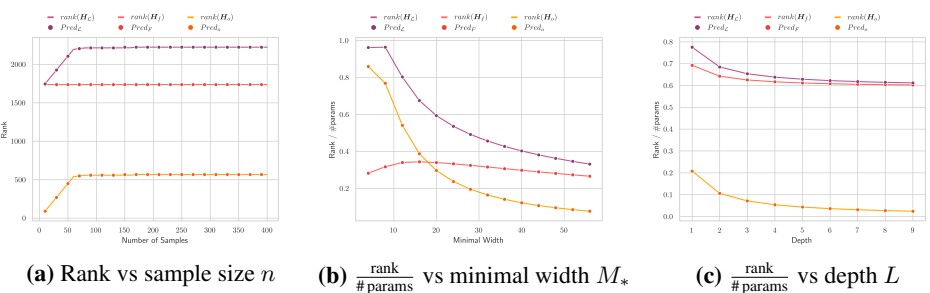

**(a)** Rank vs sample size $n$     **(b)** $\frac{\text{rank}}{\#\,\text{params}}$ vs minimal width $M_*$     **(c)** $\frac{\text{rank}}{\#\,\text{params}}$ vs depth $L$

**Figure S12:** Behaviour of rank and rank/#params on FashionMNIST using cross entropy, with hidden layers: $50, 20, 20, 20$ (Fig. S12a), $M_*, M_*$ (Fig. S12b) and $L$ layers of width $M = 25$ (Fig. S12c). The lines indicate the true value and circles denote our formula predictions.

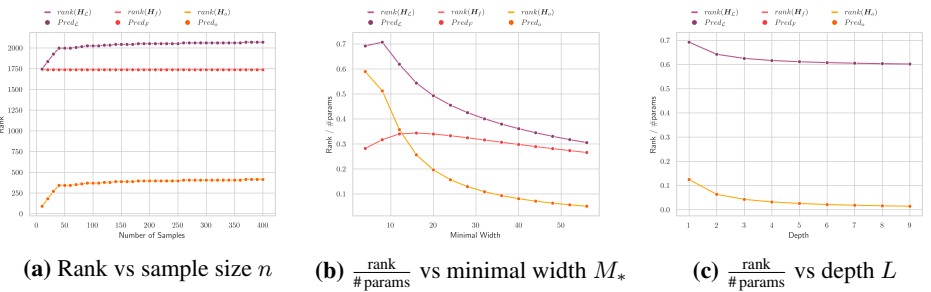

**(a)** Rank vs sample size $n$     **(b)** $\frac{\text{rank}}{\#\,\text{params}}$ vs minimal width $M_*$     **(c)** $\frac{\text{rank}}{\#\,\text{params}}$ vs depth $L$

**Figure S13:** Behaviour of rank and rank/#params on MNIST using cross entropy, with hidden layers: $50, 20, 20, 20$ (Fig. S13a), $M_*, M_*$ (Fig. S13b) and $L$ layers of width $M = 25$ (Fig. S13c). The lines indicate the true value and circles denote our formula predictions.

### S9.1.3   Cosh Loss

To highlight that our formulas are very robust to even more exotic loss functions, we consider the cosh-loss, defined as

$$\ell_{cosh}(\boldsymbol{\theta}) = \sum_{i=1}^{N} \sum_{k=1}^{K} \log\left(\cosh\left(\hat{y}_{ik} - y_{ik}\right)\right)$$

Figure S14 shows the results for CIFAR10, Figure S15 for FashionMNIST and Figure S16 for MNIST. We observe a perfect match for all the datasets. Also in this case we observe an exact match empirically for all datasets and varying sample size, width and depth.

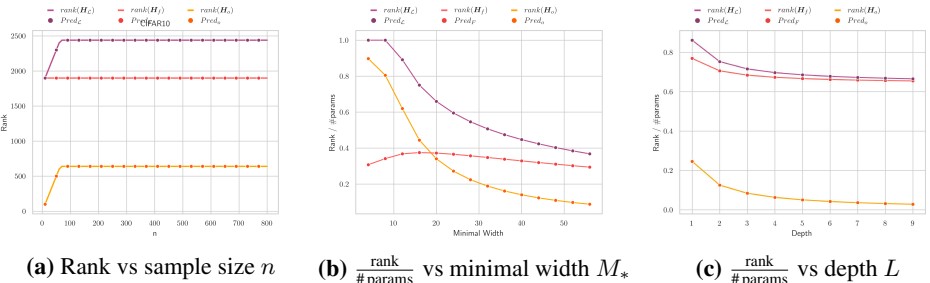

**(a)** Rank vs sample size $n$  **(b)** $\frac{\text{rank}}{\#\,\text{params}}$ vs minimal width $M_*$  **(c)** $\frac{\text{rank}}{\#\,\text{params}}$ vs depth $L$

**Figure S14:** Behaviour of rank and rank/#params on CIFAR10 using cosh loss, with hidden layers: $50, 20, 20, 20$ (Fig. S14a), $M_*, M_*$ (Fig. S14b) and $L$ layers of width $M = 25$ (Fig. S14c). The lines indicate the true value and circles denote our formula predictions.

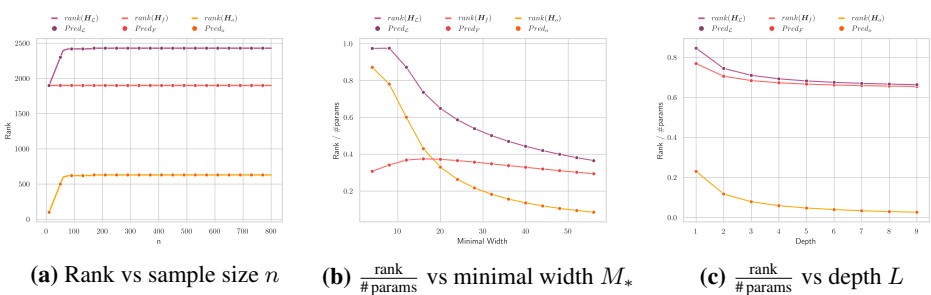

**(a)** Rank vs sample size $n$  **(b)** $\frac{\text{rank}}{\#\,\text{params}}$ vs minimal width $M_*$  **(c)** $\frac{\text{rank}}{\#\,\text{params}}$ vs depth $L$

**Figure S15:** Behaviour of rank and rank/#params on FashionMNIST using cosh loss, with hidden layers: $50, 20, 20, 20$ (Fig. S15a), $M_*, M_*$ (Fig. S15b) and $L$ layers of width $M = 25$ (Fig. S15c). The lines indicate the true value and circles denote our formula predictions.

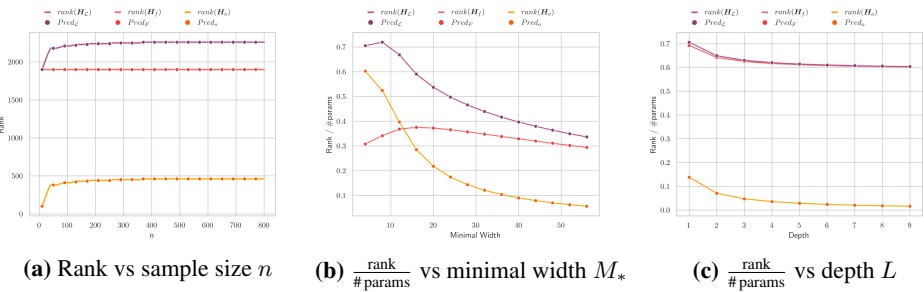

**(a)** Rank vs sample size $n$  **(b)** $\frac{\text{rank}}{\#\,\text{params}}$ vs minimal width $M_*$  **(c)** $\frac{\text{rank}}{\#\,\text{params}}$ vs depth $L$

**Figure S16:** Behaviour of rank and rank/#params on MNIST using cosh loss, with hidden layers: $50, 20, 20, 20$ (Fig. S16a), $M_*, M_*$ (Fig. S16b) and $L$ layers of width $M = 25$ (Fig. S16c). The lines indicate the true value and circles denote our formula predictions.

### S9.1.4 Different Initializations

Here we want to assess whether different initialization schemes can affect our rank predictions. Although our theoretical results suggest that our results hold for any initialization scheme that guarantees full-rank weight matrices, we perform an empirical study on CIFAR10 to check this. All the preceding experiments have used Gaussian initialization. Here we also check for uniform initialization, $W_{ij}^l \sim \mathcal{U}(-1, 1)$, and for orthogonal initialization. We display the results for uniform initialization in Figure S17 while Figure S18 shows the results for orthogonal initialization, for varying sample size, width and depth. As expected from our theoretical insights, we again observe exact matches with our predictions.

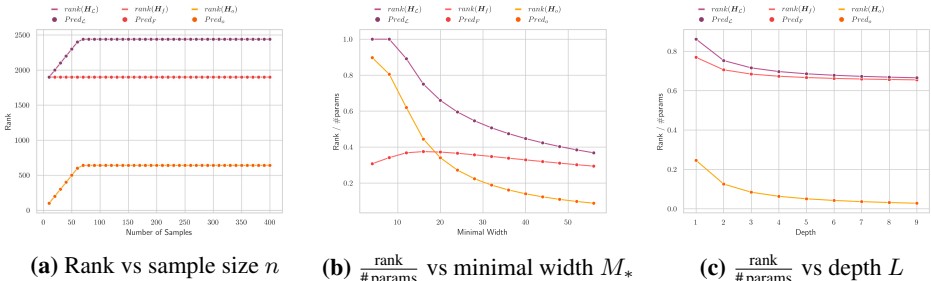

**(a)** Rank vs sample size $n$     **(b)** $\frac{\text{rank}}{\#\text{params}}$ vs minimal width $M_*$     **(c)** $\frac{\text{rank}}{\#\text{params}}$ vs depth $L$

**Figure S17:** Behaviour of rank and rank/#params on CIFAR10 using MSE and **uniform initialization** with hidden layers: $50, 20, 20, 20$ (Fig. S17a), $M_*, M_*$ (Fig. S17b) and $L$ layers of width $M = 25$ (Fig. S17c). The lines indicate the true value and circles denote our formula predictions.

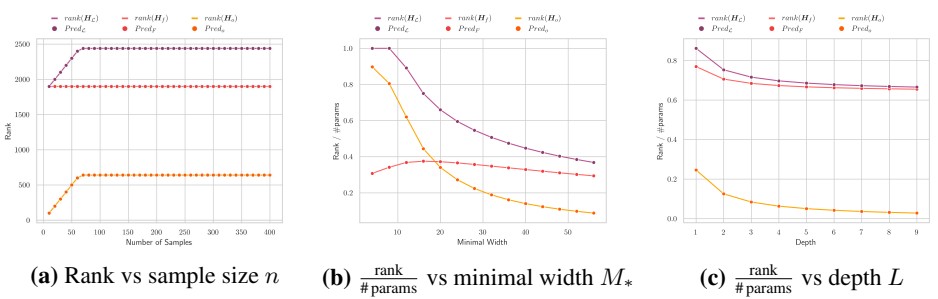

**(a)** Rank vs sample size $n$     **(b)** $\frac{\text{rank}}{\#\text{params}}$ vs minimal width $M_*$     **(c)** $\frac{\text{rank}}{\#\text{params}}$ vs depth $L$

**Figure S18:** Behaviour of rank and rank/#params on CIFAR10 using MSE and **orthogonal initialization** with hidden layers: $50, 20, 20, 20$ (Fig. S17a), $M_*, M_*$ (Fig. S17b) and $L$ layers of width $M = 25$ (Fig. S17c). The lines indicate the true value and circles denote our formula predictions.

#### S9.1.5 Rank Formulas With Bias

Here we verify the rank formulas derived for the case with bias in S7.3. We use MSE loss and CIFAR10 as the dataset. Again we see that the the rank predictions from our formulas exactly match the rank observed in practice.

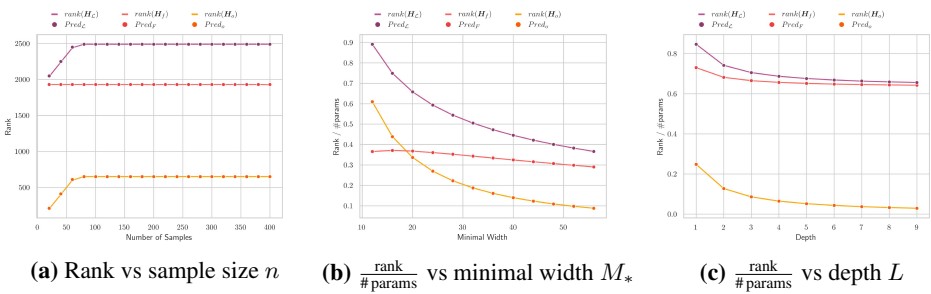

**(a)** Rank vs sample size $n$     **(b)** $\frac{\text{rank}}{\#\text{params}}$ vs minimal width $M_*$     **(c)** $\frac{\text{rank}}{\#\text{params}}$ vs depth $L$

**Figure S19:** Behaviour of rank and rank/#params on CIFAR10 using MSE and **bias** with hidden layers: $50, 20, 20, 20$ (Fig. S19a), $M_*, M_*$ (Fig. S19b) and $L$ layers of width $M = 25$ (Fig. S19c). The lines indicate the true value and circles denote our formula predictions.

### S9.2 Simulation of rank formulas for large settings

In the previous subsection, we have established how our formulas hold exactly in practice. An added benefit of these formulas is that they allow us to visualize how the rank will growth in relation to the number of parameters for bigger architectures — *without actually having to do the Hessian computations.* In Fig. S20 we show such a simulation for increasing width and depth. The simulations make the limiting behaviour of the fraction $\frac{\text{rank}}{\#params}$ even more apparent, as the fraction decreases with more and more overparametrization (in terms of both depth and width), until it reaches a threshold.

The left subfigure, which is the width-simulation plot, also shows an interesting behaviour. In the early phase, the outer-product Hessian $\mathbf{H}_o$ dominates the functional Hessian $\mathbf{H}_f$ in terms of $\frac{\text{rank}}{\#params}$, but after a certain width $\mathbf{H}_f$ starts to dominate $\mathbf{H}_o$ and continues to do so throughout. It would be of relevance for future work to further investigate the interaction between $\mathbf{H}_o$ and $\mathbf{H}_f$, and provide an understanding of these two phases.

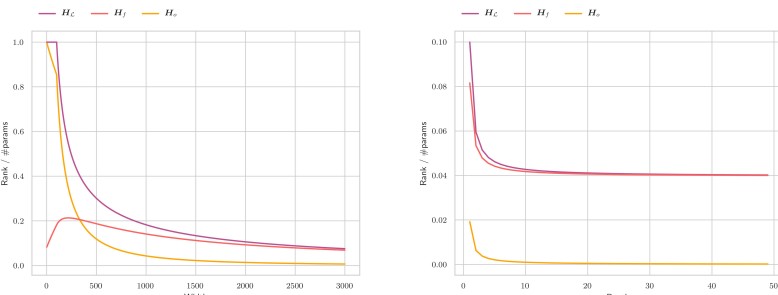

**Figure S20:** Simulating **rank/#params** over width (left) and depth (middle). For the width plot: $L = 4, r = 2352$, and for the depth plot: $r = 2352$ and hidden layer size 5000.

### S9.3    Reconstruction Error Plots for More Non-Linearities and Losses

To further illustrate how our predictions extend to different non-linearities, we repeat the reconstruction error experiment for different types of non-linearities and loss functions. In particular we study the activation functions

$$\sigma(x) = \text{ReLU}(x) \quad \sigma(x) = \tanh(x) \quad \sigma(x) = \text{ELU}(x) = \begin{cases} x & x > 0 \\ e^x - 1 & x \leq 0 \end{cases}$$

As before, we group the experiment by the loss function employed and vary the non-linearity used in each architecture. We test on this down-scaled MNIST with input dimensionality of $d = 64$ for the smaller architectures and $d = 49$ for the bigger ones. The number of samples is $N = 200$ across all settings.

### S9.3.1    Mean Squared Error

Here we expand on the Figure 4, using the same setting as presented in the main text but we consider more non-linearities. We display the results for ReLU in Figure S21, for ELU in Figure S22 and for tanh in Figure S23. We also consider slightly bigger architectures in Figures S24, S25 and S26, using the same ordering for the non-linearities as before. Again we observe that our rank prediction offers an excellent cut-off, allowing to preserve almost the entire structure of the Hessian, even for the bigger architectures. This is again strong evidence that our prediction captures the relevant eigenvalues but becomes distorted by smaller, irrelevant ones, inflating the exact rank.

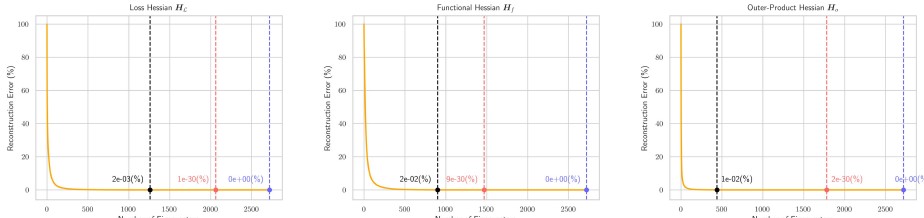

**Figure S21:** Hessian reconstruction error for **ReLU** under **MSE** as the rank of the approximation is increased. The x-axis represents the number of top eigenvectors that form the low-rank approximation. The $y$-axis displays the *reconstruction error in percentage* (100 % for zero eigenvectors used). The dashed vertical lines indicate the cut-off at various values of the rank: **first line** at the prediction based on the linear model, **second line** at the empirical measurement of rank, and **third line** based on upper bounds from [33], which become too coarse to be of any use (actually even greater than the # of parameters but not marked there for visualization purposes). The hidden layer sizes are $30, 20$.

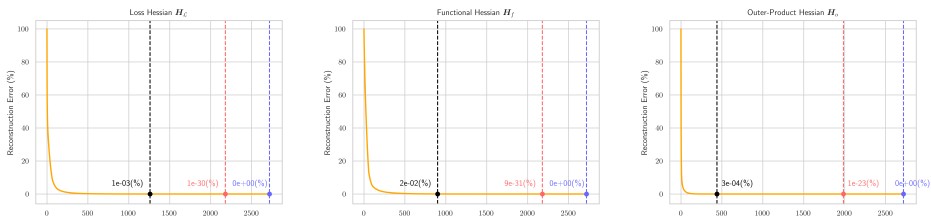

**Figure S22:** Hessian reconstruction error for **ELU** under **MSE** as the rank of the approximation is increased. The x-axis represents the number of top eigenvectors that form the low-rank approximation. The $y$-axis displays the *reconstruction error in percentage* (100 % for zero eigenvectors used). The dashed vertical lines indicate the cut-off at various values of the rank: **first line** at the prediction based on the linear model, **second line** at the empirical measurement of rank, and **third line** based on upper bounds from [33], which become too coarse to be of any use (actually even greater than the # of parameters but not marked there for visualization purposes). The hidden layer sizes are $30, 20$.

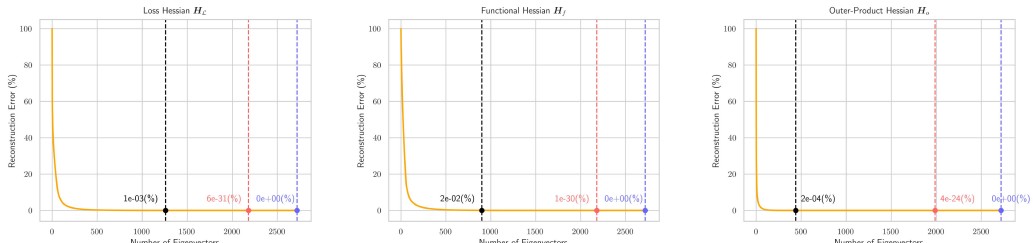

**Figure S23:** Hessian reconstruction error for **Tanh** under **MSE** as the rank of the approximation is increased. The x-axis represents the number of top eigenvectors that form the low-rank approximation. The $y$-axis displays the *reconstruction error in percentage* (100 % for zero eigenvectors used). The dashed vertical lines indicate the cut-off at various values of the rank: **first line** at the prediction based on the linear model, **second line** at the empirical measurement of rank, and **third line** based on upper bounds from [33], which become too coarse to be of any use (actually even greater than the # of parameters but not marked there for visualization purposes). The hidden layer sizes are $30, 20$.

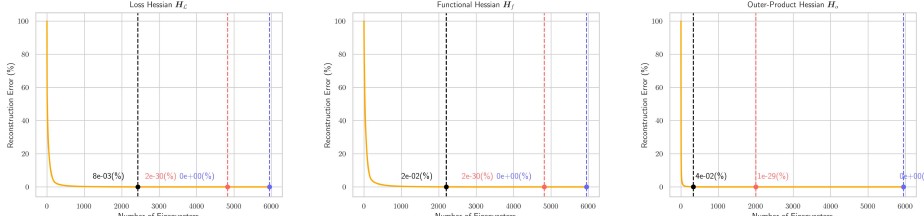

**Figure S24:** Hessian reconstruction error for **ReLU** under **MSE** as the rank of the approximation is increased. The x-axis represents the number of top eigenvectors that form the low-rank approximation. The $y$-axis displays the *reconstruction error in percentage* (100 % for zero eigenvectors used). The dashed vertical lines indicate the cut-off at various values of the rank: **first line** at the prediction based on the linear model, **second line** at the empirical measurement of rank, and **third line** based on upper bounds from [33], which become too coarse to be of any use (actually even greater than the # of parameters but not marked there for visualization purposes). The hidden layer sizes are $50, 40, 30$.

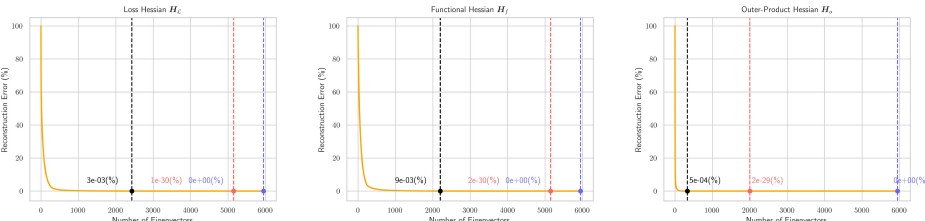

**Figure S25:** Hessian reconstruction error for **ELU** under **MSE** as the rank of the approximation is increased. The x-axis represents the number of top eigenvectors that form the low-rank approximation. The $y$-axis displays the *reconstruction error in percentage* (100 % for zero eigenvectors used). The dashed vertical lines indicate the cut-off at various values of the rank: **first line** at the prediction based on the linear model, **second line** at the empirical measurement of rank, and **third line** based on upper bounds from [33], which become too coarse to be of any use (actually even greater than the # of parameters but not marked there for visualization purposes). The hidden layer sizes are $50, 40, 30$.

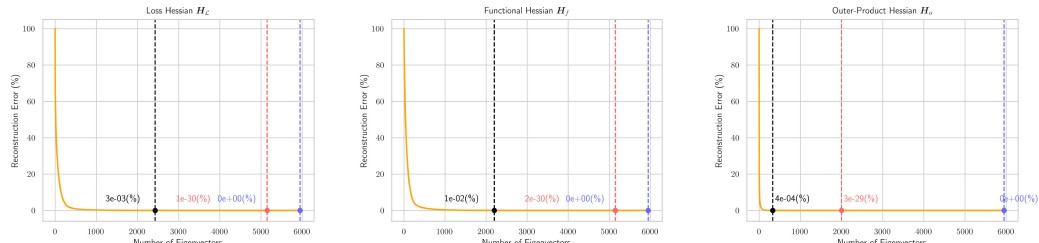

**Figure S26:** Hessian reconstruction error for **Tanh** under **MSE** as the rank of the approximation is increased. The x-axis represents the number of top eigenvectors that form the low-rank approximation. The $y$-axis displays the *reconstruction error in percentage* (100 % for zero eigenvectors used). The dashed vertical lines indicate the cut-off at various values of the rank: **first line** at the prediction based on the linear model, **second line** at the empirical measurement of rank, and **third line** based on upper bounds from [33], which become too coarse to be of any use (actually even greater than the # of parameters but not marked there for visualization purposes). The hidden layer sizes are $50, 40, 30$.

### S9.3.2 Cross Entropy

Here we repeat the same experiments for cross entropy loss. We use the adjusted formula for the linear rank predictions, i.e. we replace $K$ by $K - 1$. We display the results for ReLU in Figure S27, for ELU in Figure S28 and for tanh in Figure S29. We also obtain excellent approximations for the numerical rank in this setting, showing that our predictions also extend to other losses under non-linearities.

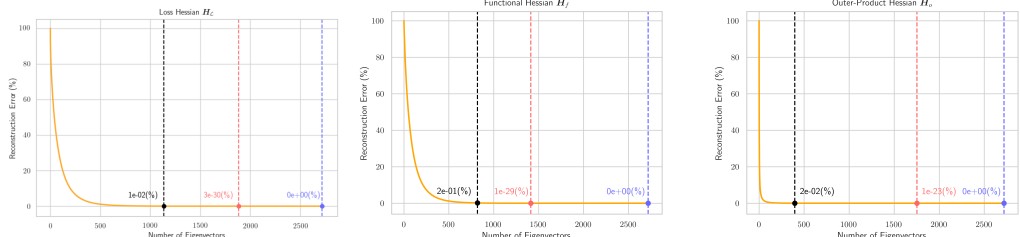

**Figure S27:** Hessian reconstruction error for **ReLU** as the rank of the approximation is increased under **cross entropy loss**. The x-axis represents the number of top eigenvectors that form the low-rank approximation. The $y$-axis displays the *reconstruction error in percentage* (100 % for zero eigenvectors used). The dashed vertical lines indicate the cut-off at various values of the rank: **first line** at the prediction based on the linear model, **second line** at the empirical measurement of rank, and **third line** based on upper bounds from [33], which become too coarse to be of any use (actually even greater than the # of parameters but not marked there for visualization purposes). The hidden layer sizes are 30, 20.

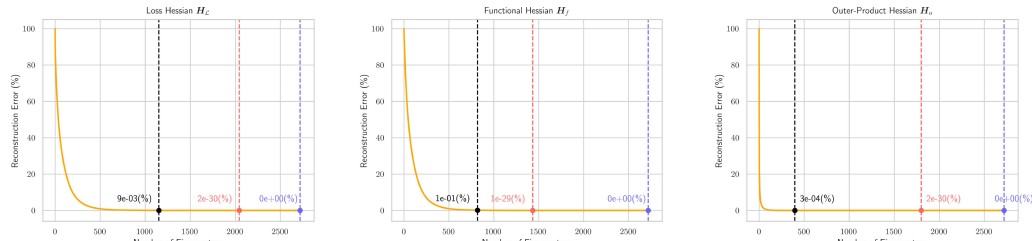

**Figure S28:** Hessian reconstruction error for **ELU** as the rank of the approximation is increased under **cross entropy loss**. The x-axis represents the number of top eigenvectors that form the low-rank approximation. The $y$-axis displays the *reconstruction error in percentage* (100 % for zero eigenvectors used). The dashed vertical lines indicate the cut-off at various values of the rank: **first line** at the prediction based on the linear model, **second line** at the empirical measurement of rank, and **third line** based on upper bounds from [33], which become too coarse to be of any use (actually even greater than the # of parameters but not marked there for visualization purposes). The hidden layer sizes are 30, 20.

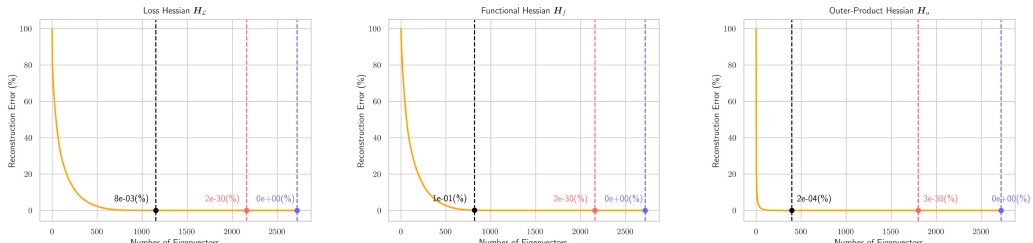

**Figure S29:** Hessian reconstruction error for **Tanh** as the rank of the approximation is increased under **cross entropy loss**. The x-axis represents the number of top eigenvectors that form the low-rank approximation. The $y$-axis displays the *reconstruction error in percentage* (100 % for zero eigenvectors used). The dashed vertical lines indicate the cut-off at various values of the rank: **first line** at the prediction based on the linear model, **second line** at the empirical measurement of rank, and **third line** based on upper bounds from [33], which become too coarse to be of any use (actually even greater than the # of parameters but not marked there for visualization purposes). The hidden layer sizes are 30, 20.

### S9.4 Spectral Plots for More Non-Linearities

To further underline the utility of our theoretical results in the non-linear setting, we present more spectral plots, super-imposing the linear and corresponding non-linear spectrum for more non-linearities and loss functions. For all experiments we use $d = 64$ and $N = 200$.

### S9.4.1 Mean Squared Error

In Figure S30, we show the results for ReLU non-linearity, observing that the plateau of the spectrum is accurately described by our predictions. The same also holds for ELU activation, as can be readily seen in Figure S31.

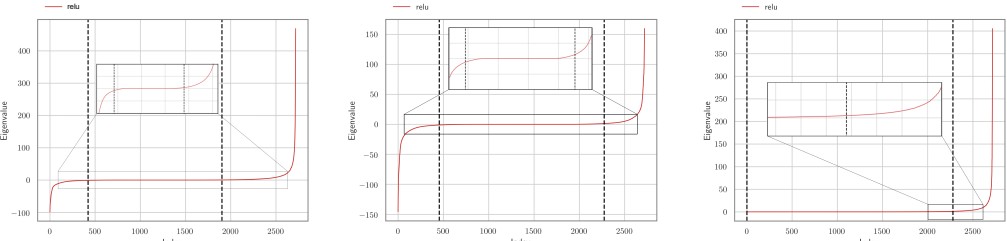

**Figure S30:** Spectrum of the loss Hessian $\mathbf{H}_{\mathcal{L}}$ (left), functional Hessian $\mathbf{H}_f$ (middle) and outer product $\mathbf{H}_o$ (right), for a **ReLU** network. Black dashed lines are the predictions of the bulk size via our rank formulas. We use 2 hidden layers of size $30, 20$ on MNIST.

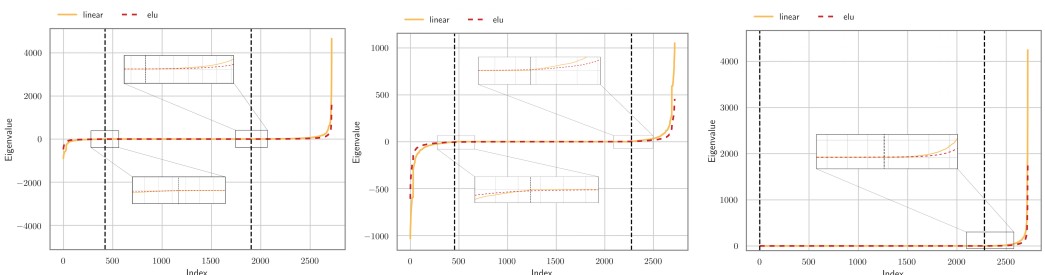

**Figure S31:** Spectrum of the loss Hessian $\mathbf{H}_{\mathcal{L}}$ (left), functional Hessian $\mathbf{H}_f$ (middle) and outer product $\mathbf{H}_o$ (right), for **linear** and **non-linear** networks. Black dashed lines are the predictions of the bulk size via our rank formulas. We use 2 hidden layers of size $30, 20$ with **ELU** activation on MNIST.

### S9.4.2 Cross Entropy

Here we show that the spectrum also behaves very similar if cross entropy is employed, again regardless of the non-linearity used. We show the results for ReLU non-linearity in Figure S32, for tanh in Figure S33 and ELU in Figure S34.

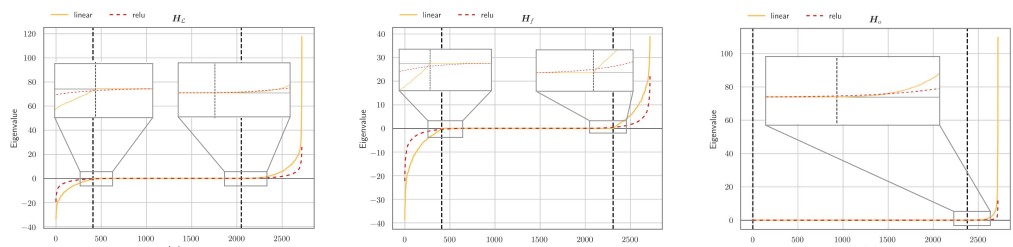

**Figure S32:** Spectrum of the loss Hessian $\mathbf{H}_{\mathcal{L}}$ (left), functional Hessian $\mathbf{H}_f$ (middle) and outer product $\mathbf{H}_o$ (right), for **linear** and **non-linear** networks. Black dashed lines are the predictions of the bulk size via our rank formulas. We use 2 hidden layers of size $30, 20$ with **ReLU** activation on MNIST under **cross entropy** loss.

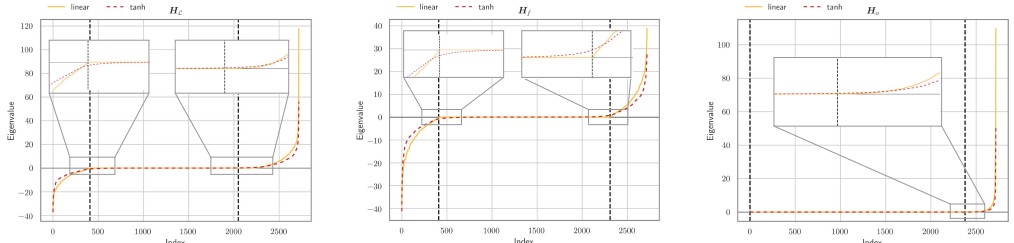

**Figure S33:** Spectrum of the loss Hessian $\mathbf{H}_{\mathcal{L}}$ (left), functional Hessian $\mathbf{H}_f$ (middle) and outer product $\mathbf{H}_o$ (right), for **linear** and **non-linear** networks. Black dashed lines are the predictions of the bulk size via our rank formulas. We use 2 hidden layers of size $30, 20$ with **tanh** activation on MNIST under **cross entropy** loss.

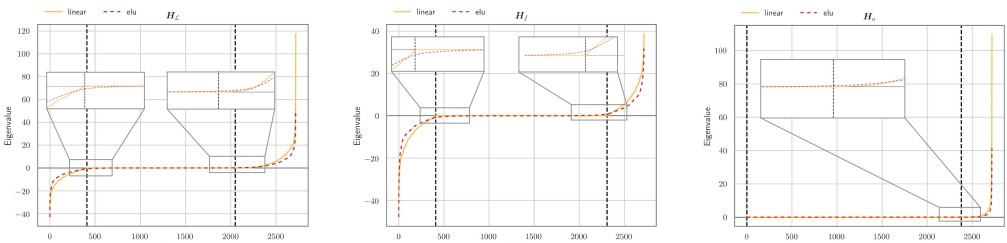

**Figure S34:** Spectrum of the loss Hessian $\mathbf{H}_{\mathcal{L}}$ (left), functional Hessian $\mathbf{H}_f$ (middle) and outer product $\mathbf{H}_o$ (right), for **linear** and **non-linear** networks. Black dashed lines are the predictions of the bulk size via our rank formulas. We use 2 hidden layers of size $30, 20$ with **ELU** activation on MNIST under **cross entropy** loss.

## S9.5   Rank Results for Neural Tangent Kernel

In this section we show that our formulas also allow for insights into rank of the Gram matrix induced by the neural tangent kernel (NTK) [44] at initialization. The NTK is a matrix defined entry-wise as

$$\hat{\Sigma}_{ij} = \left(\nabla_{\boldsymbol{\theta}} F_{\boldsymbol{\theta}}(\mathbf{x}_i)\right)^{\top} \nabla_{\boldsymbol{\theta}} F_{\boldsymbol{\theta}}(\mathbf{x}_j)$$

In Figure S35 we display the rank dynamics of the Gram matrix as a function of sample size. We use the predictions based on the outer-product Hessian. We down-scale to $d = 64$ and use hidden layer sizes $20, 20$ and use $K = 1$ classes. We observe an exact match for all datasets and sample sizes.

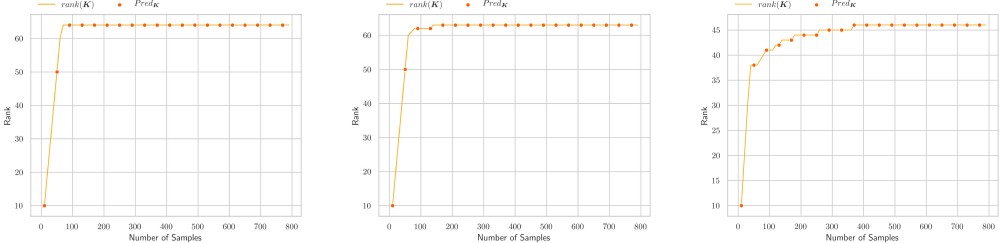

**Figure S35:** Rank of the empirical NTK versus sample size $n$ for architecture $20, 20$. We display the predictions based on the outer-product $\mathbf{H}_o$ as dots, using CIFAR10 (left), FashionMNIST (middle) and MNIST (right).