# OpenReview forum: "Analytic Insights into Structure and Rank of Neural Network Hessian Maps"
_NeurIPS.cc/2021/Conference — NeurIPS 2021 Poster_

### Official Review · Reviewer_bE99 · 2021-07-08

**Rating:** 5
**Confidence:** 4

**Summary:**

This paper analyzes the rank of the Hessian matrix of fully connected neural networks. Theoretically, for a linear fully-connected neural network with square loss, the authors provided an upper bound for the rank of Hessian with respect to the population loss. They also did many experiments for small neural networks on MNIST and CIFAR-10 to show that their upper bound is almost tight, and works for different types of non-linear activations, losses, and initializations.

**Limitations And Societal Impact:**

The authors did not seem to explicitly discuss the limitations or societal impact in their paper. I do not immediately see any potential negative societal impacts of this work, but it might be better if the authors could talk about the limitations of their work, perhaps in the discussion section.

**Main Review:**

Significance:

It is interesting to study the actual rank of the neural network Hessian matrix. However, my major concern about this is that the justification for studying the rank itself may be a bit unclear. The exact rank of such a huge matrix is not robust to noise and doesn't necessarily carry much information. I will explain this in more detail in the following paragraphs:

- People have empirically found that the neural network Hessian eigenspectrum usually contains a "spike" which has very few large eigenvalues and a "bulk" with a lot of small eigenvalues. In this case, the rank is approximately the size of the bulk, so it loses the information about the spike, which can be significant in capturing gradient change, measuring flatness, etc. In Section 5.2, line 252, the authors also claimed that "tiny but non-zero eigenvalues hold little significance" (similar statements at line 306), but they do not eliminate these eigenvalues for the linear case, which seems confusing.

- The exact rank of a huge matrix is not robust to noise, and noise always exists in the case of neural network training. For instance, the input data may contain some noise, so the second-moment matrix $\Sigma_{xx}$ of the input can become full rank with a small amount of noise even when the "true" input only lies in a low-rank subspace. Noises can also come from the finite sampling of the training data, the selection of mini-batches, and even the machine computation. All these factors might potentially inject noise into the Hessian matrix and make its rank change significantly. The authors do not seem to discuss the influences of these noises on their results, which makes their results somewhat restrictive.

- Given the two reasons mentioned above, it might be a bit unclear how the rank of Hessian could measure the effective number of parameters or inherent complexity of neural networks, both of which should intuitively be robust to the noises mentioned above.

- The implication from the rank to the structure of the Hessian might be a bit weak. The authors in this paper showed empirically that the ranges of different parts of Hessian rarely intersect, but the corresponding eigenvalues of these spaces are not analyzed. It's possible that they are random directions in high-dimensional space and parts of the bulk in Hessian eigenspectrum, but it's also possible that some directions are part of the spike but others are bulk.

Theory:

- The setting of the theoretical part in this paper is somewhat limited. The analysis only works for a linear fully-connected neural network with square loss and requires the weight matrices to be full rank (which is true at initialization but not necessarily true during training or at minima).

- The techniques used by the authors are specific, e.g., they seem to rely heavily on the rank formula for the "Z-like" matrices (Lemma 1). These techniques seem hard to generalize to other settings, which may limit the generalization of their results to other scenarios.

Experiments:

- The experimental settings are kind of restricted. The experiments are done on (Fashion-)MNIST and CIFAR-10 with small neural networks (number of parameters at most a thousand or so), and the input images are downscaled. These restrictions make the experimental setting not very realistic and the empirical results not so convincing.

- The description of how the authors calculate the exact rank of the full Hessian might be a little vague. Since the eigenspectrum is not well-separated, it's unclear how the authors choose the threshold to decide whether a singular value is 0 or not and how to justify this procedure. It might be better if the authors could talk more about how they "utilize FLOAT-64 precision" and handle the numerical errors when computing the Hessian and the corresponding SVD.

Clarity:

- Generally, this paper is well-structured. However, the figures in this paper are small, especially the legend. It's hard to read the legend on a printed copy of this paper. Besides, the figures are not very friendly to black and white printers.

- Some statements and terms in this paper might be a bit vague. For instance, at line 148, the notation $(\Sigma_{xx})_{r\times r}$ is not defined. More details about these potentially vague statements/terms will be provided in the following "Minor Comments" section.

Minor Comments:

- There might be a mismatch between the settings for the theory and experiments in the paper about the number of parameters. For the theoretical part, as also mentioned at the end of the abstract, the rank deficiency and parameter redundancy are more significant for over-parameterized networks, but the experiments were done on under-parameterized networks.

- The Hessian matrix of neural networks depends on the input data and their labels. The arguments in this paper seem to work well at the initialization and the global minimum where the Hessian is (almost, in some sense) independent of the labels. However, it is not very clear whether they still work during training since the functional Hessian may have some non-trivial correlations with the outer-product Hessian.

- In line 216, the authors provided an interpretation to their formula, i.e., they construct a hypothetical network whose width for all layers are subtracted by the minimum width, but did not explain how this hypothetical network relates to the original network. It may be better if the authors could provide more justifications on this.

- In the caption of Figure 3, the authors use "predictions of the bulk size", but I did not find where this is defined. It might be better if the authors could explain this more clearly.

- At line 240, the authors claimed that rank/#params "intuitively captures the fraction of effective parameters", which might be confusing because rank is some (potentially non-robust) second-order information about the parameters and not very directly related to the parameters themselves.

- In Figure 4, the authors showed that the linear rank value prediction provides a reconstruction of the Hessian with only 0.002% error. However, since there is no significant eigengap at or near that linear rank value, it is possible that using a much smaller value, e.g., 200, will also result in a good reconstruction of the Hessian. Therefore, it is a little unclear why the linear rank value is significant in this case.

- At line 317 (Corollary 10), the "minimum" might not be formally defined. Does it mean "global minimum with respect to the population loss"? I think this corollary may need the rank of functional Hessian to be of rank 0.

Related Works:

- The related works are generally properly cited. There are some other papers that might be relevant to this paper. [1] provided a theoretical model for the Hessian of neural network and use that to explain the existence of C large eigenvalues of Hessian where C is the number of classes, and [2] proposes "class/cross-class structure" and analyzes the Hessian eigenspectrum in a more fine-grained way.

[1] Fort, Stanislav, and Surya Ganguli. "Emergent properties of the local geometry of neural loss landscapes." arXiv preprint arXiv:1910.05929 (2019).
[2] Papyan, Vardan. "Traces of class/cross-class structure pervade deep learning spectra." Journal of Machine Learning Research 21.252 (2020): 1-64.

Typos:

- Line 214: "Fact. (7)" -> "Fact 7"
- Line 165: In Line 147 the authors already assume $\Sigma_{xx}$ is an $r\times r$ matrix, but here it's still $d\times d$. This abuse of notation may confuse the readers.

---------------------Update--------------------------

I have read all the other reviews and the authors' responses, and I have decided to increase my score by 1. The detailed reasons why I increase my score is attached below:

- Thank the authors for their very detailed responses. I understand from their response that the rank of Hessian is more robust to noise than I originally thought, so it makes more sense to consider this rank as the inherent complexity measure of neural networks.

- It is interesting to see from the "Validity of rank results during training" section in their general response that their results hold during training and might empirically become tighter throughout training. This addressed one of my minor concerns which is about the interactions between functional Hessian and outer-product Hessian.

- It is still a bit unclear how the Hessian rank can imply the effective number of parameters in neural networks. The authors take the quadratic problem as an example, but the quadratic approximations for neural networks may not be very accurate since the optimization landscapes of neural networks are always highly non-convex and non-smooth.

- The authors computed the "energy" of the spike and the bulk for a small network with K=7, and these 7 large eigenvalues contribute even more energy than the sum of energy contributed by hundreds of bulk eigenvalues. Therefore, the importance of the "bulk" of the Hessian eigenspectrum may still be limited.

**Time Spent Reviewing:**

12

---

> ### Author Response · Authors · 2021-08-10
> **Detailed response (1/3)**
>
> Thank you so much for spending a significant amount of time reviewing the paper as well as sharing comprehensive suggestions and feedback.
>
> $\quad$
>
> ### Significance
>
> $\quad$
>
> #### **Spikes, Bulk, and the long tail of eigenvalues**:
>
> 1. *Eigenvalue spikes are important, but that does not imply bulk can be ignored:*  An important thing is to consider the spectrum “energy” captured by a particular set of eigenvalues relative to the entire spectrum. Let us illustrate our point with the help of an example. Consider a (ReLU) network of shape 5, 20, 20, 7 which has a total of 640 parameters. As per the literature there are K = # of classes/targets many spikes (here K=7) large eigenvalues. Now, let us compare the spectral energy captured by these K spikes, i.e., sum of the squared eigenvalues corresponding to the spikes, compared to the rest of the spectrum.
> | Overall spectrum energy $\quad$  |   % of energy captured by spikes $\quad$   |   % of energy captured by bulk |
> |-------------------------|-----------------------------------------|---------------------------------------|
> | 182.83                  | 51.17 %                                 | 48.83 %                               |
>
>    As we can see from the Table above, the spikes only capture 51.17% of the spectrum, while the bulk captures 48.83% of the spectrum energy. *Hence, looking at the spikes alone will also lose valuable information about the Hessian.*
>
>
>
> 2. *Bulk also provides indispensable information in regards to optimization:*  In fact, the bulk of the spectrum is precisely the part that possesses **negative eigenvalues**, which are immensely crucial even for first-order optimization methods --- as these denote the directions which will result in a decrease in the loss. Therefore, knowledge of the bulk size (which is approximately provided by rank, as rightly remarked by you) can provide a useful guideline for which network architectures are more amenable to optimization.
>
>
> 3. *Confusion about “Tiny but non-zero eigenvalues hold little significance”:* We understand what is causing the confusion and we should have been more clear about this aspect. Essentially, the reason is that there is a **marked difference in ReLU vs Linear spectrums with respect to the long tail of eigenvalues**. But, let us illustrate this subtle point by considering the same network as above: 5, 20, 20, 7 with 640 parameters. At this Float64 precision with NumPy, the linear network has rank 385, while for ReLU the rank comes out to be 636 (for details on the calculation of rank and corresponding tolerance, please check out the paragraph titled “Calculation of exact rank at float64 precision” below)
> | Network Type   | # parameters | Rank (Float64 precision, NumPy) | Numerical rank (tolerance = 1e-2) | % of spectrum energy captured at numerical rank | Max eigenvalue |
> |----------------|--------------|--------------------------|-----------------------------------|-------------------------------------------------|----------------|
> | Linear network | 640          | 385                      | 380                               | 99.99 %                                         | 14.60          |
> | ReLU network   | 640          | 636                      | 448                               | 99.99 %                                         | 5.73           |
>
>     Assume, for the purpose of numerical rank, that we set a threshold of 1e-2, i.e., we treat the eigenvalues with magnitude smaller than 1e-2 to be zero. This results in the numerical rank of 380 in the linear case and 448 in the case of ReLU. The % of spectrum energy captured at this numerical rank is essentially the same (99.99%) for both. However, notice that while the numerical rank in the linear case is very close to its empirical rank of 385, but for ReLU, we find that the empirically measured rank is inflated by about 200.
>
>    This happens because in the case of ReLU there are about 200 eigenvalues, with magnitudes in the order of 1e-3, 1e-4, 1e-5, 1e-6. **These are precisely the “tiny, but non-zero eigenvalues” which inflate the rank** --- as the % of energy captured is basically the same as in the linear case. This fact can also be realized from the figures, as the linear case spectrums hit zero very sharply in comparison to the non-linear spectrums.
>
>     (Note, the maximum eigenvalue is of the same order in both the cases, so there are no other effects at play here.)
>
> $\quad$
>
> #### **“Not robust to noise”:**
>
> Thanks for mentioning this important point, which must be clarified to avoid potential misunderstanding. We agree that the inherent complexity of neural networks should be robust to noise, and our rank-based complexity measure IS indeed robust, as detailed below:
>
>  1. First of all, we encourage you to check our general comment titled, “Rank as effective # params”. This should also clarify your remark that rank is “not very directly related to the parameters themselves”.
>
> 2. Our primary interest is to investigate the effect of the network architecture itself, not the training dynamics. Thereby, our focus is on the **Hessian rank at initialization**, which consequently is devoid of the sources of noise during training, such as mini-batches, and is thus untouched by them.
>
> 3. Hessian rank is also unaffected by “noise” due to sampling of training data. In Fig 2a, we have already established that the Hessian rank formulas match with that observed empirically, when computed over any arbitrary sized subset of the dataset. Hence, the rank results are completely stable with respect to this aspect. (But obviously, to get the “right picture” about the Hessian rank and formulas, one must sample in an i.i.d. manner at least  input dimension ‘d’ many samples.)
>
> 4. Yes, it is possible that if the underlying data covariance matrix is noisy, this will lead to an increased value of ‘r’. But, this is **merely a constant additive factor** in the Hessian rank,  since the dominant term in the Hessian rank is the term ‘qM’, where M is the total number of hidden neurons, and q=minimum dimension = # classes K (for most practically used networks).
>
> 5. During training our rank bounds hold in terms of rank of weight matrices (see the comment in general response “Validity of rank results during training” for more) and rank of Omega at the particular point in training.
>
> 6. We are extremely attentive to the matter of noise arising from machine computation, which is why we compute rank in Float64 precision and compute the SVD without any approximations, even though this results in higher memory and computation costs. **As a result, there is minimal or almost no error due to this.**
>
>
> As a result, we are confident that the rank bounds and formulae do not really suffer from the mentioned concerns! *At the same time, the biggest advantage of rank is that it provides an interpretable and precise ballpark to the inherent complexity* --- as evident from the formulas. And primarily, **this is thanks to its particular discrete nature.** Other complexity measures such as weight norms, compression bounds, Lipschitz constants, etc., do not provide any such interpretable ballpark onto the model complexity.
>
> (Besides, also check out the section “Numerical Rank and relaxations” for discussion on additional relaxations of rank.)
>
> $\quad$
>
> #### **"Implication from the rank to the structure of the Hessian might be a bit weak"**
> > “It's possible that they are random directions in high-dimensional space and parts of the bulk in Hessian eigenspectrum, but it's also possible that some directions are part of the spike but others are bulk.”:
>
>  This is a very interesting question regarding which directions are present in the intersection of the outer-product and functional Hessian. Note, the dimension of intersection of their column spaces is $2qs -s^2$, which in the typical case of $q=s=K$ ($K$ being the # of classes), comes out to $K^2$. Also, the number of spikes in the eigenspectrum, reported in the literature, are equal to $K$. So it is likely that the majority of the directions in their intersection will be in the bulk. But offering a precise answer is unfortunately beyond the current scope.
>
> Also, we would like to remark that we in fact do some analysis of the eigenvalues, for instance, check out section “S9 Properties of the Hessian Spectrum” --- we discover that the spectrum can contain plateaus of repeated eigenvalues.

---

> ### Author Response · Authors · 2021-08-10
> **Detailed response (2/3)**
>
> ### Theory:
>
> $\quad$
>
> > (i) “Setting is limited”:
>
> $\quad$
>
> **A. Linear fully connected networks:**
>  1. *Fully-connected networks (FCNs):* While FCNs are not the networks one would use to obtain state-of-the-art performance, they are pretty much the **bread-and-butter in deep learning theory**. The reason is that they capture most of the salient aspects of deep learning, such as: highly non-convex objective yet optimizable by gradient descent, over-parameterized regime, the benefit of depth. As a result, most theoretical work, be it understanding implicit bias of gradient descent, over-parameterization, critical points, etc.,  are regularly done for fully-connected networks. Without working out a theory for this fundamental network type, a general theory is hard to come by.
>
>
>  2. *Non-linearities being further restrictions:* The challenge in working with non-linearities is that they result in a significantly higher dependence on the data distribution --- in contrast to the linear case where the data dependence manifests itself only through the input covariance matrix. As a result, theoretical statements are hard to make in general. *In fact, any result in the non-linear case will further delimit the setting due to additional assumptions. Plus, most analyses in the literature work only up to one-hidden layer.*
>
>
>  3. *Lazy/NTK regime of infinite width doesn't benefit here:* Most theoretical works which go beyond the linear fully connected network setting, primarily investigate the infinite-width regime. First of all, this regime does not capture the behaviour of finite-sized networks as well as fails to be at par in terms of empirical performance. Next, for the purposes of analyzing Hessian rank, the infinite-width regime is essentially futile because we are left only with a linearization of the network function, which will remove the functional Hessian part entirely. The outer-product Hessian cannot be studied because it will not be a finite-sized matrix, and one has to contend with the NTK matrix. But, for that we anyways already provide theoretical statements in the finite-width regime, both in the linear and non-linear case (see Theorems 3 and 9).
>
> Hence, in this sense, deep linear networks provide a suitable avenue to gain a broad understanding of neural networks. *In fact, for some phenomena, it might even be advantageous to study the linear case rather than the non-linear. This is highlighted by the pessimistic bound delivered on the rank in the non-linear case, despite the empirical results on numerical rank in this scenario.*
>
> $\quad$
>
> **B. MSE loss:**
>
> 1)   MSE loss just simplifies the analysis but the proofs can be easily extended to other cases. For e.g., in the case of the outer-product Hessian with Cross-Entropy (CE) loss, one has to realize there will be another matrix, which is the Hessian of the loss with respect to the network output. This is equal to $\operatorname{diag}(p) - p p^\top$, where $p$ denotes the vector of class softmax, and is of rank $K-1$ (this can be seen summing all the columns). Hence, after $K$ is replaced by $K-1$ in the formulas, they match the empirical observations.
>
>  2) See figure S9, S10 and S11 for the results on cross entropy. (Also, we present results for cosh loss in figures S12, S13 and S14, which shows our results also generalize to other losses beside the commonly used MSE and CE.)
>
> $\quad$
>
> **C. Maximal rank of weight-matrices:**
>
> 1) We should have emphasized this aspect better in our presentation. Section 6 establishes theoretically as well as empirically that the rank of weight matrices remain the same as at initialization. So our Hessian rank bound still remains a valid upper bound. But even otherwise, say if the rank of the weight matrices were to change in some setting, our upper bounds can be applied by plugging in the particular value of rank instead of the maximal rank value *(see the general response “Validity of rank results during training” for more)*.
>
>
> $\quad$
>
> >  (ii) “techniques seem hard to generalize to other settings”:
>
> $\quad$
>
> We respectfully disagree with this claim.
>
> 1) It is true that our analysis in the linear case is heavily realized on Z-like matrices, but this is just the kind of structure in the Hessian of linear fully connected networks. More fundamentally, underlying this lemma and others is the use of pseudo-inverse and oblique projectors for Kronecker products (which are always present in the Hessian structure irrespective of the network type).
>
>
> 2) If you think about the convolutional case, the matrices involved will have additional special structure, e.g, (linear) convolution can be expressed as matrix multiplication by a Toeplitz matrix, and so it will surely need related tools. *However, it is extremely unlikely that our introduced tools will become useless abruptly, since the class of fully-connected networks contains as a subset the class of convolutional networks.*
>
>
> 3) Anyhow, **the evidence for this very statement can be even seen in our proof of Theorem 9** which deals with non-linear activation functions, where **the essence was to use the Z-like matrices**. The same applies to the case where we extend our results to the case of bias, Theorem 12.
>
>
>
> ---
>
> $\quad$
>
> ### Experiments:
>
> $\quad$
>
> **“Restricted settings”:** Yes, we are well aware that the network sizes are fairly (and purposefully) small.
>
> 1. Firstly, let us simply take a look at the computational challenges involved. But before, let us emphasize once again that the computation of rank must be done by using full SVD in Float64 precision, otherwise the empirical results will not rigorously characterize the ‘true’ rank. So, the complexity of running SVD is $\mathcal{O}(p^3)$ where $p$ is the # of parameters. Then multiply with it, the number of samples which further increases the complexity by 2 orders of magnitude. Alongside, the memory costs further swell up due to using Float64 precision. Thus, it should not be any surprise that we are forced to use fairly small networks.
>
>
>
> 2. > “number of parameters at most a thousand”:
>
>    No this is not the case. We do have larger networks with around 6000 or so parameters (see Figures 2a, 2b, S) --- by which it reaches the limits of our compute capabilities. (We could go further up to say 10K or 15K parameters by additional low-level memory and cache optimizations, but that would still pale in comparison to modern day networks with millions of parameters --- and in any case, will not be in line with the point we are trying to investigate in these experiments as discussed ahead.)
>
>
>
> 3. Most importantly, our objective behind most of the rank experiments was to **investigate whether our provided formulas match exactly in practice across a large variety of initializations, losses, datasets, (arbitrary) network sizes, etc.**  --- and indeed they match do in every case. *Notice, this is in addition to a tight upper-bound which we have provably established!*
>
> Hence, to claim ‘restricted settings’ as a weakness of our method would be akin to ignoring the objective behind the design of experiments and of course their cost.
>
> $\quad$
>
> **Calculation of exact rank at float64 precision:** The float64 precision means that numbers about as small as 1e-16 can be accurately handled, without incurring an underflow or overflow. We use the default mechanism used by NumPy (as well as MATLAB) to calculate the tolerance value for deciding the eigenvalue to be zero. The details of this can be found here https://numpy.org/doc/stable/reference/generated/numpy.linalg.matrix_rank.html, but briefly for a matrix
>
> $H \in \mathbb{R}^{m\times n}$,  $\quad$ tolerance = $\lambda_{1}(H) \times \operatorname{max}(m, n) \times \epsilon$,
>
> where $\lambda_{1}(H)$ is the maximum eigenvalue, and $\epsilon = 2^{-52} \approx 2.22e-16$.
>
> $\quad$
>
>
> **Numerical errors in Hessian and SVD:** Since we compute exactly the full Hessian and full SVD --- without any approximations --- there should be extremely minimal error if any whatsoever. There is nothing better that can be done further.
>
>
> ---
>
> $\quad$
>
>
> ### Clarity:
>
> $\quad$
>
> Thank you for your comments on clarity, and we are glad to hear that you found the paper to be well-structured. We will surely incorporate your suggestions to improve it further.

---

> ### Author Response · Authors · 2021-08-10
> **Detailed response (3/3)**
>
> ### Minor Comments:
>
> $\quad$
>
> - >**“but the experiments were done on under-parameterized networks.”**
>
>   1) No, this is not true, and rather, the opposite is true: almost all of our experiments are on over-parameterized networks --- for precisely the reason you mention.
>   2) It is most clear in e.g. Fig 2a, where the network has hidden layers 50,20,20,20 and overall p=5200 parameters. We show the rank results for various values of # samples N, from as small as N=4 to N=400.
>   3) Choosing $N\geq \operatorname{rank}(\Sigma)$ does not change anything, hence we do not calculate for higher values of $N$.
>
>   *Thus, there is no mismatch between theory and experiments.* We will also clarify this aspect in the paper.
>
> $\quad$
>
> - **Results during training due to interactions between functional Hessian and outer-product Hessian:**
>
>   1) Our Hessian rank upper-bound at initialization (Line 202) was a small additive factor off from the true rank at initialization (as expressed by the formula in Line 203), since we ignored the intersection between the column spaces of the functional Hessian and the outer-product Hessian for the purposes of the upper-bound.
>
>   2) *But actually, empirically we see that during the course of training this gap even shrinks further, as shown in this figure https://imgur.com/a/p8KKryN.* This would mean that the overlap between the column spaces of the functional Hessian and outer-product is becoming smaller than that at initialization, and at the convergence (to the minimum) this gap goes to zero as the functional Hessian approaches zero.
>
>    3) Hence, not only does it confirm that our upper bound on the Hessian rank remains valid throughout training, **but it also shows that the upper bound will only get tighter during the training**, and at convergence matches exactly with the true rank. (also check out the general response on “Validity of rank results during training”)
>
>    (NB: The expression of our upper bound is entirely in terms of the rank of weight matrices and residual-input covariance \Omega, so we substitute these values at each point during training to get our corresponding upper bound. As emphasized before, our bounds do not necessarily require maximal rank of weight matrices. Maximal rank assumption only translates the theoretical results into an expression containing the total # of hidden neurons, such as that in Line 202, 203)
>
> $\quad$
>
> - >“did not explain how this hypothetical network relates to the original network”:
>   1) This is solely meant for the purposes of interpreting the formula of rank deficiency, which has a form similar to the number of parameters, but with minimum dimension subtracted from each layer. This interpretation is hinting or suggesting that intuitively there might be a sub-network within the original network that can be entirely removed/pruned, since the original network is rank deficient by that many # of parameters (sub-network size).
> $\quad$
>
>   2) Interestingly, this also bears a semblance of connection to the work on Lottery Ticket Hypothesis. However, for now, this is simply just a way of interpreting the rank-deficiency formula --- that is why we call it a ‘hypothetical’ network, but it would be a very exciting future direction to firmly establish this connection.
>
> $\quad$
>
> - **“predictions of the bulk size” in Figure 3 caption:** We meant the size of the bulk at zero, i.e.,  the set of zero eigenvalues --- which is nothing but the rank-deficiency. We will add this clarification.
>
> $\quad$
>
> - **rank/#params capturing fraction of effective parameters:** This should be clear from our general response "Rank as # effective parameters."
>
> $\quad$
>
> - **“​​using a much smaller value, e.g., 200, will also result in a good reconstruction of the Hessian”:** No, this isn’t the case. A part of the reason is that Figure 4 is not adequately zoomed-in. The other part is because Figure 4 only illustrates a single setting of MSE loss with ReLU activation and hence does not impart the complete picture. *Our rank estimate proves robust across various activation functions and loss functions, as discussed in the general response “Validity of rank results during training".*
>
> $\quad$
>
> - **Line 307, Corollary 10:** Yes, it means “global minimum with respect to the population loss". And yes, it needs the functional Hessian to be of rank 0. This is probably not as clear from the right subfigure of Figure 5 and lines 299-302,  and we should have said it explicitly. We will clarify this by emphasizing this part again near Corollary 10 itself.
>
>
> ----
>
> $\quad$
>
> ### Related works:
> Thanks for sharing those references. We will add them in our final version.
>
> $\quad$
>
> ----
>
> $\quad$
>
> ### Limitations Section
>
> Please refer to the general response.
>
> $\quad$
>
> ---
>
> $\quad$
>
>
> *We hope that our above response comprehensively answers the questions raised in your comprehensive review. Hence, we hope that you will accordingly consider reassessing your evaluation and revising your scores.* Also, please do not hesitate to ask further in case you may have more questions or comments.

---

> ### Author Response · Authors · 2021-08-20
> **Thanks for reading our response and reply to remaining concerns**
>
> Thank you so much for going through our response. We are pleased to hear that it has addressed some of your concerns, in particular, that you agree it makes more sense to consider rank as the inherent complexity measure of neural networks.
>
> We would like to take this opportunity and add some quick comments on the points you raised.
>
> **Energy of spikes vs bulk:** It is indeed true that in the particular example we shared for the small network, the set of spikes capture more energy of the spectrum than the bulk. However, this is **precisely because it is a small network**, and so the spikes have more energy than the bulk. To illustrate this point, we present additional results that better indicate how the energy of the spectrum gets divided between them, as follows.
>
> | **Network hidden layers** | **% of energy captured by spikes** | **% of energy captured by bulk** |
> |---------------------------|------------------------------------|----------------------------------|
> | 20, 20                    | 51.17 %                            | 48.83 %                          |
> | 50, 50                    | 41.06 %                            | 58.94 %                          |
> | 100, 100                  | 32.36 %                            | 67.64 %                          |
>
> Hence, it is evident that with increasing network size, the bulk in fact captures much more energy than the set of spikes (e.g. 2/3 of energy with hidden layer sizes 100, 100). The trend should similarly continue for even bigger networks, and thus we believe that --- especially for the usual case of large over-parameterized networks --- the bulk carries substantial information.
>
> $\quad$
>
> ----
>
> $\quad$
>
> **Hessian rank and effective # of parameters:**
>
> - We agree that quadratic approximation will not be 100% accurate as the landscape is in general highly non-convex (this is a great remark!). However, we would like to point out several recent works [1, 2, 3] which reveal a more refined picture of this aspect.
>
>   - Consider [1], where authors empirically find that the loss landscapes seem to be partitioned into well-defined regions of convex contours (Figure 5) surrounded by regions with non-convex contours and how typical initialization (such as the Glorot initialization) *launches the network in the well-defined convex regions, which may result in the optimization algorithms never “seeing” the pathological non-convexities.*
>
>   - More quantitatively, see Figure 2 in [2], which shows how after a few gradient steps, the Hessian becomes aligned and thus proportional to the Fisher, and so is *essentially positive semi-definite*.
>
>   - Also [3] has noted that the network, during the optimization path, manoeuvers through *regions of positive Hessian curvature ---  importantly at initialization and the end of training* (Figure 1), and further this convex behaviour is shown to increase with width.
>
>   - As a matter of fact, we are also concerned with the Hessian rank at initialization. Therefore, in light of [1, 2, 3],  using the quadratic problem as an analogy, i.e.,  considering a quadratic approximation --- is not as far-fetched an assumption as one might think at first.
>
> $\quad$
> - Yet, we completely understand that there is still room to establish the connection between rank and effective # of parameters even more thoroughly. As you can see, from the beginning to end of the paper (and even the title itself), our primary objective as well as all of our claims are about rigorously proving results on the rank of the Hessian --- including completely novel and interpretable rank formulas which are applicable for arbitrary-sized networks and how they relate to the inherent compositional structure of neural networks.
>
> Despite the fact that we noted this very important connection between rank and effective # parameters and how it can provide insights into the inherent complexity of neural network parameterization --- but **nowhere in the paper did we claim that we have proved the effective # of parameters**. We are indeed working on this end as a follow-up paper, however, we genuinely believe that it is beyond the scope of the current paper, and that we already deliver more than sufficient contributions to merit acceptance.
>
> [1] Li et. al., Visualizing the Loss Landscape of Neural Nets, NeurIPS 2018 https://papers.nips.cc/paper/2018/file/a41b3bb3e6b050b6c9067c67f663b915-Paper.pdf
>
> [2] V. Thomas et. al.,  On the interplay between noise and curvature and its effect on optimization and generalization, AISTATS 2020. https://arxiv.org/pdf/1906.07774.pdf
>
> [3] E. Littwin & L. Wolf, On the Convex Behavior of Deep Neural Networks in Relation to the Layers' Width, ICML 2019 Workshop Deep Phenomena https://openreview.net/pdf?id=rJgzwVH2nE.
>
> $\quad$
>
> ----
>
> $\quad$
>
> Hope this response clarifies your remaining concerns as well. We are more than happy to take any further comments or suggestions.

---

### Official Review · Reviewer_8A13 · 2021-07-14

**Rating:** 8
**Confidence:** 3

**Summary:**

This paper derives tight upper bounds on the Hessian rank for deep linear networks. They show that these rank formulae (derived for the linear case) faithfully capture the numerical rank in the non-linear regime. They also demonstrate that these rank bounds hold during the course of training.

**Limitations And Societal Impact:**

The work is largely theoretical, so considerations about societal impact do not directly apply.

But there is no discussion on the limitations of their method except that it is theoretically unable to capture the Hessian rank with non-linear activation functions (although empirical results suggest that it is able to capture the same).

I believe that the limitations of the current work will hold for all the methods (i.e. not specifically for this paper) that analyze the linear case and extend it to the non-linear case. A discussion about the same should be added to the paper for completeness.

**Main Review:**

1. The paper is well written and easy to understand.
2. The paper derives a closed form expression for the Hessian rank that comes with a tightness guarantee for deep linear networks.
3. Using the above expression, the authors show that the rank effectively depends on the sum of hidden layer widths while the number of parameters is proportional to the sum of hidden layer widths squared.
4. Using the rank as a proxy for the effective number of parameters, the work can prove to be useful to understand the links between generalization and flatness of minima. And also to understand various phenomenon related to overparametrization such as double descent.
5. The authors empirically show that the rank (derived for the linear case) also captures the numerical rank when the activation functions are non-linear.

The paper addresses an important problem, is well written and provides an interesting new perspective on the effective number of parameters for deep neural networks. I recommend acceptance.

**Time Spent Reviewing:**

9

---

> ### Author Response · Authors · 2021-08-10
> **Detailed response**
>
> Thank you for spending a significant amount of time reviewing our paper and sharing your feedback. We are very pleased to hear that you recognize the importance of the problem and that our contribution provides an interesting new perspective for deep neural networks.
>
> We agree with your point about writing a separate limitation section. (Also, check the general response on that.)
>
> Lastly, please feel free to raise any questions/feedback that you might have.

---

### Official Review · Reviewer_9bPu · 2021-07-17

**Rating:** 6
**Confidence:** 3

**Summary:**

This paper studies the rank of Hessian for deep linear networks. Theoretically, with squared loss, and assuming the weight matrices are full-rank, this paper provides rank upper bound for outer-product, functional and full Hessian, and empirically verifies that the bounds are almost tight. This paper further shows empirically that the numerical rank with nonlinear networks can still be captured by the linear rank bound, and that the Hessian rank decreases during training. Finally, Hessian degeneracy results for 1-hidden-layer networks and extension to the case with bias are also provided.

**Limitations And Societal Impact:**

The limitations are discussed above.

**Main Review:**

I think the rank bounds for deep linear networks are very nice, and it is also amazing that in Figure 2, the true ranks match the predictions so well.

On the other hand, I have the following concerns:
1. As also mentioned in this paper, in many settings, not only the rank matters and we also need to consider the size of eigenvalues. For example, Figure 4 tries to show that using the given rank formula, we can reconstruct the Hessian with 0.002% error; however from the same graph, we can get a small reconstruction error as long as we use more than 500 eigenvalues, while the rank formula gives a value between 1000 and 1500. Therefore I think the evidence is not strong enough to support the claim that the rank formula truly captures the numerical rank in practice.
2. Moreover, I think the tightness of the rank bounds may be partly due to random initialization. For example, as shown in Figure 5, even though the rank of each weight matrix stays constant, the Hessian rank drops a lot. I think this may suggest something like the eigenvalues of each weight matrix become more concentrated; in fact such a result has been shown for linearly separable data with the exp or logistic loss [1,2]. In other words, I am worried that the Hessian rank formula may soon become inaccurate during training.

Update after rebuttal: Thank you for your response. It is nice that the analysis can still be applied throughout training, and the empirical results look promising. I will increase my score by 1.

References:
[1] Gunasekar, Suriya, Jason Lee, Daniel Soudry, and Nathan Srebro. "Implicit bias of gradient descent on linear convolutional networks." arXiv preprint arXiv:1806.00468 (2018).
[2] Ji, Ziwei, and Matus Telgarsky. "Gradient descent aligns the layers of deep linear networks." arXiv preprint arXiv:1810.02032 (2018).

**Time Spent Reviewing:**

8

---

> ### Author Response · Authors · 2021-08-10
> **Detailed Response**
>
> Thank you for taking the time to review and sharing your reasonable concerns. We are also happy to know that you find the rank bounds very nice. We answer your concerns in detail as follows:
>
> $\quad$
>
> **Capturing numerical rank:** We understand your sentiment here, but let us explain our case in a better way:
>
> 1) *Numerical rank must be seen, firstly, in regards to the actual/true rank:* In Figure 4, we can see that the true rank (i.e., empirically measured rank of the non-linear network at Float64 precision) is significantly higher than the threshold provided by the linear rank. So technically, the linear rank threshold does in fact correspond to numerical rank.
>
>
> 2) *Non-dominating eigenvalues still present, yet a useful apriori indicator:* We agree that a smaller value than the linear rank, e.g., here 500, would correspond to a more representative threshold to measure numerical rank. However, one has to realize that such a representative threshold is not available apriori to computing the entire Hessian spectrum (which is essentially out of the question, in practice, for most networks as the cost is cubic in # parameters). As a result, the linear rank threshold still provides a useful rule of thumb for practitioners.
>
>
> 3) *Leading term in the rank result, $qM$, can give a more ‘practically oriented’ numerical rank:* If the practitioner would like to trade-off precision for efficiency, then one option is to only consider this leading order term of $qM$. E.g., in Figure 4, # hidden neurons $M = 30+20 = 50$, $q =$ # classes $K = 10$, and so $qM = 500$. Not only here, but take Figure S23, where $M = 50+40+30 = 120$, $q = 10$, and $qM = 1200$ can be regarded as a more practical numerical rank. Thus, it is possible to derive a helpful practical indication by using the linear rank threshold upto first order.
>
>
> 4) *Reliable indicator across different loss types and activation functions:* This is a very important aspect that was not adequately emphasized in the paper. Please check out the general response “Robust reconstruction with linear rank threshold”.
>
> Overall, we hence believe that using the linear rank threshold, or only up to first order, i.e., qM, can indeed serve as a reliable indicator of numerical rank in the non-linear case --- across a variety of loss functions, activation functions, architectures, datasets. *Especially, considering that a precise numerical rank indicator is being made available for the first time ever in the literature.*
>
> $\quad$
>
> **Tightness of rank bounds due to initialization & drop in Hessian rank:**
>
> 1) What initialization guarantees is that the weight matrices have maximal rank. But as remarked by **Reviewer hhmf**,
> >  the upper bounds in Theorem 3 and in Theorem 5 should still holds even when the matrices W^l are not maximal-rank during training
>
>    Even if the weight matrices do not have maximal rank, the rank results from Theorem 3 and Theorem 5 would still apply *as these theorems and their analysis themselves do not depend on the weight matrices being maximal rank.*
>    In such a case, the bounds just require plugging in the particular value of the rank for each of the weight matrices instead of the maximal rank value. Thus the tightness of the rank bounds is not contingent on the weight matrices being maximal/full rank at initialization. For more details and an empirical verification, please refer to the section “Validity of rank results during training” in the general response.
>
>
> 2) The drop in the Hessian rank comes **solely due to the decrease in the rank of the residual-input covariance matrix $\Omega$**, since the residuals approach zero during the course of training. To illustrate this point, one can plot the functional Hessian by fixing the $\Omega$ matrix to the one at initialization while still using the weight matrices at the respective point in training, whereby we find that rank remains constant throughout (see Figure https://imgur.com/a/VayNcW2). Besides, the rank of the outer-product Hessian also remains constant. Thus, the drop in Hessian rank is brought about only due to the residual going to zero in this case.
>
>
> 3) The concentration of weight matrices does not seem to be happening in this case. If it were so, then the rank of the Hessian outer-product would at least show some change during the course of training, because remember $\operatorname{rank}(H^o) = r \operatorname{rank}(W^{2:L}) + K \operatorname{rank}(W^{L−1:1}) − \operatorname{rank}(W^{2:L}) \operatorname{rank}(W^{L−1:1})$. Perhaps the result from [1, 2] does not apply here since we have MSE loss and the data is not linearly separable. However, even otherwise, say the weight matrices do indeed concentrate (like to rank-1 in the mentioned references or so), one can simply substitute them in the rank upper-bounds, as mentioned in point 1.
>
> $\quad$
>
> *We hope that the above response addresses your concerns and that you will consider updating your score. Please feel free to ask if you still have more questions/suggestions.*

---

> > ### Comment · Reviewer_9bPu · 2021-08-24
> > **Thank you for your response!**
> >
> > It is nice that the analysis can still be applied throughout training, and the empirical results look promising. I will increase my score by 1.

---

### Official Review · Reviewer_hhmf · 2021-07-20

**Rating:** 8
**Confidence:** 3

**Summary:**

This paper provides several results on the rank of the Hessian of neural networks. The main results are:

(A) upper bounds on the rank of the Hessian of a linear neural network (i.e., identity activations). These are proved by separating the Hessian into the sum of the “outer-product Hessian” and the “functional Hessian”, and bounding the rank of these separately.
(B) a conjectured formula for the exact Hessian rank that matches experiments, and is also quite close to the upper bounds.
(C) experimental evidence that these results essentially hold for neural networks with nonlinear activations as well, even after training.
(D) for 1-hidden layer nonlinear networks, under a mild assumption a very weak (but still nontrivial) bound on the rank is obtained.


**Limitations And Societal Impact:**

The authors do not have a section with the limitations of their work, but I am not sure it is necessary due to the theoretical nature of the paper and the clarity of the presentation of the results.

**Main Review:**

The paper is well-structured and clear. As far as I can tell, the proofs are correct. The takeaway of the results that I find most interesting is the following: let the network layers be of width M_0, M_1,…,M_L. Then the rank of the Hessian of a linear network is upper-bounded by O((min_i M_i) * (sum_{i=0}^L M_i)). This is an improvement over the trivial O(sum_{i=1}^L M_i M_{i-1}), which is the number of parameters. Furthermore, the proofs demystify this low-rank phenomenon: it is due to the compositional structure of the neural network map. Most of the paper is dedicated to proving these facts about the rank of the Hessian and confirming empirically that they generalize to networks with nonlinear activations. At the end, there is an interesting discussion of some the implications of these findings.

I have some questions about the results, but overall found the paper to be a pleasant read and think that it would be informative to attendees and a good fit for the conference.

* Could you elaborate on the remark on Lines 176-178. Is this supposed to be a rigorous statement? If so, can you give a reference?

* Lines 194-195: “This phenomenon is quite straightforward to see in a 2-layer network and there 195 Corollary 6 is an equality.” Please add a discussion of this to the supplementary materials?

* Line 295, you write “An implication of this result is that our upper bounds on the Hessian rank remain valid throughout the training”.
Why is this an implication of the result? It seems that the upper bounds in Theorem 3 and in Theorem 5 should still holds even when the matrices W^l are not maximal-rank during training. So it seems that the only implication of this result is that the outer-product Hessian rank stays constant.

* Lines 354-355, (i) Overparametrization. “Since rank intuitively captures the notion of effective parameters, it could be a possible alternative to benchmark overparameterization, e.g. for double descent”. <— I have a few questions about this: (1) this paper seems to show that the rank is roughly the square root of the number of parameters, so wouldn’t this still be much more than the number of data points for some large network architectures? (2) Can you make this sentence more precise? How is the rank of the Hessian capturing the amount of effective parameters besides the  fact that minima are flat, which is discussed in the next paragraph Lines 356-361?

Typos
* Line 93, “where the l-th layer map F^l” -> “where the l-th layer map is F^l”
* Line 214, “input-covariance has rank” -> “input-covariance has full rank”
* Line 218, “all layer-widths have been” -> “all layer-widths have all been”
* The bibliography seems to be missing from the supplementary materials. For example, I could not find reference [61].
* Line 323, “While these bounds” -> “These bounds”


**Time Spent Reviewing:**

3

---

> ### Author Response · Authors · 2021-08-10
> **Detailed Response**
>
> Thanks for taking the time to review and for giving useful feedback. We are also glad to hear that you found it a pleasant read. To elaborate on your comments,
>
> $\quad$
>
> **NTK (Lines 176-178):** Yes, this is indeed supposed to be a rigorous statement (for MSE loss). To clarify this aspect, we will express this as a corollary and provide the detailed proof in the supplementary. But briefly, the main idea is that, for some matrix $A$, $\operatorname{rank}(A) = \operatorname{rank}(A^\top A) = \operatorname{rank} (A A^\top)$. Now, take $A$ as the Jacobian of the network function $\nabla_\theta F$ shaped as $p \times KN$. So, we have that $\operatorname{rank}(\nabla_\theta F) = \operatorname{rank}({\nabla_\theta F}^\top \nabla_\theta F) = \operatorname{rank}(\nabla_\theta F {\nabla_\theta F}^\top)$, where the matrix ${\nabla_\theta F}^\top \nabla_\theta F$ is the NTK matrix and $\nabla_\theta F {\nabla_\theta F}^\top$ is the outer-product Hessian $H_o$.
> In case of multiple classes ($K>1$), we are assuming that NTK tensor is 'flattened' as often considered in the literature, e.g., [5].
>
> [5] Fan & Wang, NeurIPS 2020. [Spectra of the Conjugate Kernel and Neural Tangent Kernel for Linear-Width Neural Networks](https://proceedings.neurips.cc/paper/2020/file/572201a4497b0b9f02d4f279b09ec30d-Paper.pdf).
>
> $\quad$
>
> **Equality for 2-layer Functional Hessian (Lines 195)**: Yes, we will surely add this in the supplementary. Very briefly, this is because in the 2-layer case the functional Hessian is block-hollow (or block anti-diagonal). Thus, naturally, the block column spaces do not intersect.
>
> $\quad$
>
> **Implication of upper bound (Line 295):** Thanks for your comment, this is a great point and we will update it to be more clear here. What we wanted to say was that the upper bounds on the Hessian rank, at initialization,  --- which are in the form of $\mathcal{O}(qM)$ or $\mathcal{O}(\sqrt{p})$ where $p$ is the # parameters --- remain valid. However, as you have correctly noted that Theorems 3 and 5 will hold even if the rank of weight matrices decreases, but where the definition of q has to be updated to $\operatorname{min}(\operatorname{rank}(W^1), ..., \operatorname{rank}(W^L))$ and then the resulting Hessian rank bounds will also be in terms of the rank of involved matrices. Also, we would refer you to the section “Validity of rank results during training” in the general response, where we will also demonstrate this empirically.
>
> The other implication, as you said, is that the rank of the outer-product Hessian remains constant, while the rank of the functional Hessian (also the loss Hessian) decreases only due to the residual-input covariance $\Omega \rightarrow 0$ as training progresses.
>
> $\quad$
>
> **Overparametrization (Lines 354-355):**
>
> (1)  whether rank > # data points N for large architecture:
>
> -  Yes, very large networks will indeed still end up as being over-parameterized --- even in the refined sense of $\operatorname{rank} > N$. However, a significant number of networks that are currently deemed as over-parameterized --- in the vanilla sense of  $p > N$ will not essentially be over-parameterized. (E.g., many networks used in practice on CIFAR10 (N=50K) have between a million or 10 million parameters.)
>
>
> -  Additionally, since the submission deadline, we have been able to properly establish this connection between double descent and rank, and which we will be writing as a separate paper. The connection requires more components but involves a subtle usage of rank.
>
>
> - Anyhow, this is still a very new line of research with many interesting open questions in regards to the phenomenology of rank and over-parameterization. For instance, another important question is can we reparameterize neural networks so that we do not have to optimize over the ‘ineffective’ parameters in the sense of rank.
>
> For (2), please refer to the general responses, “Rank as effective # params”.
>
> $\quad$
>
> We hope this response answers your questions, and in light of this, you will consider updating your score. Please feel free to comment if you still have more questions/suggestions.

---

### Author Response · Authors · 2021-08-10
**General response to reviewers:**

## Rank as # effective parameters

$\quad$

### A. Via Quadratic problem
Let us illustrate how rank implies the effective number of parameters (or alternatively, degrees of freedom) using a simple example.

Consider a quadratic model $f(\theta) =\frac{1}{2} \theta^\top M \theta$, for $\theta \in \mathbb{R}^p$ and some given symmetric matrix $M \in \mathbb{R}^{p \times p}$. We would like to minimize this quadratic objective in $\theta$. The gradient of $f$ is given by $M \theta$, while the Hessian is $M$.

Assume $M$ has rank $r$. Then using the spectral theorem, we have $M= UDU^\top$, and let $D = \operatorname{diag}(\lambda_1, \cdots, \lambda_r, 0, \cdots, 0)$, with eigenvalues sorted in descending order from $\lambda_1$ to $\lambda_r$. Let us reparameterize the system $\theta \mapsto U^\top \theta$. Then in this new parameterization, denoted by $\widetilde{\theta}$, we can equivalently write the objective as,

$$\quad \quad \quad f(\widetilde{\theta}) = \frac{1}{2}{\widetilde{\theta}}^\top  D \widetilde{\theta} = \frac{1}{2}\sum_{i=1}^{r} \lambda_i \widetilde{\theta}_i^2 $$

Thus, it is clear due to the low-rank nature of $M$, that we have effectively `r` parameters instead of `p`. Even if optimized the full objective $f(\widetilde{\theta}) = \frac{1}{2}{\widetilde{\theta}}^\top  D \widetilde{\theta}$ with gradient descent, notice that the gradient is  $\nabla_{\widetilde{\theta}} f = D \widetilde{\theta}$. Since $D$ has rank $r$, all the coordinates of $\widetilde{\theta}$ from $r+1$ to $p$ will not be updated by gradient descent and remain stuck at their initial value.

*Here, it is strikingly clear that $\operatorname{rank}(M)$ gives the effective # of parameters.*

**Relation to neural networks:** One might wonder how the quadratic problem is relevant in regards to the optimization landscape of neural networks. However, as noted in [1], one obtains the quadratic objective by linearizing the network function and considering a second-order Taylor series approximation to the loss function. Thus, in the case of neural networks, the Hessian $H_\mathcal{L}$ will play the role of $M$.

NB: Besides [1], many other papers [2, 3] commonly use this problem as a model to motivate and derive their approach.

[1] Zhang et. al., NeurIPS 2019. Which Algorithmic Choices Matter at Which Batch
Sizes? Insights From a Noisy Quadratic Model https://arxiv.org/pdf/1907.04164.pdf

[2] Schaul et. al. ICML 2013 No more pesky learning rates. http://proceedings.mlr.press/v28/schaul13.pdf

[3] von Oswald et. al., ICLR 2021, Neural networks with late-phase weights,  https://openreview.net/forum?id=C0qJUx5dxFb

$\quad$

### B. Bayesian view

The rank can also be seen as measuring the effective # of parameters from a Bayesian view, as derived from the Occam factor (ratio of posterior accessible volume in parameter space to the prior accessible volume) in [4]. The details can be found in this reference, but this essentially delivers the following quantity $\gamma$ as a measure of effective # of parameters:
$$\gamma = \sum_{i=1}^p \frac{\lambda_i}{\lambda_i + \epsilon}$$

Here, $\epsilon$ denotes the strength of external regularization and $\lambda_i$ are the eigenvalues of the Hessian $H_\mathcal{L}$. The important thing to note is that as $\epsilon \rightarrow 0$, we obtain that $\gamma \rightarrow \operatorname{rank}(H_\mathcal{L})$.

(Besides, this approach has also received interest lately in https://openreview.net/forum?id=eBHq5irt-tk, where the authors correlate it with double descent)

[4] Mackay, NeurIPS 1991. Bayesian Model Comparison and Backprop Nets

---
$\quad$

## Numerical Rank and relaxations

$\quad$

**Numerical rank**.  In our quadratic problem example, now consider that the eigenvalues of D have a long tail. Say, $D = \operatorname{diag}(\lambda_1, ... , \lambda_{r^\star}, \lambda_{r^\star + 1}, … \lambda_r, 0, … 0)$, where $\lambda_{r^\star +1 }$ to $\lambda_r$ are tiny but non-zero eigenvalues.

Now, since in the objective, the parameters are weighted by the respective eigenvalue, we can for most practical purposes ignore these non-dominating parameters, i.e.,

$$\quad \quad \quad f(\widetilde{\theta}) \approx \frac{1}{2}\sum_{i=1}^{r^{\star}} \lambda_i \widetilde{\theta}_i^2 $$

Thus, numerical rank can be seen as effective # of parameters that are dominant. Besides, there are a few important things to note:
- Rank $r$ still remains a valid upper bound to the effective # of params.
- Obtaining the ‘ideal’ numerical rank of $r^\star$ requires knowing the threshold $\lambda_{r^\star}$. But *this threshold is highly dependent on the problem*, and so it is immensely challenging to choose apriori without looking at the eigenspectrum (and for large # of parameters computing the entire spectrum would be intractable). This makes it much more difficult to obtain precise theoretical statements on numerical rank.

Thus, the dominating # of effective parameters might still be less than the exact Hessian rank. In fact, the linear rank threshold is indeed significantly smaller than the true rank calculated at machine precision for networks with non-linearities. However, as evident from the reconstruction figures across different loss functions and activation functions (see the “Robust reconstruction” section below), *the actual numerical rank is quite likely to be of the same order as the linear rank threshold, i.e., $\mathcal{O}(qM)$ or in other words $\mathcal{O}(\sqrt{p})$*.

**Relaxations.** Finally, we would like to remark that there are additional ways to obtain relaxations to rank. One such is the $\gamma$ discussed in the above section “Bayesian view”. Another way, as suggested in our paper, is the ratio of nuclear to spectral norms of the Hessian, which will also be a lower bound to the Hessian rank.

These can be more easier to use from a practical point of view, but still possess useful theoretical guarantees thanks to the rank bounds. In fact, it is precisely the **discrete nature of rank that allows us to get neatly interpretable formulas and bounds**.

----
$\quad$

## Validity of rank results during training:

$\quad$

We would like to clarify and emphasize that our rank upper bounds and formulas remain valid throughout the course of training --- as pointed by **reviewer hhmf**,
>“It seems that the upper bounds in Theorem 3 and in Theorem 5 should still holds even when the matrices W^l are not maximal-rank during training.”

The reason is that inherently our results just use the particular value of the rank of the weight matrices and residual-input covariance matrix $\Omega = E[\delta x^\top]$, but are not dependent on them being full rank. So, while the upper-bounds may not be expressible in terms of the total # of hidden neurons $M$ (as in Line 202) during training, but nevertheless will remain valid throughout training with the corresponding substitution of the ranks of the involved matrices.

In order to demonstrate this aspect, we plot our upper bound to the Hessian rank during the course of training and compare it with the true empirically measured Hessian rank at float64 precision. To get the upper bound at a particular epoch, we simply plug in the value of the rank of $\Omega$, the rank of weight matrices (although this will remain constant throughout), and the values of $q$ and $s$ at that epoch. The resulting figure can be found here https://imgur.com/a/p8KKryN.

This not only verifies the validity of the upper bounds throughout training but *also shows that the upper bound becomes even tighter than what it already was at initialization*. Similarly, our Hessian rank formula also captures the empirically measured Hessian rank during the course of training (by similar substitution of the ranks each epoch) as well as the upper-bound and formula for the functional Hessian and outer-product Hessian respectively.

----
$\quad$

## Robust reconstruction with linear rank threshold
$\quad$

The Hessian reconstruction behaviour presented in Figure 4, which uses MSE loss and ReLU activations, alone might paint a slightly inaccurate impression. Also, the zoomed-out nature of Figure 4 does not help either (check out the zoomed-in plot here, https://imgur.com/a/b53Clxf).

E.g., this can be nicely illustrated by the scenario of Cross-Entropy (CE), see Figure S26, and the zoomed-in plot at ​​https://imgur.com/a/UBLBGfx.  This uses the identical network architecture as in Figure 4, but the behaviour of reconstruction-error is quite different. Using a value such as 500 for CE will not deliver as accurate results (1% error) as it had in Figure 4 for MSE (0.2% error). Further, see Figure S28 and the corresponding zoomed-in plot https://imgur.com/a/mTFc6pC  where Tanh is used in the same setup, but using 500 would be even worse (3% error).

*In contrast, the linear rank threshold provides a robust reconstruction across a variety of activation and loss functions.*

----
$\quad$

## Writing a separate limitations section:
$\quad$

We completely agree with **reviewers 8A13 and bE99** about this aspect. In fact, **reviewer 8A13** has raised a good point that such limitations
> “hold for all the methods (i.e. not specifically for this paper)”

that analyze the linear case. In fact, we are quite upfront about the need to take a step back and rigorously characterize the linear case. The **rigorous analysis of the Hessian rank in the linear case is by all means only a stepping stone** --- which although had long been sidelined --- to understanding the nature of redundancy in the parameterization of general neural networks. We will definitely add a separate section of limitations in the discussion for completeness.

---

### Decision · Program_Chairs · 2021-09-27

**Decision:**

Accept (Poster)

**Comment:**

This paper gives an exact characterization on the rank of the Hessian for linear networks, and show that in many settings the rank can be much smaller than the number of parameters. The paper then verify these claims empirically in nonlinear networks and show that the Hessian is still approximately low rank. Although the reviewers had some concerns initially, their opinions have improved during the author response period where it became clear that the result holds throughout training (not just for minima) and has interesting empirical observations. The authors should really incorporate all the comments given by the reviewers to improve the paper, including explaining why the rank of the Hessian is intuitively related to the number of parameters and explaining connections with related works.